# A Novel Framework for Policy Mirror Descent with General Parameterization and Linear Convergence

**Carlo Alfano**
Department of Statistics
University of Oxford
`carlo.alfano@stats.ox.ac.uk`

**Rui Yuan**
Stellantis, France*
`rui.yuan@stellantis.com`

**Patrick Rebeschini**
Department of Statistics
University of Oxford
`patrick.rebeschini@stats.ox.ac.uk`

## Abstract

Modern policy optimization methods in reinforcement learning, such as TRPO and PPO, owe their success to the use of parameterized policies. However, while theoretical guarantees have been established for this class of algorithms, especially in the tabular setting, the use of general parameterization schemes remains mostly unjustified. In this work, we introduce a framework for policy optimization based on mirror descent that naturally accommodates general parameterizations. The policy class induced by our scheme recovers known classes, e.g., softmax, and generates new ones depending on the choice of mirror map. Using our framework, we obtain the first result that guarantees linear convergence for a policy-gradient-based method involving general parameterization. To demonstrate the ability of our framework to accommodate general parameterization schemes, we provide its sample complexity when using shallow neural networks, show that it represents an improvement upon the previous best results, and empirically validate the effectiveness of our theoretical claims on classic control tasks.

## 1   Introduction

Policy optimization is one of the most widely-used classes of algorithms for reinforcement learning (RL). Among policy optimization techniques, policy gradient (PG) methods [e.g., 92, 87, 49, 6] are gradient-based algorithms that optimize the policy over a parameterized policy class and have emerged as a popular class of algorithms for RL [e.g., 42, 75, 12, 69, 78, 80, 54].

The design of gradient-based policy updates has been key to achieving empirical success in many settings, such as games [9] and autonomous driving [81]. In particular, a class of PG algorithms that has proven successful in practice consists of building updates that include a hard constraint (e.g., a trust region constraint) or a penalty term ensuring that the updated policy does not move too far from the previous one. Two examples of algorithms belonging to this category are trust region policy optimization (TRPO) [78], which imposes a Kullback-Leibler (KL) divergence [53] constraint on its updates, and policy mirror descent (PMD) [e.g. 88, 54, 93, 51, 89], which applies mirror descent (MD) [70] to RL. Shani et al. [82] propose a variant of TRPO that is actually a special case of PMD, thus linking TRPO and PMD.

From a theoretical perspective, motivated by the empirical success of PMD, there is now a concerted effort to develop convergence theories for PMD methods. For instance, it has been established that PMD converges linearly to the global optimum in the tabular setting by using a geometrically increasing step-size [54, 93], by adding entropy regularization [17], and more generally by adding convex regularization [100]. Linear convergence of PMD has also been established for the negative

---

*The work was done when the author was affiliated with Télécom Paris.

37th Conference on Neural Information Processing Systems (NeurIPS 2023).

entropy mirror map in the linear function approximation regime, i.e., for log-linear policies, either by adding entropy regularization [15], or by using a geometrically increasing step-size [20, 2, 98]. The proofs of these results are based on specific policy parameterizations, i.e., tabular and log-linear, while PMD remains mostly unjustified for general policy parameterizations and mirror maps, leaving out important practical cases such as neural networks. In particular, it remains to be seen whether the theoretical results obtained for tabular policy classes transfer to this more general setting.

In this work, we introduce Approximate Mirror Policy Optimization (AMPO), a novel framework designed to incorporate general parameterization into PMD in a theoretically sound manner. In summary, AMPO is an MD-based method that recovers PMD in different settings, such as tabular MDPs, is capable of generating new algorithms by varying the mirror map, and is amenable to theoretical analysis for any parameterization class. Since the MD update can be viewed as a two-step procedure, i.e., a gradient update step on the dual space and a mapping step onto the probability simplex, our starting point is to define the policy class based on this second MD step (Definition 3.1). This policy class recovers the softmax policy class as a special case (Example 3.2) and accommodates any parameterization class, such as tabular, linear, or neural network parameterizations. We then develop an update procedure for this policy class based on MD and PG.

We provide an analysis of AMPO and establish theoretical guarantees that hold for any parameterization class and any mirror map. More specifically, we show that our algorithm enjoys quasi-monotonic improvements (Proposition 4.2), sublinear convergence when the step-size is non-decreasing, and linear convergence when the step-size is geometrically increasing (Theorem 4.3). To the best of our knowledge, AMPO is the first policy-gradient-based method with linear convergence that can accommodate any parameterization class. Furthermore, the convergence rates hold for any choice of mirror map. The generality of our convergence results allows us not only to unify several current best-known results with specific policy parameterizations, i.e., tabular and log-linear, but also to achieve new state-of-the-art convergence rates with neural policies. Tables 1 and 2 in Appendix A.2 provide an overview of our results. We also refer to Appendix A.2 for a thorough literature review.

The key point of our analysis is Lemma 4.1, which allows us to keep track of the errors incurred by the algorithm (Proposition 4.2). It is an application of the three-point descent lemma by [19, Lemma 3.2] (see also Lemma F.2), which is possible thanks to our formulations of the policy class (Definition 3.1) and the policy update (Line 2 of Algorithm 1). The convergence rates of AMPO are obtained by building on Lemma 4.1 and leveraging modern PMD proof techniques [93].

In addition, we show that for a large class of mirror maps, i.e., the $\omega$-potential mirror maps in Definition 3.5, AMPO can be implemented in $\widetilde{\mathcal{O}}(|\mathcal{A}|)$ computations. We give two examples of mirror maps belonging to this class, Examples 3.6 and 3.7, that illustrate the versatility of our framework. Lastly, we examine the important case of shallow neural network parameterization both theoretically and empirically. In this setting, we provide the sample complexity of AMPO, i.e., $\widetilde{\mathcal{O}}(\varepsilon^{-4})$ (Corollary 4.5), and show how it improves upon previous results.

## 2 Preliminaries

Let $\mathcal{M} = (\mathcal{S}, \mathcal{A}, P, r, \gamma, \mu)$ be a discounted Markov Decision Process (MDP), where $\mathcal{S}$ is a possibly infinite state space, $\mathcal{A}$ is a finite action space, $P(s' \mid s, a)$ is the transition probability from state $s$ to $s'$ under action $a$, $r(s, a) \in [0, 1]$ is a reward function, $\gamma$ is a discount factor, and $\mu$ is a target state distribution. The behavior of an agent on an MDP is then modeled by a *policy* $\pi \in (\Delta(\mathcal{A}))^{\mathcal{S}}$, where $a \sim \pi(\cdot \mid s)$ is the density of the distribution over actions at state $s \in \mathcal{S}$, and $\Delta(\mathcal{A})$ is the probability simplex over $\mathcal{A}$.

Given a policy $\pi$, let $V^{\pi} : \mathcal{S} \to \mathbb{R}$ denote the associated *value function*. Letting $s_t$ and $a_t$ be the current state and action at time $t$, the value function $V^{\pi}$ is defined as the expected discounted cumulative reward with the initial state $s_0 = s$, namely,

$$V^{\pi}(s) := \mathbb{E}_{a_t \sim \pi(\cdot|s_t), s_{t+1} \sim P(\cdot|s_t, a_t)} \left[ \sum_{t=0}^{\infty} \gamma^t r(s_t, a_t) \mid \pi, s_0 = s \right].$$

Now letting $V^{\pi}(\mu) := \mathbb{E}_{s \sim \mu}[V^{\pi}(s)]$, our objective is for the agent to find an optimal policy

$$\pi^{\star} \in \operatorname{argmax}_{\pi \in (\Delta(\mathcal{A}))^{\mathcal{S}}} V^{\pi}(\mu). \tag{1}$$

As with the value function, for each pair $(s, a) \in \mathcal{S} \times \mathcal{A}$, the state-action value function, or *Q-function*, associated with a policy $\pi$ is defined as

$$Q^\pi(s, a) := \mathbb{E}_{a_t \sim \pi(\cdot|s_t), s_{t+1} \sim P(\cdot|s_t, a_t)}\Big[\sum\nolimits_{t=0}^\infty \gamma^t r(s_t, a_t) \mid \pi, s_0 = s, a_0 = a\Big].$$

We also define the discounted state visitation distribution by

$$d_\mu^\pi(s) := (1 - \gamma)\mathbb{E}_{s_0 \sim \mu}\Big[\sum\nolimits_{t=0}^\infty \gamma^t P(s_t = s \mid \pi, s_0)\Big], \tag{2}$$

where $P(s_t = s \mid \pi, s_0)$ represents the probability of the agent being in state $s$ at time $t$ when following policy $\pi$ and starting from $s_0$. The probability $d_\mu^\pi(s)$ represents the time spent on state $s$ when following policy $\pi$.

The gradient of the value function $V^\pi(\mu)$ with respect to the policy is given by the policy gradient theorem (PGT) [87]:

$$\nabla_s V^\pi(\mu) := \frac{\partial V^\pi(\mu)}{\partial \pi(\cdot \mid s)} = \frac{1}{1 - \gamma} d_\mu^\pi(s) Q^\pi(s, \cdot). \tag{3}$$

## 2.1 Mirror descent

The first tools we recall from the mirror descent (MD) framework are mirror maps and Bregman divergences [14, Chapter 4]. Let $\mathcal{Y} \subseteq \mathbb{R}^{|\mathcal{A}|}$ be a convex set. A *mirror map* $h : \mathcal{Y} \to \mathbb{R}$ is a strictly convex, continuously differentiable and essentially smooth function[1] such that $\nabla h(\mathcal{Y}) = \mathbb{R}^{|\mathcal{A}|}$. The convex conjugate of $h$, denoted by $h^*$, is given by

$$h^*(x^*) := \sup_{x \in \mathcal{Y}} \langle x^*, x \rangle - h(x), \quad x^* \in \mathbb{R}^{|\mathcal{A}|}.$$

The gradient of the mirror map $\nabla h : \mathcal{Y} \to \mathbb{R}^{|\mathcal{A}|}$ allows to map objects from the primal space $\mathcal{Y}$ to its dual space $\mathbb{R}^{|\mathcal{A}|}$, $x \mapsto \nabla h(x)$, and viceversa for $\nabla h^*$, i.e., $x^* \mapsto \nabla h^*(x^*)$. In particular, from $\nabla h(\mathcal{Y}) = \mathbb{R}^{|\mathcal{A}|}$, we have: for all $(x, x^*) \in \mathcal{Y} \times \mathbb{R}^{|\mathcal{A}|}$,

$$x = \nabla h^*(\nabla h(x)) \quad \text{and} \quad x^* = \nabla h(\nabla h^*(x^*)). \tag{4}$$

Furthermore, the mirror map $h$ induces a *Bregman divergence* [13, 18] , defined as

$$\mathcal{D}_h(x, y) := h(x) - h(y) - \langle \nabla h(y), x - y \rangle,$$

where $\mathcal{D}_h(x, y) \geq 0$ for all $x, y \in \mathcal{Y}$. We can now present the standard MD algorithm [70, 14]. Let $\mathcal{X} \subseteq \mathcal{Y}$ be a convex set and $V : \mathcal{X} \to \mathbb{R}$ be a differentiable function. The MD algorithm can be formalized[2] as the following iterative procedure in order to solve the minimization problem $\min_{x \in \mathcal{X}} V(x)$: for all $t \geq 0$,

$$y^{t+1} = \nabla h(x^t) - \eta_t \nabla V(x)|_{x=x^t}, \tag{5}$$

$$x^{t+1} = \text{Proj}_{\mathcal{X}}^h(\nabla h^*(y^{t+1})), \tag{6}$$

where $\eta_t$ is set according to a step-size schedule $(\eta_t)_{t \geq 0}$ and $\text{Proj}_{\mathcal{X}}^h(\cdot)$ is the *Bregman projection*

$$\text{Proj}_{\mathcal{X}}^h(y) := \text{argmin}_{x \in \mathcal{X}} \mathcal{D}_h(x, y). \tag{7}$$

Precisely, at time $t$, $x^t \in \mathcal{X}$ is mapped to the dual space through $\nabla h(\cdot)$, where a gradient step is performed as in (5) to obtain $y^{t+1}$. The next step is to map $y^{t+1}$ back in the primal space using $\nabla h^*(\cdot)$. In case $\nabla h^*(y^{t+1})$ does not belong to $\mathcal{X}$, it is projected as in (6).

# 3 Approximate Mirror Policy Optimization

The starting point of our novel framework is the introduction of a novel parameterized policy class based on the Bregman projection expression recalled in (7).

---

[1] $h$ is essentially smooth if $\lim_{x \to \partial \mathcal{Y}} \|\nabla h(x)\|_2 = +\infty$, where $\partial \mathcal{Y}$ denotes the boundary of $\mathcal{Y}$.
[2] See a different formulation of MD in (11) and in Appendix B (Lemma B.1).

---

**Algorithm 1:** Approximate Mirror Policy Optimization

**Input:** Initial policy $\pi^0$, mirror map $h$, parameterization class $\mathcal{F}^\Theta$, iteration number $T$, step-size schedule $(\eta_t)_{t \geq 0}$, state-action distribution sequence $(v^t)_{t \geq 0}$.

**for** $t = 0$ **to** $T - 1$ **do**

1      Obtain $\theta^{t+1} \in \Theta$ such that $\theta^{t+1} \in \text{argmin}_{\theta \in \Theta} \left\| f^\theta - Q^t - \eta_t^{-1} \nabla h(\pi^t) \right\|_{L_2(v^t)}^2$.[4]

2      Update $\pi_s^{t+1} \in \underset{p \in \Delta(\mathcal{A})}{\text{argmin}} \, \mathcal{D}_h(p, \nabla h^*(\eta_t f_s^{\theta^{t+1}})) = \text{Proj}_{\Delta(\mathcal{A})}^h(\nabla h^*(\eta_t f_s^{\theta^{t+1}})), \ \forall s \in \mathcal{S}$.

---

**Definition 3.1.** Given a parameterized function class $\mathcal{F}^\Theta = \{f^\theta : \mathcal{S} \times \mathcal{A} \to \mathbb{R}, \theta \in \Theta\}$, a mirror map $h : \mathcal{Y} \to \mathbb{R}$, where $\mathcal{Y} \subseteq \mathbb{R}^{|\mathcal{A}|}$ is a convex set with $\Delta(\mathcal{A}) \subseteq \mathcal{Y}$, and $\eta > 0$, the *Bregman projected policy* class associated with $\mathcal{F}^\Theta$ and $h$ consists of all the policies of the form:

$$\left\{ \pi^\theta : \pi_s^\theta = \text{Proj}_{\Delta(\mathcal{A})}^h(\nabla h^*(\eta f_s^\theta)), \ s \in \mathcal{S}; \ \theta \in \Theta \right\},$$

where for all $s \in \mathcal{S}$, $\pi_s^\theta, f_s^\theta \in \mathbb{R}^{|\mathcal{A}|}$ denote vectors $[\pi^\theta(a \mid s)]_{a \in \mathcal{A}}$ and $[f^\theta(s, a)]_{a \in \mathcal{A}}$, respectively.

In this definition, the policy is induced by a mirror map $h$ and a parameterized function $f^\theta$, and is obtained by mapping $f^\theta$ to $\mathcal{Y}$ with the operator $\nabla h^*(\cdot)$, which may not result in a well-defined probability distribution, and is thus projected on the probability simplex $\Delta(\mathcal{A})$. Note that the choice of $h$ is key in deriving convenient expressions for $\pi^\theta$. The Bregman projected policy class contains large families of policy classes. Below is an example of $h$ that recovers a widely used policy class [7, Example 9.10].

**Example 3.2** (Negative entropy). If $\mathcal{Y} = \mathbb{R}_+^{|\mathcal{A}|}$ and $h$ is the negative entropy mirror map, i.e., $h(\pi(\cdot|s)) = \sum_{a \in \mathcal{A}} \pi(a|s) \log(\pi(a|s))$, then the associated Bregman projected policy class becomes

$$\left\{ \pi^\theta : \pi_s^\theta = \frac{\exp(\eta f_s^\theta)}{\|\exp(\eta f_s^\theta)\|_1}, \ s \in \mathcal{S}; \ \theta \in \Theta \right\}, \tag{8}$$

where the exponential and the fraction are element-wise and $\|\cdot\|_1$ is the $\ell_1$ norm. In particular, when $f^\theta(s, a) = \theta_{s,a}$, the policy class (8) becomes the tabular softmax policy class; when $f^\theta$ is a linear function, (8) becomes the log-linear policy class; and when $f^\theta$ is a neural network, (8) becomes the neural policy class defined by Agarwal et al. [1]. We refer to Appendix C.1 for its derivation.

We now construct an MD-based algorithm to optimize $V^{\pi^\theta}$ over the Bregman projected policy class associated with a mirror map $h$ and a parameterization class $\mathcal{F}^\Theta$ by adapting Section 2.1 to our setting. We define the following shorthand: at each time $t$, let $\pi^t := \pi^{\theta^t}$, $f^t := f^{\theta^t}$, $V^t := V^{\pi^t}$, $Q^t := Q^{\pi^t}$, and $d_\mu^t := d_\mu^{\pi^t}$. Further, for any function $y : \mathcal{S} \times \mathcal{A} \to \mathbb{R}$ and distribution $v$ over $\mathcal{S} \times \mathcal{A}$, let $y_s := y(s, \cdot) \in \mathbb{R}^{|\mathcal{A}|}$ and $\|y\|_{L_2(v)}^2 = \mathbb{E}_{(s,a) \sim v}[(y(s, a))^2]$. Ideally, we would like to execute the exact MD algorithm: for all $t \geq 0$ and for all $s \in \mathcal{S}$,

$$f_s^{t+1} = \nabla h(\pi_s^t) + \eta_t (1 - \gamma) \nabla_s V^t(\mu)/d_\mu^t(s) \overset{(3)}{=} \nabla h(\pi_s^t) + \eta_t Q_s^t, ^3 \tag{9}$$

$$\pi_s^{t+1} = \text{Proj}_{\Delta(\mathcal{A})}^h(\nabla h^*(\eta_t f_s^{t+1})). \tag{10}$$

Here, (10) reflects our Bregman projected policy class 3.1. However, we usually cannot perform the update (9) exactly. In general, if $f^\theta$ belongs to a parameterized class $\mathcal{F}^\Theta$, there may not be any $\theta^{t+1} \in \Theta$ such that (9) is satisfied for all $s \in \mathcal{S}$.

To remedy this issue, we propose Approximate Mirror Policy Optimization (AMPO), described in Algorithm 1. At each iteration, AMPO consists in minimizing a surrogate loss and projecting the result onto the simplex to obtain the updated policy. In particular, the surrogate loss in Line 1 of Algorithm 1 is a standard regression problem where we try to approximate $Q^t + \eta_t^{-1} \nabla h(\pi^t)$ with $f^{t+1}$, and has been studied extensively when $f^\theta$ is a neural network [3, 35]. We can then readily use (10) to update $\pi^{t+1}$ within the Bregman projected policy class defined in 3.1, which gives Line 2 of Algorithm 1.

To better illustrate the novelty of our framework, we provide below two remarks on how the two steps of AMPO relate and improve upon previous works.

---

[3] The update is (5) up to a scaling $(1 - \gamma)/d_\mu^t(s)$ of $\eta_t$.

[4] With a slight abuse of notation, denote $\nabla h(\pi^t)(s, a)$ as $[\nabla h(\pi_s^t)]_a$.

*Remark* 3.3. Line 1 associates AMPO with the *compatible function approximation* framework developed by [87, 42, 1], as both frameworks define the updated parameters $\theta^{t+1}$ as the solution to a regression problem aimed at approximating the current $Q$-function $Q^t$. A crucial difference is that, Agarwal et al. [1] approximate $Q^t$ linearly with respect to $\nabla_\theta \log \pi^t$ (see (61)), while in Line 1 we approximate $Q^t$ and the gradient of the mirror map of the previous policy with any function $f^\theta$. This generality allows our algorithm to achieve approximation guarantees for a wider range of assumptions on the structure of $Q^t$. Furthermore, the regression problem proposed by Agarwal et al. [1] depends on the distribution $d^t_\mu$, while ours has no such constraint and allows off-policy updates involving an arbitrary distribution $v^t$. See Appendix E for more details.

*Remark* 3.4. Line 2 associates AMPO to previous approximations of PMD [89, 88]. For instance, Vaswani et al. [89] aim to maximize an expression equivalent to

$$\pi^{t+1} \in \text{argmax}_{\pi^\theta \in \Pi(\Theta)} \, \mathbb{E}_{s \sim d^t_\mu}[\eta_t \langle Q^t_s, \pi^\theta_s \rangle - \mathcal{D}_h(\pi^\theta_s, \pi^t_s)], \tag{11}$$

where $\Pi(\Theta)$ is a given parameterized policy class, while the Bregman projection step of AMPO can be rewritten as

$$\pi^{t+1}_s \in \text{argmax}_{p \in \Delta(\mathcal{A})} \langle \eta_t f^{t+1}_s - \nabla h(\pi^t_s), p \rangle - \mathcal{D}_h(p, \pi^t_s), \quad \forall s \in \mathcal{S}. \tag{12}$$

We formulate this result as Lemma F.1 in Appendix F with a proof. When the policy class $\Pi(\Theta)$ is the entire policy space $\Delta(\mathcal{A})^\mathcal{S}$, (11) is equivalent to the two-step procedure (9)-(10) thanks to the PGT (3). A derivation of this observation is given in Appendix B for completeness. The issue with the update in (11), which is overcome by (12), is that $\Pi(\Theta)$ in (11) is often a non-convex set, thus the three-point-descent lemma [19] cannot be applied. The policy update in (12) circumvents this problem by defining the policy implicitly through the Bregman projection, which is a convex problem and thus allows the application of the three-point-descent lemma [19]. We refer to Appendix F for the conditions of the three-point-descent lemma in details.

## 3.1 $\omega$-potential mirror maps

In this section, we provide a class of mirror maps that allows to compute the Bregman projection in Line 2 with $\widetilde{\mathcal{O}}(|\mathcal{A}|)$ operations and simplifies the minimization problem in Line 1.

**Definition 3.5** ($\omega$-potential mirror map [50]). For $u \in (-\infty, +\infty]$, $\omega \leq 0$, let an $\omega$-*potential* be an increasing $C^1$-diffeomorphism $\phi : (-\infty, u) \to (\omega, +\infty)$ such that

$$\lim_{x \to -\infty} \phi(x) = \omega, \qquad \lim_{x \to u} \phi(x) = +\infty, \qquad \int_0^1 \phi^{-1}(x) dx \leq \infty.$$

For any $\omega$-potential $\phi$, the associated mirror map $h_\phi$, called $\omega$-*potential mirror map*, is defined as

$$h_\phi(\pi_s) = \sum_{a \in \mathcal{A}} \int_1^{\pi(a|s)} \phi^{-1}(x) dx.$$

Thanks to Krichene et al. [50, Proposition 2] (see Proposition C.1 as well), the policy $\pi^{t+1}$ in Line 2 induced by the $\omega$-potential mirror map can be obtained with $\widetilde{\mathcal{O}}(|\mathcal{A}|)$ computations and can be written as

$$\pi^{t+1}(a \mid s) = \sigma(\phi(\eta_t f^{t+1}(s, a) + \lambda^{t+1}_s)), \quad \forall s \in \mathcal{S}, a \in \mathcal{A}, \tag{13}$$

where $\lambda^{t+1}_s \in \mathbb{R}$ is a normalization factor to ensure $\sum_{a \in \mathcal{A}} \pi^{t+1}(a \mid s) = 1$ for all $s \in \mathcal{S}$, and $\sigma(z) = \max(z, 0)$ for $z \in \mathbb{R}$. We call this policy class the $\omega$-*potential policy* class. By using (13) and the definition of the $\omega$-potential mirror map $h_\phi$, the minimization problem in Line 1 is simplified to be

$$\theta^{t+1} \in \text{argmin}_{\theta \in \Theta} \left\| f^\theta - Q^t - \eta_t^{-1} \max(\eta_{t-1} f^t, \phi^{-1}(0) - \lambda^t_s) \right\|^2_{L_2(v^t)}, \tag{14}$$

where $\max(\cdot, \cdot)$ is applied element-wisely. The $\omega$-potential policy class allows AMPO to generate a wide range of applications by simply choosing an $\omega$-potential $\phi$. In fact, it recovers existing approaches to policy optimization, as we show in the next two examples.

**Example 3.6** (Squared $\ell_2$-norm). If $\mathcal{Y} = \mathbb{R}^{|\mathcal{A}|}$ and $\phi$ is the identity function, then $h_\phi$ is the squared $\ell_2$-norm, that is $h_\phi(\pi_s) = \|\pi_s\|^2_2 / 2$, and $\nabla h_\phi$ is the identity function. So, Line 1 in Algorithm 1 becomes

$$\theta^{t+1} \in \text{argmin}_{\theta \in \Theta} \left\| f^\theta - Q^t - \eta_t^{-1} \pi^t \right\|^2_{L_2(v^t)}. \tag{15}$$

The $\nabla h_\phi^*$ also becomes the identity function, and the policy update is given for all $s \in \mathcal{S}$ by

$$\pi_s^{t+1} = \text{Proj}_{\Delta(\mathcal{A})}^{h_\phi}(\eta_t f_s^{t+1}) = \underset{\pi_s \in \Delta(\mathcal{A})}{\text{argmin}} \left\| \pi_s - \eta_t f_s^{t+1} \right\|_2^2, \tag{16}$$

which is the Euclidean projection on the probability simplex. In the tabular setting, where $\mathcal{S}$ and $\mathcal{A}$ are finite and $f^\theta(s, a) = \theta_{s,a}$, (15) can be solved exactly with the minimum equal to zero, and Equations (15) and (16) recover the projected Q-descent algorithm [93]. As a by-product, we generalize the projected Q-descent algorithm from the tabular setting to a general parameterization class $\mathcal{F}^\Theta$, which is a novel algorithm in the RL literature.

**Example 3.7** (Negative entropy). If $\mathcal{Y} = \mathbb{R}_+^{|\mathcal{A}|}$ and $\phi(x) = \exp(x - 1)$, then $h_\phi$ is the negative entropy mirror map from Example 3.2 and Line 1 in Algorithm 1 becomes

$$\theta^{t+1} \in \text{argmin}_{\theta \in \Theta} \left\| f^{t+1} - Q^t - \frac{\eta_{t-1}}{\eta_t} f^t \right\|_{L_2(v^t)}^2. \tag{17}$$

Consequently, based on Example 3.2, we have $\pi_s^{t+1} \propto \exp(\eta_t f_s^{t+1})$ for all $s \in \mathcal{S}$. In this example, AMPO recovers tabular NPG [82] when $f^\theta(s, a) = \theta_{s,a}$, and Q-NPG with log-linear policies [98] when $f^\theta$ and $Q^t$ are linear functions for all $t \geq 0$.

We refer to Appendix C.2 for detailed derivations of the examples in this section and an efficient implementation of the Bregman projection step. In addition to the $\ell_2$-norm and the negative entropy, several other mirror maps that have been studied in the optimization literature fall into the class of $\omega$-potential mirror maps, such as the Tsallis entropy [74, 57] and the hyperbolic entropy [34], as well as a generalization of the negative entropy [50]. These examples illustrate how the class of $\omega$-potential mirror maps recovers known methods and can be used to explore new algorithms in policy optimization. We leave the study of the application of these mirror maps in RL as future work, both from an empirical and theoretical point of view, and provide additional discussion and details in Appendix C.2.

## 4 Theoretical analysis

In our upcoming theoretical analysis of AMPO, we rely on the following key lemma.

**Lemma 4.1.** *For any policies $\pi$ and $\bar{\pi}$, for any function $f^\theta \in \mathcal{F}^\Theta$ and for $\eta > 0$, we have*

$$\langle \eta f_s^\theta - \nabla h(\bar{\pi}_s), \pi_s - \tilde{\pi}_s \rangle \leq \mathcal{D}_h(\pi_s, \bar{\pi}_s) - \mathcal{D}_h(\tilde{\pi}_s, \bar{\pi}_s) - \mathcal{D}_h(\pi_s, \tilde{\pi}_s), \quad \forall s \in \mathcal{S},$$

*where $\tilde{\pi}$ is the Bregman projected policy induced by $f^\theta$ and $h$ according to Definition 3.1, that is $\tilde{\pi}_s = \text{argmin}_{p \in \Delta(\mathcal{A})} \mathcal{D}_h(p, \nabla h^*(\eta f_s^\theta))$ for all $s \in \mathcal{S}$.*

The proof of Lemma 4.1 is given in Appendix D.1. Lemma 4.1 describes a relation between any two policies and a policy belonging to the Bregman projected policy class associated with $\mathcal{F}^\Theta$ and $h$. As mentioned in Remark 3.4, Lemma 4.1 is the direct consequence of (12) and can be interpreted as an application of the three-point descent lemma [19], while it cannot be applied to algorithms based on the update in (11) [88, 89] due to the non-convexity of the optimization problem (see also Appendix F). Notice that Lemma 4.1 accommodates naturally with general parameterization also thanks to (12). In contrast, similar results have been obtained and exploited for specific policy and mirror map classes [93, 60, 36, 98], while our result allows any parameterization class $\mathcal{F}^\Theta$ and any choice of mirror map, thus greatly expanding the scope of applications of the lemma. A similar result for general parameterization has been obtained by Lan [55, Proposition 3.5] in the setting of strongly convex mirror maps.

Lemma 4.1 becomes useful when we set $\bar{\pi} = \pi^t$, $f^\theta = f^{t+1}$, $\eta = \eta_t$ and $\pi = \pi^t$ or $\pi = \pi^\star$. In particular, when $\eta_t f_s^{t+1} - \nabla h(\pi_s^t) \approx \eta_t Q_s^\pi$, Lemma 4.1 allows us to obtain telescopic sums and recursive relations, and to handle error terms efficiently, as we show in Appendix D.

### 4.1 Convergence for general policy parameterization

In this section, we consider the parameterization class $\mathcal{F}^\Theta$ and the fixed but arbitrary mirror map $h$. We show that AMPO enjoys quasi-monotonic improvement and sublinear or linear convergence, depending on the step-size schedule. The first step is to control the approximation error of AMPO.

(A1) (*Approximation error*). There exists $\varepsilon_{\mathrm{approx}} \geq 0$ such that, for all times $t \geq 0$,

$$\mathbb{E}\big[\,\big\|f^{t+1} - Q^t - \eta_t^{-1}\nabla h(\pi^t)\big\|_{L_2(v^t)}^2\,\big] \leq \varepsilon_{\mathrm{approx}},$$

where $(v^t)_{t\geq 0}$ is a sequence of distributions over states and actions and the expectation is taken over the randomness of the algorithm that obtains $f^{t+1}$.

Assumption (A1) is common in the conventional compatible function approximation approach[5] [1]. It characterizes the loss incurred by Algorithm 1 in solving the regression problem in Line 1. When the step-size $\eta_t$ is sufficiently large, Assumption (A1) measures how well $f^{t+1}$ approximates the current Q-function $Q^t$. Hence, $\varepsilon_{\mathrm{approx}}$ depends on both the accuracy of the policy evaluation method used to obtain an estimate of $Q^t$ [86, 79, 28] and the error incurred by the function $f^\theta \in \mathcal{F}^\Theta$ that best approximates $Q^t$, that is the representation power of $\mathcal{F}^\Theta$. Later in Section 4.2, we show how to solve the minimization problem in Line 1 when $\mathcal{F}^\Theta$ is a class of shallow neural networks so that Assumption (A1) holds. We highlight that Assumption (A1) is weaker than the conventional assumptions [1, 98], since we do not constrain the minimization problem to be linear in the parameters (see (61)). We refer to Appendix A.2 for a discussion on its technical novelty and Appendix G for a relaxed version of the assumption.

As mentioned in Remark 3.3, the distribution $v^t$ does not depend on the current policy $\pi^t$ for all times $t \geq 0$. Thus, Assumption (A1) allows for off-policy settings and the use of replay buffers [68]. We refer to Appendix A.3 for details. To quantify how the choice of these distributions affects the error terms in the convergence rates, we introduce the following coefficient.

(A2) (*Concentrability coefficient*). There exists $C_v \geq 0$ such that, for all times $t$,

$$\mathbb{E}_{(s,a)\sim v^t}\left[\left(\frac{d_\mu^\pi(s)\pi(a\mid s)}{v^t(s,a)}\right)^2\right] \leq C_v,$$

whenever $(d_\mu^\pi, \pi)$ is either $(d_\mu^\star, \pi^\star)$, $(d_\mu^{t+1}, \pi^{t+1})$, $(d_\mu^\star, \pi^t)$, or $(d_\mu^{t+1}, \pi^t)$.

The concentrability coefficient $C_v$ quantifies how much the distribution $v^t$ overlaps with the distributions $(d_\mu^\star, \pi^\star)$, $(d_\mu^{t+1}, \pi^{t+1})$, $(d_\mu^\star, \pi^t)$ and $(d_\mu^{t+1}, \pi^t)$. It highlights that the distribution $v^t$ should have full support over the environment, in order to avoid large values of $C_v$. Assumption (A2) is weaker than the previous best-known concentrability coefficient [98, Assumption 9], in the sense that we have the full control over $v^t$. We refer to Appendix H for a more detailed discussion. We can now present our first result on the performance of Algorithm 1.

**Proposition 4.2** (Quasi-monotonic updates)**.** *Let* (A1)*,* (A2) *be true. We have, for all* $t \geq 0$,

$$\mathbb{E}\left[V^{t+1}(\mu) - V^t(\mu)\right] \geq \mathbb{E}\left[\mathbb{E}_{s\sim d_\mu^{t+1}}\left[\frac{\mathcal{D}_h(\pi_s^{t+1}, \pi_s^t) + \mathcal{D}_h(\pi_s^t, \pi_s^{t+1})}{\eta_t(1-\gamma)}\right]\right] - \frac{2\sqrt{C_v\varepsilon_{\mathrm{approx}}}}{1-\gamma},$$

*where the expectation is taken over the randomness of AMPO.*

We refer to Appendix D.3 for the proof. Proposition 4.2 ensures that an update of Algorithm 1 cannot lead to a performance degradation, up to an error term. The next assumption concerns the coverage of the state space for the agent at each time $t$.

(A3) (*Distribution mismatch coefficient*). Let $d_\mu^\star := d_\mu^{\pi^\star}$. There exists $\nu_\mu \geq 0$ such that

$$\sup_{s\in\mathcal{S}} \frac{d_\mu^\star(s)}{d_\mu^t(s)} \leq \nu_\mu, \quad \text{for all times } t \geq 0.$$

Since $d_\mu^t(s) \geq (1-\gamma)\mu(s)$ for all $s \in \mathcal{S}$, obtained from the definition of $d_\mu$ in (2), we have that

$$\sup_{s\in\mathcal{S}} \frac{d_\mu^\star(s)}{d_\mu^t(s)} \leq \frac{1}{1-\gamma} \sup_{s\in\mathcal{S}} \frac{d_\mu^\star(s)}{\mu(s)},$$

where assuming boundedness for the term on the right-hand side is standard in the literature on both the PG [e.g., 101, 90] and NPG convergence analysis [e.g., 1, 15, 93]. We refer to Appendix I for details. It is worth mentioning that the quasi-monotonic improvement in Proposition 4.2 holds without (A3).

We define the weighted Bregman divergence between the optimal policy $\pi^\star$ and the initial policy $\pi^0$ as $\mathcal{D}_0^\star = \mathbb{E}_{s\sim d_\mu^\star}[\mathcal{D}_h(\pi_s^\star, \pi_s^0)]$. We then have our main results below.

---

[5]An extended discussion of this approach is provided in Appendix G.

**Theorem 4.3.** *Let* (A1)*,* (A2) *and* (A3) *be true. If the step-size schedule is non-decreasing, i.e.,* $\eta_t \leq \eta_{t+1}$ *for all* $t \geq 0$*, the iterates of Algorithm 1 satisfy: for every* $T \geq 0$,

$$V^\star(\mu) - \frac{1}{T}\sum_{t<T} \mathbb{E}\left[V^t(\mu)\right] \leq \frac{1}{T}\left(\frac{\mathcal{D}_0^\star}{(1-\gamma)\eta_0} + \frac{\nu_\mu}{1-\gamma}\right) + \frac{2(1+\nu_\mu)\sqrt{C_v \varepsilon_{\text{approx}}}}{1-\gamma}.$$

*Furthermore, if the step-size schedule is geometrically increasing, i.e., satisfies*

$$\eta_{t+1} \geq \frac{\nu_\mu}{\nu_\mu - 1}\eta_t \qquad \forall t \geq 0, \tag{18}$$

*we have: for every* $T \geq 0$,

$$V^\star(\mu) - \mathbb{E}\left[V^T(\mu)\right] \leq \frac{1}{1-\gamma}\left(1 - \frac{1}{\nu_\mu}\right)^T \left(1 + \frac{\mathcal{D}_0^\star}{\eta_0(\nu_\mu - 1)}\right) + \frac{2(1+\nu_\mu)\sqrt{C_v \varepsilon_{\text{approx}}}}{1-\gamma}.$$

Theorem 4.3 is, to the best of our knowledge, the first result that establishes linear convergence for a PG-based method involving general policy parameterization. For the same setting, it also matches the previous best known $O(1/T)$ convergence [55], without requiring regularization. Lastly, Theorem 4.3 provides a convergence rate for a PMD-based algorithm that allows for arbitrary mirror maps and policy parameterization without requiring the assumption on the approximation error to hold in $\ell_\infty$-norm, in contrast to Lan [55]. We give here a brief discussion of Theorem 4.3 w.r.t. previous results and refer to Tables 1 and 2 in Appendix A.2 for a detailed comparison.

In terms of iteration complexity, Theorem 4.3 recovers the best-known convergence rates in the tabular setting [93], for both non-decreasing and exponentially increasing step-size schedules. While considering a more general setting, Theorem 4.3 matches or improves upon the convergence rate of previous work on policy gradient methods for non-tabular policy parameterizations that consider constant step-size schedules [60, 82, 61, 90, 1, 89, 16, 55], and matches the convergence speed of previous work that employ NPG, log-linear policies, and geometrically increasing step-size schedules [2, 98].

In terms of generality, the results in Theorem 4.3 hold without the need to implement regularization [17, 100, 15, 16, 54], to impose bounded updates or smoothness of the policy [1, 61], to restrict the analysis to the case where the mirror map $h$ is the negative entropy [60, 36], or to make $\ell_\infty$-norm assumptions on the approximation error [55]. We improve upon the latest results for PMD with general policy parameterization by Vaswani et al. [89], which only allow bounded step-sizes, where the bound can be particularly small, e.g., $(1-\gamma)^3/(2\gamma|\mathcal{A}|)$, and can slow down the learning process.

When $\mathcal{S}$ is a finite state space, a sufficient condition for $\nu_\mu$ in (A3) to be bounded is requiring $\mu$ to have full support on $\mathcal{S}$. If $\mu$ does not have full support, one can still obtain linear convergence for $V^\star(\mu') - V^T(\mu')$, for an arbitrary state distribution $\mu'$ with full support, and relate this quantity to $V^\star(\mu) - V^T(\mu)$. We refer to Appendix I for a detailed discussion on the distribution mismatch coefficient.

**Intuition.** An interpretation of our theory can be provided by connecting AMPO to the Policy Iteration algorithm (PI), which also enjoys linear convergence. To see this, first recall (12)

$$\pi_s^{t+1} \in \text{argmin}_{p\in\Delta(\mathcal{A})} \langle -f_s^{t+1} + \eta_t^{-1}\nabla h(\pi_s^t), p\rangle + \eta_t^{-1}\mathcal{D}_h(p, \pi_s^t), \quad \forall s \in \mathcal{S}.$$

Secondly, solving Line 1 of Algorithm 1 leads to $f_s^{t+1} - \eta_t^{-1}\nabla h(\pi_s^t) \approx Q_s^t$. When the step-size $\eta_t \to \infty$, that is $\eta_t^{-1} \to 0$, the above viewpoint of the AMPO policy update becomes

$$\pi_s^{t+1} \in \text{argmin}_{p\in\Delta(\mathcal{A})}\langle -Q_s^t, p\rangle \iff \pi_s^{t+1} \in \text{argmax}_{p\in\Delta(\mathcal{A})}\langle Q_s^t, p\rangle, \quad \forall s \in \mathcal{S},$$

which is the PI algorithm. Here we ignore the Bregman divergence term $\mathcal{D}_h(\pi, \pi_s^t)$, as it is multiplied by $1/\eta_t$, which goes to 0. So AMPO behaves more and more like PI with a large enough step-size and thus is able to converge linearly like PI.

**Proof idea.** We provide a sketch of the proof here; the full proof is given in Appendix D. In a nutshell, the convergence rates of AMPO are obtained by building on Lemma 4.1 and leveraging modern PMD proof techniques [93]. Following the conventional compatible function approximation approach [1], the idea is to write the global optimum convergence results in an additive form, that is

$$\text{sub-optimality gap} \leq \text{optimization error} + \text{approximation error}.$$

The separation between the two errors is allowed by Lemma 4.1, while the optimization error is bounded through the PMD proof techniques from Xiao [93] and the approximation error is characterized by Assumption (A1). Overall, the proof consists of three main steps.

*Step 1.* Using Lemma 4.1 with $\bar{\pi} = \pi^t$, $f^\theta = f^{t+1}$, $\eta = \eta_t$, $\tilde{\pi} = \pi^{t+1}$, and $\pi_s = \pi_s^t$, we obtain

$$\langle \eta_t f_s^{t+1} - \nabla h(\pi_s^t), \pi_s^{t+1} - \pi_s^t \rangle \geq 0,$$

which characterizes the improvement of the updated policy.

*Step 2.* Assumption (A1), Step 1, the performance difference lemma (Lemma D.4), and Lemma 4.1 with $\bar{\pi} = \pi^t$, $f^\theta = f^{t+1}$, $\eta = \eta_t$, $\tilde{\pi} = \pi^{t+1}$, and $\pi_s = \pi_s^\star$ permit us to obtain the following.

**Proposition 4.4.** *Let $\Delta_t := V^\star(\mu) - V^t(\mu)$. For all $t \geq 0$, we have*

$$\mathbb{E}\left[\nu_\mu \left(\Delta_{t+1} - \Delta_t\right) + \Delta_t\right] \leq \mathbb{E}\left[\frac{\mathbb{E}_{s \sim d_\mu^\star}\left[\mathcal{D}_h(\pi_s^\star, \pi_s^t)\right]}{(1-\gamma)\eta_t} - \frac{\mathbb{E}_{s \sim d_\mu^\star}\left[\mathcal{D}_h(\pi_s^\star, \pi_s^{t+1})\right]}{(1-\gamma)\eta_t}\right] + (1 + \nu_\mu)\frac{2\sqrt{C_v \varepsilon_{\text{approx}}}}{1-\gamma}.$$

*Step 3.* Proposition 4.4 leads to sublinear convergence using a telescoping sum argument, and to linear convergence by properly defining step-sizes and by rearranging terms into the following contraction,

$$\mathbb{E}\left[\Delta_{t+1} + \frac{\mathbb{E}_{s \sim d_\mu^\star}\left[\mathcal{D}_h(\pi_s^\star, \pi_s^{t+1})\right]}{(1-\gamma)\eta_{t+1}(\nu_\mu - 1)}\right] \leq \left(1 - \frac{1}{\nu_\mu}\right)\mathbb{E}\left[\Delta_t + \frac{\mathbb{E}_{s \sim d_\mu^\star}\left[\mathcal{D}_h(\pi_s^\star, \pi_s^t)\right]}{(1-\gamma)\eta_t(\nu_\mu - 1)}\right] + \left(1 + \frac{1}{\nu_\mu}\right)\frac{2\sqrt{C_v \varepsilon_{\text{approx}}}}{1-\gamma}.$$

## 4.2  Sample complexity for neural network parameterization

Neural networks are widely used in RL due to their empirical success in applications [67, 68, 84]. However, few theoretical guarantees exist for using this parameterization class in policy optimization [60, 90, 16]. Here, we show how we can use our framework and Theorem 4.3 to fill this gap by deriving a sample complexity result for AMPO when using neural network parameterization. We consider the case where the parameterization class $\mathcal{F}^\Theta$ from Definition 3.1 belongs to the family of shallow ReLU networks, which have been shown to be universal approximators [38, 4, 27, 39]. That is, for $(s, a) \in (\mathcal{S} \times \mathcal{A}) \subseteq \mathbb{R}^d$, define $f^\theta(s, a) = c^\top \sigma(W(s, a) + b)$ with $\theta = (c, W, b)$, where $\sigma(y) = \max(y, 0)$ for all $y \in \mathbb{R}$ is the ReLU activation function and is applied element-wisely, $c \in \mathbb{R}^m$, $W \in \mathbb{R}^{m \times d}$ and $b \in \mathbb{R}^m$.

At each iteration $t$ of AMPO, we set $v^t = d_\mu^t$ and solve the regression problem in Line 1 of Algorithm 1 through stochastic gradient descent (SGD). In particular, we initialize entry-wise $W_0$ and $b$ as i.i.d. random Gaussian variables from $\mathcal{N}(0, 1/m)$, and $c$ as i.i.d. random Gaussian variables from $\mathcal{N}(0, \epsilon_A)$ with $\epsilon_A \in (0, 1]$. Assuming access to a simulator for the distribution $v^t$, we run SGD for $K$ steps on the matrix $W$, that is, for $k = 0, \ldots, K - 1$,

$$W_{k+1} = W_k - \alpha\left(f^{(k)}(s, a) - \widehat{Q}^t(s, a) - \eta_t^{-1}\nabla h(\pi_s^t)\right)\nabla_W f^{(k)}(s, a), \tag{19}$$

where $f^{(k)}(s, a) = c^\top \sigma((W_0 + W_k)(s, a) + b)$, $(s, a) \sim v^t$ and $\widehat{Q}^t(s, a)$ is an unbiased estimate of $Q^t(s, a)$ obtained through Algorithm 4. We can then present our result on the sample complexity of AMPO for neural network parameterization, which is based on our convergence Theorem 4.3 and an analysis of neural networks by Allen-Zhu et al. [3, Theorem 1].

**Corollary 4.5.** *In the setting of Theorem 4.3, let the parameterization class $\mathcal{F}^\Theta$ consist of sufficiently wide shallow ReLU neural networks. Using an exponentially increasing step-size and solving the minimization problem in Line 1 with SGD as in (19), the number of samples required by AMPO to find an $\varepsilon$-optimal policy with high probability is $\widetilde{\mathcal{O}}(C_v^2 \nu_\mu^5 / \varepsilon^4 (1 - \gamma)^6)$, where $\varepsilon$ has to be larger than a fixed and non-vanishing error floor.*

We provide a proof of Corollary 4.5 and an explicit expression for the error floor in Appendix J. Note that the sample complexity in Corollary 4.5 might be impacted by an additional poly($\varepsilon^{-1}$) term. We refer to Appendix J for more details and an alternative result (Corollary J.4) which does not include an additional poly($\varepsilon^{-1}$) term, enabling comparison with prior works.

# 5  Numerical experiments

We provide an empirical evaluation of AMPO in order to validate our theoretical findings. We note that the scope of this work is mainly theoretical and that we do not aim at establishing state-of-the-art results in the setting of deep RL. Our implementation is based upon the PPO implementation

from PureJaxRL [63], which obtains the estimates of the $Q$-function through generalized advantage estimation (GAE) [79] and performs the policy update using ADAM optimizer [48] and mini-batches. To implement AMPO, we (i) replaced the PPO loss with the expression to minimize in Equation (14), (ii) replaced the softmax projection with the Bregman projection, (iii) saved the constants $\lambda$ along the sampled trajectories in order to compute Equation (14). The code is available here.

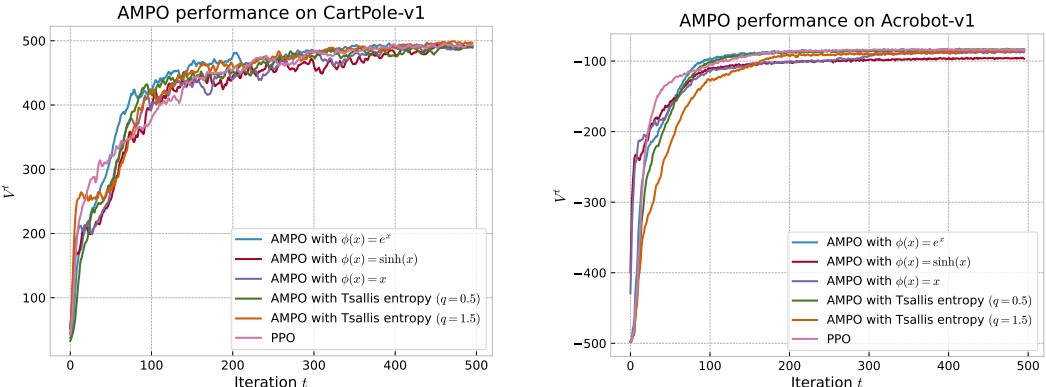

Figure 1: Averaged performance over 50 runs of AMPO in CartPole and Acrobot environments. Note that the maximum values for CartPole and Acrobot are 500 and -80, respectively.

In Figure 1, we show the averaged performance over 100 runs of AMPO in two classic control environments, i.e. CartPole and Acrobot, in the setting of $\omega$-potential mirror maps. In particular, we choose: $\phi(x) = e^x$, which corresponds to the negative entropy (Example 3.7); $\phi(x) = \sinh(x)$, which corresponds to the hyperbolic entropy [34, see also (41)]; $\phi(x) = x$, which corresponds to the Euclidean norm (Example 3.6); and the Tsallis entropy for two values of the entropic index $q$ [74, 57, see also (40)]. We refer to Appendix C.2 for a detailed discussion on these mirror maps. We set the step-size to be constant and of value 1. For a comparison, we also plot the averaged performance over 100 runs of PPO.

The plots in Figure 1 confirm our results on the quasi-monotonicity of the updates of AMPO and on its convergence to the optimal policy. We observe that instances of AMPO with different mirror maps are very competitive as compared to PPO. We also note that, despite the convergence rates in Theorem 4.3 depend on the mirror map only in terms of a $\mathcal{D}_0^\star$ term, different mirror maps may result in different convergence speeds and error floors in practice. In particular, our experiments suggest that the negative entropy mirror map may not be the best choice for AMPO, and that exploring different mirror maps is a promising direction of research.

## 6   Conclusion

We have introduced a novel framework for RL which, given a mirror map and any parameterization class, induces a policy class and an update rule. We have proven that this framework enjoys sublinear and linear convergence for non-decreasing and geometrically increasing step-size schedules, respectively. Future venues of investigation include studying the sample complexity of AMPO in on-policy and off-policy settings other than neural network parameterization, exploiting the properties of specific mirror maps to take advantage of the structure of the MDP and efficiently including representation learning in the algorithm. We refer to Appendix A.3 for a thorough discussion of future work. We believe that the main contribution of AMPO is to provide a general framework with theoretical guarantees that can help the analysis of specific algorithms and MDP structures. AMPO recovers and improves several convergence rate guarantees in the literature, but it is important to keep in consideration how previous works have exploited particular settings, while AMPO tackles the most general case. It will be interesting to see whether these previous works combined with our fast linear convergence result can derive new efficient sample complexity results.

## Acknowledgments and Disclosure of Funding

We thank the anonymous reviewers for their helpful comments. Carlo Alfano was supported by the Engineering and Physical Sciences Research Council and thanks G-Research for partly funding attendance to NeurIPS 2023.

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

# Appendix

## Table of Contents

Here we provide the related work discussion, the deferred proofs from the main paper and some additional noteworthy observations.

## A   Related work

We provide an extended discussion for the context of our work, including a comparison of different PMD frameworks and a comparison of the convergence theories of PMD in the literature. Furthermore, we discuss future work, such as extending our analysis to the dual averaging updates and developing sample complexity analysis of AMPO.

### A.1   Comparisons with other policy mirror descent frameworks

In this section, we give a comparison of AMPO with some of the most popular policy optimization algorithms in the literature. First, recall AMPO's update through (12), that is, for all $s \in \mathcal{S}$,

$$\pi_s^{t+1} \in \underset{\pi_s \in \Delta(\mathcal{A})}{\operatorname{argmax}} \langle \eta_t f_s^{t+1} - \nabla h(\pi_s^t), \pi_s \rangle - \mathcal{D}_h(\pi_s, \pi_s^t), \tag{20}$$

where $\eta_t f_s^{t+1} - \nabla h(\pi_s^t) \approx \eta_t Q_s^t$ following Line 1 of Algorithm 1. The proof of (20) can be found in Lemma F.1 in Appendix F.

**Generalized Policy Iteration (GPI) [86].** The update consists of evaluating the Q-function of the policy and obtaining the new policy by acting greedily with respect to the estimated Q-function. That is, for all $s \in \mathcal{S}$,

$$\pi_s^{t+1} \in \underset{\pi_s \in \Delta(\mathcal{A})}{\operatorname{argmax}} \langle Q_s^t, \pi_s \rangle. \tag{21}$$

AMPO behaves like GPI when we perfectly approximate $f_s^{t+1}$ to the value of $Q_s^t$ (e.g. when we consider the tabular case) and $\eta_t \to +\infty$ (or $\eta_t^{-1} \to 0$) which is the case with the use of geometrically increasing step-size schedule.

**Mirror Descent Modified Policy Iteration (MD-MPI) [33].** Consider the full policy space $\Pi = \Delta(\mathcal{A})^{\mathcal{S}}$. The MD-MPI's update is as follows:

$$\pi_s^{t+1} \in \operatorname*{argmax}_{\pi_s \in \Delta(\mathcal{A})} \langle Q_s^t, \pi_s \rangle - \mathcal{D}_h(\pi_s, \pi_s^t), \quad \forall s \in \mathcal{S}. \tag{22}$$

In this case, the PMD framework of Xiao [93], which is a special case of AMPO, recovers MD-MPI with the fixed step-size $\eta_t = 1$. Consequently, Assumption (A1) holds with $\varepsilon_{\text{approx}} = 0$, and we obtain the sublinear convergence of MD-MPI through Theorem 4.3, which is

$$V^\star(\mu) - \frac{1}{T} \sum_{t<T} \mathbb{E}\left[V^t(\mu)\right] \leq \frac{1}{T}\left(\frac{\mathcal{D}_0^\star}{(1-\gamma)} + \frac{\nu_\mu}{1-\gamma}\right).$$

As explained later in Appendix H that the distribution mismatch coefficient $\nu_\mu$ is upper bounded by $\mathcal{O}(\frac{1}{1-\gamma})$, we obtain an average regret of MD-MPI as $\mathcal{O}(\frac{1}{(1-\gamma)^2 T})$, which matches the convergence results in Geist et al. [33, Corollary 3].

**Trust Region Policy Optimization (TRPO) [78].** The TRPO's update is as follows:

$$\pi^{t+1} \in \operatorname*{argmax}_{\pi \in \Pi} \mathbb{E}_{s \sim d_\mu^t}\left[\langle A_s^t, \pi_s \rangle\right], \tag{23}$$

$$\text{such that } \mathbb{E}_{s \sim d_\mu^t}\left[D_h(\pi_s^t, \pi_s)\right] \leq \delta,$$

where $A_s^t = Q_s^t - V^t$ represents the advantage function, $h$ is the negative entropy and $\delta > 0$. Like GPI, TRPO is equivalent to AMPO when at each time $t$, the admissible policy class is $\Pi^t = \{\pi \in \Delta(\mathcal{A})^{\mathcal{S}} : \mathbb{E}_{s \sim d_\mu^t} D_h(\pi_s^t, \pi_s) \leq \delta\}$, and we perfectly approximate $Q_s^t$ with $\eta_t \to +\infty$.

**Proximal Policy Optimization (PPO) [80].** The PPO's update consists of maximizing a surrogate function depending on the policy gradient with respect to the new policy. Namely,

$$\pi^{t+1} \in \operatorname*{argmax}_{\pi \in \Pi} \mathbb{E}_{s \sim d_\mu^t}\left[L(\pi_s, \pi_s^t)\right], \tag{24}$$

with

$$L(\pi_s, \pi_s^t) = \mathbb{E}_{a \sim \pi^t}[\min\left(r^\pi(s,a)A^t(s,a), \text{clip}(r^\pi(s,a), 1 \pm \epsilon)A^t(s,a)\right)],$$

where $r^\pi(s,a) = \pi(s,a)/\pi^t(s,a)$ is the probability ratio between the current policy $\pi^t$ and the new one, and the function $\text{clip}(r^\pi(s,a), 1 \pm \epsilon)$ clips the probability ratio $r^\pi(s,a)$ to be no more than $1 + \epsilon$ and no less than $1 - \epsilon$. PPO has also a KL variation [80, Section 4], where the objective function $L$ is defined as

$$L(\pi_s, \pi_s^t) = \eta_t \langle A_s^t, \pi_s \rangle - D_h(\pi_s^t, \pi_s),$$

where $h$ is the negative entropy. In an exact setting and when $\Pi = \Delta(\mathcal{A})^{\mathcal{S}}$, the KL variation of PPO still differs from AMPO because it inverts the terms in the Bregman divergence penalty.

**Mirror Descent Policy Optimization (MDPO.) [88].** The MDPO's update is as follows:

$$\pi^{t+1} \in \operatorname*{argmax}_{\pi \in \Pi} \mathbb{E}_{s \sim d_\mu^t}[\langle \eta_t A_s^t, \pi_s \rangle - D_h(\pi_s, \pi_s^t)], \tag{25}$$

where $\Pi$ is a parameterized policy class. While it is equivalent to AMPO in an exact setting and when $\Pi = \Delta(\mathcal{A})^{\mathcal{S}}$, as we show in Appendix B, the difference between the two algorithms lies in the approximation of the exact algorithm.

**Functional Mirror Ascent Policy Gradient (FMA-PG) [89].** The FMA-PG's update is as follows:

$$\pi^{t+1} \in \operatorname*{argmax}_{\pi^\theta : \theta \in \Theta} \mathbb{E}_{s \sim d_\mu^t}[V^t(\mu) + \langle \eta_t \nabla_{\pi_s} V^t(\mu)\big|_{\pi = \pi^t}, \pi_s^\theta - \pi_s^t \rangle - D_h(\pi_s^\theta, \pi_s^t)] \tag{26}$$

$$\in \operatorname*{argmax}_{\pi^\theta : \theta \in \Theta} \mathbb{E}_{s \sim d_\mu^t}[\langle \eta_t Q_s^t, \pi_s^\theta \rangle - D_h(\pi_s^\theta, \pi_s^t)],$$

The second line is obtained by the definition of $V^t$ and the policy gradient theorem (3). The discussion is the same as the previous algorithm.

**Mirror Learning [51].** The on-policy version of the algorithm consists of the following update:

$$\pi^{t+1} = \underset{\pi \in \Pi(\pi^t)}{\operatorname{argmax}} \mathbb{E}_{s \sim d_\mu^t}[\langle Q_s^t, \pi_s \rangle - D(\pi_s, \pi_s^t)], \tag{27}$$

where $\Pi(\pi^t)$ is a policy class that depends on the current policy $\pi^t$ and the drift functional $D$ is defined as a map $D : \Delta(\mathcal{A}) \times \Delta(\mathcal{A}) \to \mathbb{R}$ such that $D(\pi_s, \bar{\pi}_s) \geq 0$ and $\nabla_{\pi_s} D(\pi_s, \bar{\pi}_s)\big|_{\pi_s = \bar{\pi}_s} = 0$. The drift functional $D$ recovers the Bregman divergence as a particular case, in which case Mirror Learning is equivalent to AMPO in an exact setting and when $\Pi = \Delta(\mathcal{A})^{\mathcal{S}}$. Again, the main difference between the two algorithms lies in the approximation of the exact algorithm.

## A.2 Discussion on related work

*Our Contributions.* Our work provides a framework for policy optimization – AMPO. For AMPO, we establish in Theorem 4.3 both $\mathcal{O}(1/T)$ convergence guarantee by using a non-decreasing step-size and linear convergence guarantee by using a geometrically increasing step-size. Our contributions to the prior literature on sublinear and linear convergence of policy optimization methods can be summarized as follows.

- The generality of our framework allows Theorem 4.3 to unify previous results in the literature and generate new theoretically sound algorithms under one guise. Both the sublinear and the linear convergence analysis of natural policy gradient (NPG) with softmax tabular policies [93] or with log-linear policies [2, 98] are special cases of our general analysis. As mentioned in Appendix A.1, MD-MPI [33] in the tabular setting is also a special case of AMPO. Thus, Theorem 4.3 recovers the best-known convergence rates in both the tabular setting [33, 93] and the non-tabular setting [16, 2, 98]. AMPO also generates new algorithms by selecting mirror maps, such as the $\epsilon$-negative entropy mirror map in Appendix C.2 associated with Algorithm 2, and generalizes the projected Q-descent algorithm [93] from the tabular setting to a general parameterization class $\mathcal{F}^\Theta$.

- As discussed in Section 4.1, the results of Theorem 4.3 hold for a general setting with fewer restrictions than in previous work. The generality of the assumptions of Theorem 4.3 allows the application of our theory to specific settings, where existing sample complexity analyses could be improved thanks to the linear convergence of AMPO. For instance, since Theorem 4.3 holds for any structural MDP, AMPO could be applied directly to the linear MDP setting to derive a sample complexity analysis of AMPO which could improve that of Zanette et al. [99] and Hu et al. [36]. As we discuss in Appendix A.3, this is a promising direction for future work.

- From a technical point of view, our main contributions are: Definition 3.1 introduces a novel way of incorporating general parameterization into the policy; the update in Line 1 of Algorithm 1 simplifies the policy optimization step into a regression problem; and Lemma 4.1 establishes a key result for policies belonging to the class in Definition 3.1. Together, these innovations have allowed us to establish new state-of-the-art results in Theorem 4.3 by leveraging the modern PMD proof techniques of Xiao [93].

In particular, our technical novelty with respect to Xiao [93], Alfano and Rebeschini [2], and Yuan et al. [98] can be summarized as follows.

- In terms of algorithm design, AMPO is an innovation. The PMD algorithm proposed by Xiao [93] is strictly limited to the tabular setting and, although it is well defined for any mirror map, it cannot include general parameterization. Alfano and Rebeschini [2] and Yuan et al. [98] propose a first generalization of the PMD algorithm in the function approximation regime thanks to the linear compatible function approximation framework [1], but are limited to considering the log-linear policy parameterization and the entropy mirror map. On the contrary, AMPO solves the problem of incorporating general parameterizations in the policy thanks to Definition 3.1 and the extension of the compatible function approximation framework from linear to nonlinear, which corresponds to the parameter update in Line 1 of Algorithm 1. This innovation is key to the generality of the algorithm, as it allows AMPO to employ any mirror map and any parameterization class. Moreover, AMPO is computationally efficient for a large class of mirror maps (see Appendix C.2 and Algorithms 2 and 3). Our design is readily applied to deep RL, where the policy is usually parameterized by a neural network whose

last layer is a softmax transformation. Our policy definition can be implemented in this setting by replacing the softmax layer with a Bregman projection, as shown in Example 3.7.

- Regarding the assumptions necessary for convergence guarantees, we have weaker assumptions. Xiao [93] requires an $\ell_\infty$-norm on the approximation error of $Q^t$, i.e., $\|\widehat{Q}^t - Q^t\|_\infty \leq \varepsilon_{\text{approx}}$, for all $t \leq T$. Alfano and Rebeschini [2] and Yuan et al. [98] require an $L_2$-norm bound on the error of the linear approximation of $Q^t$, i.e., $\|w^\top \phi - Q^t\|^2_{L_2(v^t)} \leq \varepsilon_{\text{approx}}$ for some feature mapping $\phi : \mathcal{S} \times \mathcal{A} \to \mathbb{R}^d$ and vector $w \in \mathbb{R}^d$, for all $t \leq T$. Our approximation error $\varepsilon_{\text{approx}}$ in Assumption (A1) is an improvement since it does not require the bound to hold in $\ell_\infty$-norm, and allows any regression model instead of linear function approximation, especially neural networks, which greatly increases the representation power of $\mathcal{F}^\Theta$ and expands the range of applications. We further relax Assumption (A1) in Appendix G and show that the approximation error bound can be larger for earlier iterations. In addition, we improve the concentrability coefficients of Yuan et al. [98] by defining the expectation under an arbitrary state-action distribution $v^t$ instead of the state-action visitation distribution with a fixed initial state-action pair (see Yuan et al. [98, Equation (4)]).

- As for the analysis of the algorithm, while we borrow tools from Xiao [93], Alfano and Rebeschini [2], and Yuan et al. [98], our results are not simple extensions. In fact, without our work, it is not clear from Xiao [93], Alfano and Rebeschini [2], and Yuan et al. [98] whether PMD could have theoretical guarantees in a setting with general parameterization and an arbitrary mirror map. The two main problems on this front are the non-convexity of the policy class, which prevents the use of the three-point descent lemma by Chen and Teboulle [19, Lemma 3.2] (or by Xiao [93, Lemma 6]), and the fact that the three-point identity used by Alfano and Rebeschini [2, Equation 4] holds only for the negative entropy mirror map. Our Lemma 4.1 successfully addresses general policy parameterization and arbitrary mirror maps thanks to the design of AMPO. Additionally, we provide a sample complexity analysis of AMPO when employing shallow neural networks that improves upon previous state-of-the-art results in this setting. We further improve this sample complexity analysis in Appendix J, where we consider an approximation error assumption that is weaker than Assumption (A1) (see Appendix G).

We also include a comparison wih Lan [55]. Our diffences can be outlined in two points.

- Lan [55] propose a PMD algorithm (Algorithm 2 in their paper) that can accommodate general parameterization and arbitrary mirror maps. As AMPO, it involves a two-step procedure where the first step is to find an approximation of $Q^t - \eta_t^{-1}\nabla h(\pi^t)$ and the second step is to find the policy through a Bregman projection. However, it is unclear how to implement their algorithm in practice, as they do not propose a specific method to perform either step. We provide an explicit implementation of AMPO and identify a class of mirror maps that is computationally efficient for AMPO (see Appendix C.2 and Algorithms 2 and 3).

- In terms of theoretical analysis, they assume for their results that the approximation error is bounded in $\ell_\infty$-norm over the action space. Let

$$\varepsilon_{\text{det}} = \mathbb{E}_{s \sim v^\star} \left[ \left\| \mathbb{E}[f_s^{t+1}] - Q_s^t - \eta_t^{-1}\nabla h(\pi_s^t) \right\|_\infty \right],$$

$$\varepsilon_{sto} = \mathbb{E}_{s \sim v^\star} \left[ \left\| \mathbb{E}[f_s^{t+1}] - f_s^{t+1} \right\|^2_\infty \right],$$

where the expectation $\mathbb{E}[f_s^{t+1}]$ is taken w.r.t. the stochasticity of the algorithm employed to obtain $f^{t+1}$. Lan [55] assume that both $\varepsilon_{\text{det}}$ and $\varepsilon_{sto}$ are bounded for all iterations $t$. In contrast, our assumptions are weaker as they are required to hold for the $L_2(v)$-norm we define in Section 3. Additionally, Lan [55] establishes a $\mathcal{O}(1/\sqrt{T})$ convergence rate for their algorithm without regularization and a $\mathcal{O}(1/T)$ convergence rate in the regularized case, in both cases using bounded step-sizes. We improve upon these results by obtaining a $\mathcal{O}(1/T)$ convergence rate without regularization and a linear convergence rate.

*Related literature.* Recently, the impressive empirical success of policy gradient (PG)-based methods has catalyzed the development of theoretically sound algorithms for policy optimization. In particular, there has been a lot of attention around algorithms inspired by mirror descent (MD) [70, 8] and, more specifically, by natural gradient descent [5]. These two approaches led to policy mirror descent (PMD) methods [82, 54] and natural policy gradient (NPG) methods [42], which, as first shown by

Neu et al. [72], is a special case of PMD. For instance, PMD and NPG are the building blocks of the state-of-the-art policy optimization algorithms, TRPO [78] and PPO [80]. Leveraging various techniques from the MD literature, it has been established that PMD, NPG, and their variants converge to the global optimum in different settings. We refer to global optimum convergence as an analysis that guarantees that $V^\star(\mu) - \mathbb{E}\left[V^T(\mu)\right] \leq \epsilon$ after $T$ iterations with $\epsilon > 0$. As an important variant of NPG, we will also discuss the literature of the convergence analysis of natural actor-critic (NAC) [75, 12]. The comparison of AMPO with different methods will proceed from the tabular case to different function approximation regimes.

**Sublinear convergence analyses of PMD, NPG and NAC.** For softmax tabular policies, Shani et al. [82] establish a $\mathcal{O}(1/\sqrt{T})$ convergence rate for unregularized NPG and $\mathcal{O}(1/T)$ for regularized NPG. Agarwal et al. [1], Khodadadian et al. [45] and Xiao [93] improve the convergence rate for unregularized NPG and NAC to $\mathcal{O}(1/T)$ and Xiao [93] extends the same convergence rate to projected Q-descent. The same convergence rate is established by MD-MPI [33] through the PMD framework.

In the function approximation regime, Zanette et al. [99] and Hu et al. [36] achieve $\mathcal{O}(1/\sqrt{T})$ convergence rate by developing variants of PMD methods for the linear MDP [40] setting. The same $\mathcal{O}(1/\sqrt{T})$ convergence rate is obtained by Agarwal et al. [1] for both log-linear and smooth policies, while Yuan et al. [98] improve the convergence rate to $\mathcal{O}(1/T)$ for log-linear policies. For smooth policies, the convergence rate is later improved to $\mathcal{O}(1/T)$ either by adding an extra Fisher-non-degeneracy condition on the policies [61] or by analyzing NAC under Markovian sampling [94]. Yang et al. [95] and Huang et al. [37] consider Lipschitz and smooth policies [97], obtain $\mathcal{O}(1/\sqrt{T})$ convergence rates for PMD-type methods and faster $\mathcal{O}(1/T)$ convergence rates by applying the variance reduction techniques SARAH [73] and STORM [22], respectively. As for neural policy parameterization, Liu et al. [60] establish a $\mathcal{O}(1/\sqrt{T})$ convergence rate for two-layer neural PPO. The same $\mathcal{O}(1/\sqrt{T})$ convergence rate is established by Wang et al. [90] for two-layer neural NAC, which is later improved to $\mathcal{O}(1/T)$ by Cayci et al. [16], using entropy regularization.

We highlight that all of the above sublinear convergence analyses, for both softmax tabular policies and the function approximation regime, are obtained either by using a decaying step-size or a constant step-size. Under these step-size schemes, our AMPO's $\mathcal{O}(1/T)$ sublinear convergence rate is the state of the art: it recovers the best-known convergence rates in the tabular setting [33, 93] without regularization; it improves the $\mathcal{O}(1/\sqrt{T})$ convergence rate of Zanette et al. [99] and Hu et al. [36] to $\mathcal{O}(1/T)$ for the linear MDP setting; it recovers the best-known convergence rates for the log-linear policies [98]; it matches the $\mathcal{O}(1/T)$ sublinear convergence rate for smooth and Fisher-non-degenerate policies [61] and the same convergence rate of Yang et al. [95] and Huang et al. [37] for Lipschitz and smooth policies without introducing variance reduction techniques; it matches the previous best-known convergence result in the neural network settings [16] without regularization; lastly, it goes beyond all these results by allowing general parameterization. We refer to Table 1 for an overview of recent sublinear convergence analyses of NPG/PMD.

**Linear convergence analysis of PMD, NPG, NAC and other PG methods.** In the softmax tabular policy settings, the linear convergence guarantees of NPG and PMD are achieved by either adding regularization [17, 100, 54, 58] or by varying the step-sizes [11, 46, 47, 93].

In the function approximation regime, the linear convergence guarantees are achieved for NPG with log-linear policies, either by adding entropy regularization [15] or by choosing geometrically increasing step-sizes [2, 98]. It can also be achieved for NAC with log-linear policy by using adaptive increasing step-sizes [20].

Again, our AMPO's linear convergence rate is the state of the art: not only it recovers the best-known convergence rates in both the tabular setting [93] and the log-linear policies [2, 98] without regularization [17, 100, 54, 58], nor adaptive step-sizes [11, 46, 47, 20], but also it achieves the new state-of-the-art linear convergence rate for PG-based methods with general parameterization, including the neural network parameterizations. We refer to Table 2 for an overview of recent linear convergence analyses of NPG/PMD.

Alternatively, by exploiting a Polyak-Lojasiewicz (PL) condition [76, 62], fast linear convergence results can be achieved for PG methods under different settings, such as linear quadratic control problems [31] and softmax tabular policies with entropy regularization [65, 97]. The PL condition

is extensively studied by Bhandari and Russo [10] to identify more general MDP settings. Like the cases of NPG and PMD, linear convergence of PG can also be obtained for the softmax tabular policy without regularization by choosing adaptive step sizes through exact line search [11] or by exploiting non-uniform smoothness [66]. When the PL condition is relaxed to other weaker conditions, PG methods combined with variance reduction methods such as SARAH [73] and PAGE [59] can also achieve linear convergence. This is shown by Fatkhullin et al. [29, 30] when the PL condition is replaced by the weak PL condition [97], which is satisfied by Fisher-non-degenerate policies [24]. It is also shown by Zhang et al. [102], where the MDP satisfies some hidden convexity property that contains a similar property to the weak PL condition studied by Zhang et al. [101]. Lastly, linear convergence is established for the cubic-regularized Newton method [71], a second-order method, applied to Fisher-non-degenerate policies combined with variance reduction [64].

Outside of the literature focusing on finite time convergence guarantees, Vaswani et al. [89] and Kuba et al. [51] provide a theoretical analysis for variations of PMD and show monotonic improvements for their frameworks. Additionally, Kuba et al. [51] give an infinite time convergence guarantee for their framework.

### A.3 Future work

Our work opens several interesting research directions in both algorithmic and theoretical aspects.

From an algorithmic point of view, the updates in Lines 1 and 2 of AMPO are not explicit. This might be an issue in practice, especially for large scale RL problems. It would be interesting to design efficient regression solver for minimizing the approximation error in Line 1 of Algorithm 1. For instance, by using the dual averaging algorithm [7, Chapter 4], it could be possible to replace the term $\nabla h(\pi_s^t)$ with $f_s^t$ for all $s \in \mathcal{S}$, to make the computation of the algorithm more efficient. That is, it could be interesting to consider the following variation of Line 1 in Algortihm 1:

$$\left\| f^{t+1} - Q^t - \frac{\eta_{t-1}}{\eta_t} f^t \right\|_{L_2(v^t)}^2 \leq \varepsilon_{\mathrm{approx}}. \tag{28}$$

Notice that (28) has the same update as (17), however, (28) is not restricted to using the negative entropy mirror map. To efficiently solve the regression problem in Line 1 of Algorithm 1, one may want to apply modern variance reduction techniques [73, 22, 59]. This has been done by Liu et al. [61] for NPG method.

From a theoretical point of view, it would be interesting to derive a sample complexity analysis for AMPO in specific settings, by leveraging its linear convergence. As mentioned for the linear MDP [40] in Appendix A.2, one can apply the linear convergence theory of AMPO to other structural MDPs, e.g., block MDP [25], factored MDP [44, 85], RKHS linear MDP and RKHS linear mixture MDP [26], to build new sample complexity results for these settings, since the assumptions of Theorem 4.3 do not impose any constraint on the MDP. On the other hand, it would be interesting to explore the interaction between the Bregman projected policy class and the expected Lipschitz and smooth policies [97] and the Fish-non-degenerate policies [61] to establish new improved sample complexity results in these settings, again thanks to the linear convergence theory of AMPO.

Additionally, it would be interesting to study the application of AMPO to the offline setting. In the main text, we have discussed how to extend Algorithm 1 and Theorem 4.3 to the offline setting, where $v^t$ can be set as the state-action distribution induced by an arbitrary behavior policy that generates the data. However, we believe that this direction requires further investigation. One of the major challenges of offline RL is dealing with the distribution shifts that stem from the mismatch between the trained policy $\pi^t$ and the behaviour policy. Several methods have been introduced to deal with this issue, such as constraining the current policy to be close to the behavior policy [56]. We leave introducing offline RL techniques in AMPO as future work.

Another direction for future work is extending the policy update of AMPO to mirror descent algorithm based on value iteration and Bellman operators, such as MD-MPI [33], in order to extend existing results to the general parameterization setting. Other interesting settings that have been addressed using the PMD framework are mean-field games [96] and constrained MDPs [23]. We hope to build on the existing literature for these settings and see whether our results can bring any improvements.

Finally, this work theoretically indicates that, perhaps the most important future work of PMD-type algorithms is to design efficient policy evaluation algorithms to make the estimation of the $Q$-function

Table 1: Overview of sublinear convergence results for NPG and PMD methods with constant step-size in different settings. The dark blue cells contain our new results. The light blue cells contain previously known results that we recover as special cases of our analysis. The pink cells contain previously known results that we improve upon by providing a faster convergence rate. The white cells contain existing results that have already been improved by other literature or that we could not recover under our general analysis.

| Algorithm | Rate | Comparisons to our works |
|---|---|---|
| **Setting**: Softmax tabular policies | | |
| Adaptive TRPO [82] | $\mathcal{O}(1/\sqrt{T})$ | They employ regularization |
| Tabular off-policy NAC [45] | $\mathcal{O}(1/T)$ | We have a weaker approximation error Assumption (A1) with $L_2$ instead of $\ell_\infty$ norm |
| Tabular NPG [1] | $\mathcal{O}(1/T)$ | |
| MD-MPI [33] | $\mathcal{O}(1/T)$ | We match their results when $f^\theta(s,a) = \theta_{s,a}$. |
| Tabular NPG/ projected Q-descent [93] | $\mathcal{O}(1/T)$ | We recover their results when $f^\theta(s,a) = \theta_{s,a}$; we have a weaker approximation error Assumption (A1) with $L_2$ instead of $\ell_\infty$ norm. |
| **Setting**: Log-linear policies | | |
| Q-NPG [1] | $\mathcal{O}(1/\sqrt{T})$ | |
| Q-NPG/NPG [98] | $\mathcal{O}(1/T)$ | We recover their results when $f^\theta(s,a)$ is linear to $\theta$. |
| **Setting**: Softmax two-layer neural policies | | |
| Neural PPO [60] | $\mathcal{O}(1/\sqrt{T})$ | |
| Neural NAC [90] | $\mathcal{O}(1/\sqrt{T})$ | |
| Regularized neural NAC [16] | $\mathcal{O}(1/T)$ | We match their results without regularization. |
| **Setting**: Linear MDP | | |
| NPG [99, 36] | $\mathcal{O}(1/\sqrt{T})$ | |
| **Setting**: Smooth policies | | |
| NPG [1] | $\mathcal{O}(1/\sqrt{T})$ | |
| NAC under Markovian sampling [94] | $\mathcal{O}(1/T)$ | |
| NPG with Fisher-non-degenerate policies [61] | $\mathcal{O}(1/T)$ | |
| **Setting**: Lipschitz and Smooth policies | | |
| Variance reduced PMD [95, 37] | $\mathcal{O}(1/T)$ | We match their results without variance reduction. |
| **Setting**: Bregman projected policies with general parameterization and mirror map | | |
| Regularized PMD [55] | $\mathcal{O}(1/T)$ | We match their results without regularization; we have a weaker approximation error Assumption (A1) with $L_2$ instead of $\ell_\infty$ norm. |
| AMPO (Theorem 4.3, this work) | $\mathcal{O}(1/T)$ | |

Table 2: Overview of linear convergence results for NPG and PMD methods in different settings. The darker cells contain our new results. The light cells contain previously known results that we recover as special cases of our analysis, and extend the permitted concentrability coefficients settings. The white cells contain existing results that we could not recover under our general analysis.

| Algorithm | Reg. | C.S. | A.I.S. | N.I.S.* | Error assumption** |
|---|---|---|---|---|---|
| **Setting**: Softmax tabular policies | | | | | |
| NPG [17] | ✓ | ✓ | | | $\ell_\infty$ |
| PMD [100] | ✓ | ✓ | | | $\ell_\infty$ |
| NPG [54] | ✓ | | | ✓ | $\ell_\infty$ |
| NPG [58] | ✓ | | | ✓ | $\ell_\infty$ |
| NPG [11] | | | ✓ | | |
| NPG [46, 47] | | | ✓ | | $\ell_\infty$ |
| NPG / Projected Q-descent [93] | | | | ✓ | $\ell_\infty$ |
| **Setting**: Log-linear policies | | | | | |
| NPG [15] | ✓ | ✓ | | | $L_2$ |
| Off-policy NAC [20] | | | ✓ | | $\ell_\infty$ |
| Q-NPG [2] | | | | ✓ | $L_2$ |
| Q-NPG/NPG [98] | | | | ✓ | $L_2$ |
| **Setting**: Bregman projected policies with general parameterization and mirror map | | | | | |
| AMPO (Theorem 4.3, this work) | | | | ✓ | $L_2$ |

* **Reg.**: regularization; **C.S.**: constant step-size; **A.I.S.**: Adaptive increasing step-size; **N.I.S.**: Non-adaptive increasing step-size.

** **Error assumption.**: $\ell_\infty$ means that the approximation error assumption uses the $\ell_\infty$-norm; and $L_2$ means that the approximation error assumption uses the weaker $L_2$ norm.

as accurate as possible, such as using offline data for training, and to construct adaptive representation learning for $\mathcal{F}^\Theta$ to closely approximate $Q$-function, so that $\epsilon_{\mathrm{approx}}$ is guaranteed to be small. This matches one of the most important research questions for deep Q-learning type algorithms for general policy optimization problems.

# B   Equivalence of (9)-(10) and (11) in the tabular case

To demonstrate the equivalence between the two-step update (9)-(10) and the one-step update (11) for policy mirror descent in the tabular case, it is sufficient to validate the following lemma, which comes from the optimization literature. The proof of this lemma can be found in Bubeck [14, Chapter 4.2]. However, for the sake of completeness, we present the proof here.

**Lemma B.1** (Right after Theorem 4.2 in Bubeck [14]). *Consider the mirror descent update in* (5)-(6) *for the minimization of a function* $V(\cdot)$*, that is,*

$$y^{t+1} = \nabla h(x^t) - \eta_t \nabla V(x)|_{x=x^t}, \tag{29}$$

$$x^{t+1} = Proj_{\mathcal{X}}^h(\nabla h^*(y^{t+1})). \tag{30}$$

*Then the mirror descent update can be rewritten as*

$$x^{t+1} \in \operatorname*{argmin}_{x \in \mathcal{X}} \eta_t \langle x, \nabla V(x)|_{x=x^t} \rangle + \mathcal{D}_h(x, x^t). \tag{31}$$

*Proof.* From definition of the Bregman projection step, starting from (29) we have

$$
\begin{aligned}
x^{t+1} \;&=\; \operatorname{Proj}^h_{\mathcal{X}}(\nabla h^*(y^{t+1})) \;=\; \operatorname*{argmin}_{x \in \mathcal{X}} \mathcal{D}_h(x, \nabla h^*(y^{t+1})) \\
&\in\; \operatorname*{argmin}_{x \in \mathcal{X}} \nabla h(x) - \nabla h(\nabla h^*(y^{t+1})) - \big\langle \nabla h(\nabla h^*(y^{t+1})), x - \nabla h^*(y^{t+1}) \big\rangle \\
&\overset{(4)}{\in}\; \operatorname*{argmin}_{x \in \mathcal{X}} \nabla h(x) - y^{t+1} - \big\langle y^{t+1}, x - \nabla h^*(y^{t+1}) \big\rangle \\
&\in\; \operatorname*{argmin}_{x \in \mathcal{X}} \nabla h(x) - \langle x, y^{t+1} \rangle \\
&\overset{(29)}{\in}\; \operatorname*{argmin}_{x \in \mathcal{X}} \nabla h(x) - \langle x, \nabla h(x^t) - \eta_t \nabla V(x)|_{x=x^t} \rangle \\
&\in\; \operatorname*{argmin}_{x \in \mathcal{X}} \eta_t \langle x, \nabla V(x)|_{x=x^t} \rangle + \nabla h(x) - \nabla h(x^t) - \langle \nabla h(x^t), x - x^t \rangle \\
&\in\; \operatorname*{argmin}_{x \in \mathcal{X}} \eta_t \langle x, \nabla V(x)|_{x=x^t} \rangle + \mathcal{D}_h(x, x^t),
\end{aligned}
$$

where the second and the last lines are both obtained by the definition of the Bregman divergence. $\square$

The one-step update in (31) is often taken as the definition of mirror descent [8], which provides a proximal view point of mirror descent, i.e., a gradient step in the primal space with a regularization of Bregman divergence.

# C  AMPO for specific mirror maps

In this section, we give the derivations for Example 3.2, which is based on the Karush-Kuhn-Tucker (KKT) conditions [43, 52], and then provide details about the $\omega$-potential mirror map class from Section 3.1.

## C.1  Derivation of Example 3.2

We give here the derivation of Example 3.2. Let $h$ be the negative entropy mirror map, that is

$$
h(\pi_s) = \sum_{a \in \mathcal{A}} \pi(a \mid s) \log(\pi(a \mid s)), \qquad \forall \pi_s \in \Delta(\mathcal{A}) \text{ and } \forall s \in \mathcal{S}.
$$

For every state $s \in \mathcal{S}$, we solve the minimization problem

$$
\pi^\theta_s \in \operatorname*{argmin}_{\pi_s \in \Delta(\mathcal{A})} \mathcal{D}_h(\pi_s, \nabla h^*(\eta f^\theta_s))
$$

through the KKT conditions. We formalize it as

$$
\begin{aligned}
\pi^\theta_s \in \; &\operatorname*{argmin}_{\pi_s \in \mathbb{R}^{|\mathcal{A}|}} \; \mathcal{D}_h(\pi_s, \nabla h^*(\eta f^\theta_s)) \\
&\text{subject to } \langle \pi_s, \mathbf{1} \rangle = 1, \\
&\qquad\qquad\quad \pi(a \mid s) \geq 0, \qquad \forall\, a \in \mathcal{A},
\end{aligned}
$$

where $\mathbf{1}$ denotes a vector in $\mathbb{R}^{|\mathcal{A}|}$ with coordinates equal to 1 element-wisely. The conditions then become

$$
\begin{aligned}
\text{(stationarity)} \qquad & \log(\pi^\theta_s) - \eta f^\theta_s + (\lambda_s + 1)\mathbf{1} - c_s = 0, \\
\text{(complementary slackness)} \qquad & c^a_s \pi^\theta(a \mid s) = 0, \quad \forall\, a \in \mathcal{A}, \\
\text{(primal feasibility)} \qquad & \langle \pi^\theta_s, \mathbf{1} \rangle = 1,\; \pi^\theta(a \mid s) \geq 0, \qquad \forall\, a \in \mathcal{A}, \\
\text{(dual feasibility)} \qquad & c^a_s \geq 0, \quad \forall\, a \in \mathcal{A},
\end{aligned}
$$

where $\log(\pi_s)$ is applied element-wisely, $\lambda_s \in \mathbb{R}$ and $c^a_s \in \mathbb{R}$ are the dual variables, and $c_s$ devotes the vector $[c^a_s]_{a \in \mathcal{A}}$. It is easy to verify that the solution

$$
\pi^\theta_s = \frac{\exp(\eta f^\theta_s)}{\|\exp(\eta f^\theta_s)\|_1},
$$

with $\lambda_s = \log\left(\sum_{a \in \mathcal{A}} \exp(\eta f^\theta(s,a))\right) - 1$ and $c_s^a = 0$ for all $a \in \mathcal{A}$, satisfies all the conditions.

When $f^\theta(s,a) = \theta_{s,a}$ we obtain the tabular softmax policy $\pi^\theta(a \mid s) \propto \exp(\eta\theta_{s,a})$. When $f^\theta(s,a) = \theta^\top\phi(s,a)$ is a linear function, for $\theta \in \mathbb{R}^d$ and for a feature function $\phi : \mathcal{S} \times \mathcal{A} \to \mathbb{R}^d$, we obtain the log-linear policy $\pi^\theta(a \mid s) \propto \exp(\eta\theta^\top\phi(s,a))$. When $f^\theta : \mathcal{S} \times \mathcal{A} \to \mathbb{R}$ is a neural network, we obtain the softmax neural policy $\pi^\theta(a \mid s) \propto \exp(\eta f^\theta(s,a))$.

## C.2 More on $\omega$-potential mirror maps

In this section, we provide details about the $\omega$-potential mirror map class from Section 3.1, including the derivation of (14), several instantiations of $\omega$-potential mirror map mentioned in Section 3.1 with their derivations, and an iterative algorithm to find approximately the Bregman projection induced by $\omega$-potential mirror map when an exact solution is not available.

We give a different but equivalent formulation of Proposition 2 of Krichene et al. [50].

**Proposition C.1.** *For $u \in (-\infty, +\infty]$ and $\omega \leq 0$, an increasing $C^1$-diffeomorphism $\phi : (-\infty, u) \to (\omega, +\infty)$ is called an $\omega$-potential if*

$$\lim_{x \to -\infty} \phi(x) = \omega, \quad \lim_{x \to u} \phi(x) = +\infty, \quad \int_0^1 \phi^{-1}(x)dx \leq \infty.$$

*Let the mirror map $h_\phi$ be defined as*

$$h_\phi(\pi_s) = \sum_{a \in \mathcal{A}} \int_1^{\pi(a|s)} \phi^{-1}(x)dx.$$

*We have that $\pi_s^t$ is a solution to the Bregman projection*

$$\min_{\pi \in \Delta_s} Proj_{\Delta(\mathcal{A})}^h(\nabla h_\phi^*(\eta_{t-1}f_s^t)),$$

*if and only if there exist a normalization constant $\lambda_s^t \in \mathbb{R}$ such that*

$$\pi^t(a \mid s) = \sigma(\phi(\eta_{t-1}f^t(s,a) + \lambda_s^t)), \quad \forall a \in \mathcal{A}, \tag{32}$$

*and $\sum_{a \in \mathcal{A}} \pi^t(a \mid s) = 1$, where for all $s \in \mathcal{S}$ and $\sigma(z) = \max(z,0)$ for $z \in \mathbb{R}$.*

We can now use Proposition C.1 to derive (14).

Consider an $\omega$-potential mirror map $h_\phi$ associated with an $\omega$-potential $\phi$. By definition, we have

$$\nabla h_\phi(\pi_s^t) = [\phi^{-1}(\pi^t(a \mid s))]_{a \in \mathcal{A}}. \tag{33}$$

Plugging (32) into (33), we have

$$
\begin{aligned}
\nabla h_\phi(\pi_s^t) &\overset{(33)}{=} [\phi^{-1}(\pi^t(a \mid s))]_{a \in \mathcal{A}} \\
&\overset{(32)}{=} \left[\phi^{-1}\left(\sigma\left(\phi(\eta_{t-1}f^t(s,a) + \lambda_s^t)\right)\right)\right]_{a \in \mathcal{A}} \\
&= \left[\max\left(\phi^{-1}\left(\phi(\eta_{t-1}f^t(s,a) + \lambda_s^t)\right), \phi^{-1}(0)\right)\right]_{a \in \mathcal{A}} \\
&= \left[\max\left(\eta_{t-1}f^t(s,a) + \lambda_s^t, \phi^{-1}(0)\right)\right]_{a \in \mathcal{A}} \\
&= \left[\max\left(\eta_{t-1}f^t(s,a), \phi^{-1}(0) - \lambda_s^t\right)\right]_{a \in \mathcal{A}} + \lambda_s^t,
\end{aligned}
$$

where the third line is obtained by using the increasing property of $\phi^{-1}$, as $\phi$ is increasing. Finally, plugging the above expression of $\nabla h_\phi(\pi_s^t)$ into Line 1, we obtain (14), which is

$$\theta^{t+1} \in \text{argmin}_{\theta \in \Theta} \mathbb{E}_{(s,a) \sim v^t}\left[(f^\theta(s,a) - Q^t(s,a) - \eta_t^{-1}\max(\eta_{t-1}f^t(s,a), \phi^{-1}(0) - \lambda_s^t))^2\right],$$

where the term $\lambda_s^t$ is dropped, as it is constant over actions and does not affect the resulting policy.

Once (14) is obtained, we can instantiate AMPO for mirror maps belonging to this class. We highlight that due to the definition of the Bregman divergence, two mirror maps that only differ for a constant term are equivalent and generate the same algorithm. We start with the negative entropy, which leads to a closed solution for $\lambda_s^t$ and therefore for the Bregman projection.

**Negative entropy.** Let $\phi(x) = \exp(x - 1)$, which is an $\omega$-potential with $\omega = 0$, $u = +\infty$, and

$$\int_0^1 \phi^{-1}(x)dx \;=\; \int_0^1 \log(x) + 1 dx \;=\; [x\log(x)]_0^1 = 0 \leq +\infty.$$

The mirror map $h_\phi$ becomes the negative entropy, as

$$h_\phi(\pi_s) = \sum_{a \in \mathcal{A}} \int_1^{\pi(a|s)} (\log(x) + 1)dx = \sum_{a \in \mathcal{A}} \pi(a \mid s)\log(\pi(a \mid s)),$$

and the associated Bregman divergence becomes the KL divergence, i.e., $D_{h_\phi}(\pi_s, \bar{\pi}_s) = \mathrm{KL}(\pi_s, \bar{\pi}_s)$. Equation (17) follows from Equation (14) by the fact that

$$\phi^{-1}(0) = \log(0) + 1 = -\infty,$$

which means that $\max(\eta_{t-1}f^t, \phi^{-1}(0) - \lambda_s^t) = \eta_{t-1}f^t$ element-wisely.

As we showed in Appendix C.1, the Bregman projection (Line 2 of Algorithm 1) for the negative entropy has a closed form which is $\pi_s^{t+1} \propto \exp(\eta_t f_s^{t+1})$.

In NPG with softmax tabular policies [1, 93], at time $t$, the updates for the policies have

$$\pi_s^{t+1} \propto \pi_s^t \odot \exp(\eta_t Q_s^t), \quad \forall s \in \mathcal{S}. \tag{34}$$

When considering AMPO with $f^\theta(s, a) = \theta_{s,a} \in \mathbb{R}$, from (17), we obtain that for all $s, a \in \mathcal{S}, \mathcal{A}$, we have

$$\theta^{t+1} \in \operatorname*{argmin}_{\theta \in \mathbb{R}^{|\mathcal{S}| \times |\mathcal{A}|}} \left\| \theta - Q^t - \frac{\eta_{t-1}}{\eta_t}\theta^t \right\|_{L_2(v^t)}^2 \iff f^{t+1}(s, a) = Q^t(s, a) + \frac{\eta_{t-1}}{\eta_t}f^t(s, a)$$

$$\iff \eta_t f^{t+1}(s, a) = \eta_t Q^t(s, a) + \eta_{t-1}f^t(s, a).$$

With the above expression, we have the AMPO's updates for the policies rewritten as

$$\pi_s^{t+1} \propto \exp(\eta_t f_s^{t+1})$$
$$= \exp(\eta_t Q_s^t + \eta_{t-1}f_s^t)$$
$$\propto \pi_s^t \odot \exp(\eta_t Q_s^t), \quad \forall s \in \mathcal{S},$$

which recovers (34). In particular, the summation in the second line is element-wise and the third line is obtained because of $\pi_s^t \propto \exp(\eta_{t-1}f_s^t)$, as shown in Appendix C.1.

In Q-NPG with log-linear policies [1, 98, 2], at time $t$, the updates for the policies have

$$\pi^{t+1}(a \mid s) \propto \pi^t(a \mid s)\exp(\eta_t \phi(s, a)^\top w^t), \tag{35}$$

where $\phi : \mathcal{S} \times \mathcal{A} \to \mathbb{R}^d$ is a feature map, and

$$w^t \in \operatorname*{argmin}_{w \in \mathbb{R}^d} \mathbb{E}_{(s,a) \sim d_\mu^t} \left[ (Q^t(s, a) - \phi(s, a)^\top w)^2 \right]. \tag{36}$$

Like in the tabular case, when considering AMPO with $\theta \in \mathbb{R}^d$, $f^\theta(s, a) = \phi(s, a)^\top \theta$ and $v^t = d_\mu^t$, from (17), we obtain that for all $s, a \in \mathcal{S}, \mathcal{A}$, we have

$$\theta^{t+1} \in \operatorname*{argmin}_{\theta \in \mathbb{R}^d} \mathbb{E}_{(s,a) \sim d_\mu^t} \left[ \left( \phi(s, a)^\top \theta - Q^t(s, a) - \frac{\eta_{t-1}}{\eta_t}\phi(s, a)^\top \theta^t \right)^2 \right]$$

$$\in \operatorname*{argmin}_{\theta \in \mathbb{R}^d} \mathbb{E}_{(s,a) \sim d_\mu^t} \left[ \left( Q^t(s, a) - \phi(s, a)^\top \underbrace{\left( \theta - \frac{\eta_{t-1}}{\eta_t}\theta^t \right)}_{w} \right)^2 \right].$$

Compared the above form with (36), we obtain that

$$w^t = \theta^{t+1} - \frac{\eta_{t-1}}{\eta_t}\theta^t \iff \eta_t \theta^{t+1} = \eta_t w^t + \eta_{t-1}\theta^t. \tag{37}$$

**Algorithm 2:** Bregman projection for $\epsilon$-negative entropy

---

**Input:** vector to project $x \in \mathbb{R}^{|\mathcal{A}|}$, parameter $\epsilon$.
1 Initialize $y = \exp(x)$ element-wisely.
2 Let $y^{(i)}$ be the $i$-th smallest element of $y$.
3 Let $i^\star$ be the smallest index for which

$$(1 + \epsilon(|\mathcal{A}| - i + 1))y^{(i)} - \epsilon \sum_{j \geq i} y^{(j)} > 0.$$

Set

$$\lambda = \frac{\sum_{i \geq i^\star} y^{(i)}}{1 + \epsilon(|\mathcal{A}| - i^\star + 1)}.$$

**Return:** the projected vector $(\sigma(-\epsilon + y_a/\lambda))_{a \in \mathcal{A}}$.

---

So, the AMPO's updates for the policies can be rewritten as

$$\begin{aligned}
\pi^{t+1}(a \mid s) &\propto \exp(\eta_t \phi(s,a)^\top \theta^{t+1}) \\
&\stackrel{(37)}{=} \exp(\phi(s,a)^\top(\eta_t w^t + \eta_{t-1}\theta^t)) \\
&\propto \pi^t(a \mid s)\exp(\eta_t \phi(s,a)^\top w^t),
\end{aligned}$$

where the last line is obtained because of $\pi_s^t \propto \exp(\eta_{t-1}f_s^t)$, as shown in Appendix C.1, and we recover (35).

We next present the squared $\ell_2$-norm and the $\epsilon$-negative entropy. For these two mirror maps, the Bregman projection can be computed exactly but has no closed form.

**Squared $\ell_2$-norm.** Let $\phi$ be the identity function. The mirror map $h_\phi$ becomes the squared $\ell_2$-norm, up to a constant term, as

$$h_\phi(\pi_s) = \sum_{a \in \mathcal{A}} \int_1^{\pi(a|s)} x \, dx = \frac{1}{2}\sum_{a \in \mathcal{A}}\left(\pi(a \mid s)^2 - 1\right).$$

The associated Bregman divergence becomes the squared Euclidean distance, i.e., $D_{h_\phi}(\pi_s, \bar{\pi}_s) = \frac{1}{2}\|\pi_s - \bar{\pi}_s\|_2^2$, and $\nabla h^*(\cdot)$ is the identity function. The update in (15) follows immediately and the Bregman projection step with the Euclidean distance becomes, for all $s \in \mathcal{S}$,

$$\pi_s^{t+1} = \mathrm{Proj}_{\Delta(\mathcal{A})}^{h_\phi}(\nabla h^*(\eta_t f_s^{t+1})) = \mathrm{Proj}_{\Delta(\mathcal{A})}^{l_2}(\eta_t f_s^{t+1}) = \underset{p \in \Delta(\mathcal{A})}{\mathrm{argmin}}\left\|p - \eta_t f_s^{t+1}\right\|_2^2. \tag{38}$$

In the projected-Q descent for tabular policies developed by Xiao [93], at time $t$, the updates for the policies are

$$\pi_s^{t+1} \in \underset{p \in \Delta(\mathcal{A})}{\mathrm{argmin}}\left\|\pi_s^t + \eta_t Q_s^t - p\right\|_2^2, \quad \forall s \in \mathcal{S}. \tag{39}$$

When considering AMPO with $f^\theta(s,a) = \theta_{s,a}$ and $\Theta = \mathbb{R}^{|\mathcal{S}| \times |\mathcal{A}|}$, (15) is solved with

$$f^{t+1}(s,a) = \theta_{s,a}^{t+1} = Q^t(s,a) + \eta_t^{-1}\pi^t(a \mid s).$$

Plugging the above expression into (38), we recover (39).

Notice that the Euclidean projection onto the probability simplex can be obtained exactly, as shown by Wang and Carreira-Perpiñán [91].

**$\epsilon$-negative entropy [50].** Let $\epsilon \geq 0$ and define the $\epsilon$-exponential potential as $\phi(x) = \exp(x-1) - \epsilon$. The mirror map $h_\phi$ becomes

$$h_\phi(\pi_s) = \sum_{a \in \mathcal{A}} \int_1^{\pi(a|s)} (\log(x+\epsilon)+1)dx = \sum_{a \in \mathcal{A}}\left[(\pi(a \mid s) + \epsilon)\ln(\pi(a \mid s) + \epsilon) - (1+\epsilon)\ln(1+\epsilon)\right].$$

---

**Algorithm 3:** Bregman projection for $\omega$-potential mirror maps

---

**Input:** vector to project $x \in \mathbb{R}^{|\mathcal{A}|}$, $\omega$-potential $\phi$, precision $\varepsilon$.

**1** Initialize

$$\bar{\nu} = \phi^{-1}(1) - \max_{a \in \mathcal{A}} x_a$$
$$\underline{\nu} = \phi^{-1}(1/|\mathcal{A}|) - \max_{a \in \mathcal{A}} x_a$$

**2** Define $\tilde{x}(\nu) = (\sigma(\phi(x_a + \nu)))_{a \in \mathcal{A}}$.

   **while** $\|\tilde{x}(\bar{\nu}) - \tilde{x}(\underline{\nu})\|_1 > \varepsilon$ **do**

**3**     Let $\nu^+ \leftarrow (\bar{\nu} + \underline{\nu})/2$

**4**     **if** $\sum_{a \in \mathcal{A}} \tilde{x}_a(\nu^+) > 1$ **then**

**5**        $\bar{\nu} \leftarrow \nu^+$

**6**     **else**

**7**        $\underline{\nu} \leftarrow \nu^+$

**8** **Return** $\tilde{x}(\bar{\nu})$

---

An exact solution to the associated projection can then be found in $\widetilde{\mathcal{O}}(|\mathcal{A}|)$ computations using Algorithm 2, which has been proposed by Krichene et al. [50, Algorithm 4]. Additionally, following (14), the regression problem in Line 1 of Algorithm 1 becomes

$$\theta^{t+1} \in \underset{\theta \in \Theta}{\operatorname{argmin}} \left\| f^\theta - Q^t - \eta_t^{-1} \max(\eta_{t-1} f^t, 1 + \log(\epsilon) - \lambda_s^t) \right\|_{L_2(v^t)}^2,$$

where $\lambda_s^t$ can be obtained through Algorithm 2.

The Bregman projection for generic mirror maps can be computed approximately in $\widetilde{\mathcal{O}}(|\mathcal{A}|)$ computations through a bisection algorithm. Krichene et al. [50] propose one such algorithm, which we report in Algorithm 3 for completeness. We next provide two mirror maps that have appeared before in the optimization literature, but do not lead to an exact solution to the Bregman projection. We leave them as object for future work.

**Negative Tsallis entropy [74, 57].** Let $q > 0$ and define $\phi$ as

$$\phi_q(x) = \begin{cases} \exp(x-1) & \text{if } q = 1, \\ \left[\sigma\left(\frac{(q-1)x}{q}\right)\right]^{\frac{1}{q-1}} & \text{else.} \end{cases}$$

The mirror map $h_{\phi_q}$ becomes the negative Tsallis entropy, that is

$$h_{\phi_q}(\pi_s) = \sum \pi(a \mid s) \log_q(\pi(a \mid s)), \tag{40}$$

where, for $y > 0$,

$$\log_q(y) = \begin{cases} \log(y) & \text{if } q = 1, \\ \frac{-y^{q-1}}{q-1} & \text{else.} \end{cases}$$

If $q \neq 1$ and following (14), the regression problem in Line 1 of Algorithm 1 becomes

$$\theta^{t+1} \in \underset{\theta \in \Theta}{\operatorname{argmin}} \left\| f^\theta - Q^t - \frac{\eta_{t-1}}{\eta_t} f^t \right\|_{L_2(v^t)}^2,$$

**Hyperbolic entropy [34].** Let $b > 0$ and define $\phi$ as

$$\phi_b(x) = b \sinh(x)$$

The mirror map $h_{\phi_b}$ becomes the hyperbolic entropy, that is

$$h_{\phi_b}(\pi_s) = \sum_{a \in \mathcal{A}} \pi(a \mid s) \operatorname{arcsinh}(\pi(a \mid s)/b) - \sqrt{\pi(a \mid s)^2 + b^2}, \tag{41}$$

and, following (14), the regression problem in Line 1 of Algorithm 1 becomes

$$\theta^{t+1} \in \underset{\theta \in \Theta}{\operatorname{argmin}} \left\| f^\theta - Q^t - \eta_t^{-1} \max(\eta_{t-1} f^t, -\lambda_s^t) \right\|_{L_2(v^t)}^2.$$

**Hyperbolic-tangent entropy.** Inspired by the hyperbolic entropy, we consider $\phi$ as
$$\phi(x) = \tanh(x)/2 + 0.5$$

The mirror map $h_\phi$ becomes

$$h_\phi(\pi_s) = \frac{1}{2}\sum_{a \in \mathcal{A}}(2\pi(a \mid s) - 1)\operatorname{arctanh}(2\pi(a \mid s) - 1) + \frac{1}{2}\log\pi(a \mid s)(1 - \pi(a \mid s)), \quad (42)$$

which, to the best of our knowledge, is not equivalent to any mirror map studied in the literature. Following (14), the regression problem in Line 1 of Algorithm 1 becomes

$$\theta^{t+1} \in \operatorname*{argmin}_{\theta \in \Theta}\left\|f^\theta - Q^t - \frac{\eta_{t-1}}{\eta_t}f^t\right\|_{L_2(v^t)}^2.$$

Regarding the limitations of the $\omega$-potential mirror map class, we are aware of two previously used mirror maps that cannot be recovered using $\omega$-potentials: $h(x) = \frac{1}{2}x^\top \mathbf{A}x$ [32], for some positive-definite matrix $\mathbf{A}$, which generates the Mahalanobis distance, and $p$-norms, i.e. $h(x) = \|x\|_p^2$ [74]. Note that the case where $h(x) = \|x\|_p^p$ can be recovered by setting $\phi(x) = (px)^{p/(1-p)}$.

We note that tuning the mirror map and the step-size can lead AMPO to encompass the case of deterministic policies, which can be obtained when using softmax policies by sending the step-size to infinity, effectively turning the softmax operator into a max operator. Another simple way of introducing deterministic policies in our framework is to choose the mirror map to be the Euclidean norm and to choose the step-size large enough. Doing so will cause the Bregman projection to put all the probability on the action that corresponds to the maximum value of $\hat{f}_s^\theta$. Our results hold in this setting because our analysis does not use the policy gradient theorem (3), which has a different expression for deterministic policies [83].

# D    Deferred proofs from Section 4.1

## D.1    Proof of Lemma 4.1

Here we provide the proof of Lemma 4.1, an application of the three-point descent lemma that accommodates arbitrary parameterized functions. Lemma 4.1 is the key tool for our analysis of AMPO. It is a generalization of both Xiao [93, Equation (44)] and Yuan et al. [98, Equation (50)] thanks to our two-step PMD framework. First, we recall some technical conditions of the mirror map [14, Chapter 4].

Suppose that $\mathcal{Y} \subset \mathbb{R}^{|\mathcal{A}|}$ is a closed convex set, we say a function $h : \mathcal{Y} \to \mathbb{R}$ is a *mirror map* if it satisfies the following properties:

(i) $h$ is strictly convex and differentiable;

(ii) $h$ is essentially smooth, i.e., the graident of $h$ diverges on the boundary of $\mathcal{Y}$, that is $\lim_{x \to \partial\mathcal{Y}}\|\nabla h(x)\| \to \infty$;

(iii) the gradient of $h$ takes all possible values, that is $\nabla h(\mathcal{Y}) = \mathbb{R}^{|\mathcal{A}|}$.

To prove Lemma 4.1, we also need the following rather simple properties, i.e., the three-point identity and the generalized Pythagorean theorem, satisfied by the Bregman divergence. We provide their proofs for self-containment.

**Lemma D.1** (Three-point identity, Lemma 3.1 in Chen and Teboulle [19])**.** *Let $h$ be a mirror map. For any $a, b$ in the relative interior of $\mathcal{Y}$ and $c \in \mathcal{Y}$, we have that:*

$$\mathcal{D}_h(c, a) + \mathcal{D}_h(a, b) - \mathcal{D}_h(c, b) = \langle\nabla h(b) - \nabla h(a), c - a\rangle. \quad (43)$$

*Proof.* Using the definition of the Bregman divergence $\mathcal{D}_h$, we have

$$\langle\nabla h(a), c - a\rangle = h(c) - h(a) - \mathcal{D}_h(c, a), \quad (44)$$
$$\langle\nabla h(b), a - b\rangle = h(a) - h(b) - \mathcal{D}_h(a, b), \quad (45)$$
$$\langle\nabla h(b), c - b\rangle = h(c) - h(b) - \mathcal{D}_h(c, b). \quad (46)$$

Subtracting (44) and (45) from (46) yields (43). $\qquad\square$

**Lemma D.2** (Generalized Pythagorean Theorem of Bregman divergence, Lemma 4.1 in Bubeck [14])**.** *Let $\mathcal{X} \subseteq \mathcal{Y}$ be a closed convex set. Let $h$ be a mirror map defined on $\mathcal{Y}$. Let $x \in \mathcal{X}$, $y \in \mathcal{Y}$ and $y^\star = Proj_{\mathcal{X}}^{h}(y)$, then*

$$\langle \nabla h\,(y^\star) - \nabla h(y), y^\star - x \rangle \le 0,$$

*which also implies*

$$\mathcal{D}_h\,(x, y^\star) + \mathcal{D}_h\,(y^\star, y) \le \mathcal{D}_h(x, y). \tag{47}$$

*Proof.* From the definition of $y^\star$, which is

$$y^\star \in \underset{y' \in \mathcal{X}}{\arg\min}\, \mathcal{D}_h(y', y),$$

and from the first-order optimality condition [14, Proposition 1.3], with

$$\nabla_{y'} \mathcal{D}_h(y', y) = \nabla h(y') - \nabla h(y), \quad \text{for all } y' \in \mathcal{Y},$$

we have

$$\langle \nabla_{y'} \mathcal{D}_h(y', y)|_{y'=y^\star}, y^\star - x \rangle \le 0 \quad \Longrightarrow \quad \langle \nabla h\,(y^\star) - \nabla(y), y^\star - x \rangle \le 0,$$

which implies (47) by applying the definition of Bregman divergence and rearranging terms. $\square$

Now we are ready to prove Lemma 4.1.

**Lemma D.3** (Lemma 4.1)**.** *Let $\mathcal{Y} \subset \mathbb{R}^{|\mathcal{A}|}$ be a closed convex set with $\Delta(\mathcal{A}) \subseteq \mathcal{Y}$. For any policies $\pi \in \Delta(\mathcal{A})^{\mathcal{S}}$ and $\bar{\pi}$ in the relative interior of $\Delta(\mathcal{A})^{\mathcal{S}}$, any function $f^\theta$ with $\theta \in \Theta$, any $s \in \mathcal{S}$ and for $\eta > 0$, we have that,*

$$\langle \eta f_s^\theta - \nabla h(\bar{\pi}_s), \pi_s - \tilde{\pi}_s \rangle \le \mathcal{D}_h(\pi_s, \bar{\pi}_s) - \mathcal{D}_h(\tilde{\pi}_s, \bar{\pi}_s) - \mathcal{D}_h(\pi, \tilde{\pi}_s),$$

*where $\tilde{\pi}$ is induced by $f^\theta$ and $\eta$ according to Definition 3.1, that is, for all $s \in \mathcal{S}$,*

$$\tilde{\pi}_s = Proj_{\Delta(\mathcal{A})}^{h}\left( \nabla h^*(\eta f_s^\theta) \right) = \underset{\pi'_s \in \Delta(\mathcal{A})}{\arg\min}\, \mathcal{D}_h(\pi'_s, \nabla h^*(\eta f_s^\theta)). \tag{48}$$

*Proof.* For clarity of exposition, let $p_s = \nabla h^*(\eta f_s^\theta)$. Plugging $a = \bar{\pi}_s$, $b = p_s$ and $c = \pi_s$ in the three-point identity lemma D.1, we obtain

$$\mathcal{D}_h(\pi_s, \bar{\pi}_s) - \mathcal{D}_h(\pi_s, p_s) + \mathcal{D}_h(\bar{\pi}_s, p_s) = \langle \nabla h(\bar{\pi}_s) - \nabla h(p_s), \bar{\pi}_s - \pi_s \rangle. \tag{49}$$

Similarly, plugging $a = \bar{\pi}_s, b = p_s$ and $c = \tilde{\pi}_s$ in the three-point identity lemma D.1, we obtain

$$\mathcal{D}_h(\tilde{\pi}_s, \bar{\pi}_s) - \mathcal{D}_h(\tilde{\pi}_s, p_s) + \mathcal{D}_h(\bar{\pi}_s, p_s) = \langle \nabla h(\bar{\pi}_s) - \nabla h(p_s), \bar{\pi}_s - \tilde{\pi}_s \rangle. \tag{50}$$

From (49), we have

$$\begin{aligned}
& \mathcal{D}_h(\pi_s, \bar{\pi}_s) - \mathcal{D}_h(\pi_s, p_s) + \mathcal{D}_h(\bar{\pi}_s, p_s) \\
=\ & \langle \nabla h(\bar{\pi}_s) - \nabla h(p_s), \bar{\pi}_s - \pi_s \rangle \\
=\ & \langle \nabla h(\bar{\pi}_s) - \nabla h(p_s), \bar{\pi}_s - \tilde{\pi}_s \rangle + \langle \nabla h(\bar{\pi}_s) - \nabla h(p_s), \tilde{\pi}_s - \pi_s \rangle \\
\overset{(50)}{=}\ & \mathcal{D}_h(\tilde{\pi}_s, \bar{\pi}_s) - \mathcal{D}_h(\tilde{\pi}_s, p_s) + \mathcal{D}_h(\bar{\pi}_s, p_s) + \langle \nabla h(\bar{\pi}_s) - \nabla h(p_s), \tilde{\pi}_s - \pi_s \rangle.
\end{aligned}$$

By rearranging terms, we have

$$\mathcal{D}_h(\pi_s, \bar{\pi}_s) - \mathcal{D}_h(\tilde{\pi}_s, \bar{\pi}_s) - \mathcal{D}_h(\pi_s, p_s) + \mathcal{D}_h(\tilde{\pi}_s, p_s) = \langle \nabla h(\bar{\pi}_s) - \nabla h(p_s), \tilde{\pi}_s - \pi_s \rangle. \tag{51}$$

From the Generalized Pythagorean Theorem of the Bregman divergence in Lemma D.2, also known as non-expansivity property, and from the fact that $\tilde{\pi}_s = Proj_{\Delta(\mathcal{A})}^{h}(p_s)$, we have that

$$\mathcal{D}_h(\pi_s, \tilde{\pi}_s) + \mathcal{D}_h(\tilde{\pi}_s, p_s) \le \mathcal{D}_h(\pi_s, p_s) \quad \Longleftrightarrow \quad -\mathcal{D}_h(\pi_s, p_s) + \mathcal{D}_h(\tilde{\pi}_s, p_s) \le -\mathcal{D}_h(\pi_s, \tilde{\pi}_s).$$

Plugging the above inequality into the left hand side of (51) yields

$$\mathcal{D}_h(\pi_s, \bar{\pi}_s) - \mathcal{D}_h(\tilde{\pi}_s, \bar{\pi}_s) - \mathcal{D}_h(\pi_s, \tilde{\pi}_s) \ge \langle \nabla h(\bar{\pi}_s) - \nabla h(p_s), \tilde{\pi}_s - \pi_s \rangle,$$

which concludes the proof with $\nabla h(p_s) = \eta f_s^\theta$. $\square$

We also provide an alternative proof of Lemma 4.1 later in Appendix F.

## D.2 Bounding errors

In this section, we will bound error terms of the type

$$\mathbb{E}_{s \sim d_\mu^\pi, a \sim \pi_s} \left[ Q^t(s,a) + \eta_t^{-1} [\nabla h(\pi_s^t)]_a - f^{t+1}(s,a) \right], \tag{52}$$

where $(d_\mu^\pi, \pi) \in \{(d_\mu^\star, \pi^\star), (d_\mu^{t+1}, \pi^{t+1}), (d_\mu^\star, \pi^t), (d_\mu^{t+1}, \pi^t)\}$. These error terms appear in the forthcoming proofs of our results and directly induce the error floors in the convergence rates.

In the rest of Appendix D, let $q^t : \mathcal{S} \times \mathcal{A} \to \mathbb{R}$ such that, for every $s \in \mathcal{S}$,

$$q_s^t := f_s^{t+1} - \eta_t^{-1} \nabla h(\pi_s^t) \in \mathbb{R}^{|\mathcal{A}|}.$$

So (52) can be rewritten as

$$\mathbb{E}_{s \sim d_\mu^\pi, a \sim \pi_s} \left[ Q^t(s,a) + \eta_t^{-1} [\nabla h(\pi_s^t)]_a - f^{t+1}(s,a) \right] = \mathbb{E}_{s \sim d_\mu^\pi, a \sim \pi_s} \left[ Q^t(s,a) - q^t(s,a) \right]. \tag{53}$$

To bound it, let $(v^t)_{t \geq 0}$ be a sequence of distributions over states and actions. By using Cauchy-Schwartz's inequality, we have

$$
\begin{aligned}
\mathbb{E}_{s \sim d_\mu^\pi, a \sim \pi_s} & \left[ Q^t(s,a) - q^t(s,a) \right] \\
&= \int_{s \in \mathcal{S}, a \in \mathcal{A}} \frac{d_\mu^\pi(s)\pi(a \mid s)}{\sqrt{v^t(s,a)}} \cdot \sqrt{v^t(s,a)}(Q^t(s,a) - q^t(s,a)) \\
&\leq \sqrt{ \int_{s \in \mathcal{S}, a \in \mathcal{A}} \frac{\left(d_\mu^\pi(s)\pi(a \mid s)\right)^2}{v^t(s,a)} \cdot \int_{s \in \mathcal{S}, a \in \mathcal{A}} v^t(s,a)(Q^t(s,a) - q^t(s,a))^2 } \\
&= \sqrt{ \mathbb{E}_{(s,a) \sim v^t} \left[ \left( \frac{d_\mu^\pi(s)\pi(a \mid s)}{v^t(s,a)} \right)^2 \right] \cdot \mathbb{E}_{(s,a) \sim v^t} \left[ (Q^t(s,a) - q^t(s,a))^2 \right] } \\
&\leq \sqrt{ C_v \mathbb{E}_{(s,a) \sim v^t} \left[ (Q^t(s,a) - q^t(s,a))^2 \right] },
\end{aligned}
$$

where the last line is obtained by Assumption (A2). Using the concavity of the square root and Assumption (A1), we have that

$$\mathbb{E} \left[ \mathbb{E}_{s \sim d_\mu^\pi, a \sim \pi_s} \left[ Q^t(s,a) - q^t(s,a) \right] \right] \leq \sqrt{C_v \varepsilon_{\text{approx}}}. \tag{54}$$

## D.3 Quasi-monotonic updates – Proof of Proposition 4.2

In this section, we show Proposition 4.2 with its proof that the AMPO updates guarantee a quasi-monotonic property, i.e., a non-decreasing property up to a certain error floor due to the approximation error, which allows us to establish an important recursion about the AMPO iterates next. First, we recall the performance difference lemma [41] which is the second key tool for our analysis and a well known result in the RL literature. Here we use a particular form of the lemma presented by Xiao [93, Lemma 1].

**Lemma D.4** (Performance difference lemma, Lemma 1 in [93]). *For any policy* $\pi, \pi' \in \Delta(\mathcal{A})^\mathcal{S}$ *and* $\mu \in \Delta(\mathcal{S})$,

$$V^\pi(\mu) - V^{\pi'}(\mu) = \frac{1}{1-\gamma} \mathbb{E}_{s \sim d_\mu^\pi} \left[ \left\langle Q_s^{\pi'}, \pi_s - \pi_s' \right\rangle \right].$$

For clarity of exposition, we introduce the notation

$$\tau := \frac{2\sqrt{C_v \varepsilon_{\text{approx}}}}{1-\gamma}.$$

Proposition 4.2 characterizes the non-decreasing property of AMPO. The error bound (54) in Appendix D.2 will be used to prove the the result.

**Proposition D.5** (Proposition 4.2). *For the iterates of Algorithm 1, at each time $t \geq 0$, we have*

$$\mathbb{E}[V^{t+1}(\mu) - V^t(\mu)] \geq \mathbb{E}\left[\mathbb{E}_{s \sim d_\mu^{t+1}}\left[\frac{\mathcal{D}_h(\pi_s^{t+1}, \pi_s^t) + \mathcal{D}_h(\pi_s^t, \pi_s^{t+1})}{\eta_t(1-\gamma)}\right]\right] - \tau.$$

*Proof.* Using Lemma 4.1 with $\bar{\pi} = \pi^t$, $f^\theta = f^{t+1}$, $\eta = \eta_t$, thus $\tilde{\pi} = \pi^{t+1}$ by Definition 3.1 and Algorithm 1, and $\pi_s = \pi_s^t$, we have

$$\langle \eta_t q_s^t, \pi_s^t - \pi_s^{t+1}\rangle \leq \mathcal{D}_h(\pi_s^t, \pi_s^t) - \mathcal{D}_h(\pi_s^{t+1}, \pi_s^t) - \mathcal{D}_h(\pi_s^t, \pi_s^{t+1}). \tag{55}$$

By rearranging terms and noticing $\mathcal{D}_h(\pi_s^t, \pi_s^t) = 0$, we have

$$\langle \eta_t q_s^t, \pi_s^{t+1} - \pi_s^t\rangle \geq \mathcal{D}_h(\pi_s^{t+1}, \pi_s^t) + \mathcal{D}_h(\pi_s^t, \pi_s^{t+1}) \geq 0. \tag{56}$$

Then, by the performance difference lemma D.4, we have

$$
\begin{aligned}
(1-\gamma)\mathbb{E}[V^{t+1}(\mu) - V^t(\mu)] \;=\;& \mathbb{E}\left[\mathbb{E}_{s \sim d_\mu^{t+1}}\left[\langle Q_s^t, \pi_s^{t+1} - \pi_s^t\rangle\right]\right] \\
=\;& \mathbb{E}\left[\mathbb{E}_{s \sim d_\mu^{t+1}}\left[\langle q_s^t, \pi_s^{t+1} - \pi_s^t\rangle\right]\right] \\
& + \mathbb{E}\left[\mathbb{E}_{s \sim d_\mu^{t+1}}\left[\langle Q_s^t - q_s^t, \pi_s^{t+1} - \pi_s^t\rangle\right]\right] \\
\overset{(55)}{\geq}\;& \mathbb{E}\left[\mathbb{E}_{s \sim d_\mu^{t+1}}\left[\frac{\mathcal{D}_h(\pi_s^{t+1}, \pi_s^t) + \mathcal{D}_h(\pi_s^t, \pi_s^{t+1})}{\eta_t}\right]\right] \\
& - \left|\mathbb{E}\left[\mathbb{E}_{s \sim d_\mu^{t+1}}\left[\langle Q_s^t - q_s^t, \pi_s^{t+1} - \pi_s^t\rangle\right]\right]\right| \\
\geq\;& \mathbb{E}\left[\mathbb{E}_{s \sim d_\mu^{t+1}}\left[\frac{\mathcal{D}_h(\pi_s^{t+1}, \pi_s^t) + \mathcal{D}_h(\pi_s^t, \pi_s^{t+1})}{\eta_t}\right]\right] - \tau(1-\gamma),
\end{aligned}
$$

which concludes the proof after dividing both sides by $(1-\gamma)$. The last line follows from

$$
\begin{aligned}
\left|\mathbb{E}\left[\mathbb{E}_{s \sim d_\mu^{t+1}}\left[\langle Q_s^t - q_s^t, \pi_s^{t+1} - \pi_s^t\rangle\right]\right]\right| \;\leq\;& \left|\mathbb{E}\left[\mathbb{E}_{s \sim d_\mu^{t+1}, a \sim \pi_s^{t+1}}\left[Q^t(s,a) - q^t(s,a)\right]\right]\right| \tag{57} \\
& + \left|\mathbb{E}\left[\mathbb{E}_{s \sim d_\mu^{t+1}, a \sim \pi_s^t}\left[Q^t(s,a) - q^t(s,a)\right]\right]\right| \\
\overset{(54)}{\leq}\;& 2\sqrt{C_1 \varepsilon_{\text{error}}} = \tau(1-\gamma), \tag{58}
\end{aligned}
$$

where both terms are upper bounded by $\sqrt{C_v \varepsilon_{\text{approx}}}$ through (54) with $(d_\mu^\pi, \pi) = (d_\mu^{t+1}, \pi^{t+1})$ and $(d_\mu^\pi, \pi) = (d_\mu^{t+1}, \pi^t)$, respectively. $\qquad\square$

## D.4 Main passage – An important recursion about the AMPO method

In this section, we show an important recursion result for the AMPO updates, which will be used for both the sublinear and the linear convergence analysis of AMPO.

For clarity of exposition in the rest of Appendix D, let

$$\nu_t := \left\|\frac{d_\mu^\star}{d_\mu^{t+1}}\right\|_{L_\infty} := \sup_{s \in \mathcal{S}}\frac{d_\mu^\star(s)}{d_\mu^{t+1}(s)}.$$

For two different time $t, t' \geq 0$, let $\mathcal{D}_{t'}^t$ denote the expected Bregman divergence between the policy $\pi^t$ and policy $\pi^{t'}$, where the expectation is taken over the discounted state visitation distribution of the optimal policy $d_\mu^\star$, that is,

$$\mathcal{D}_{t'}^t := \mathbb{E}_{s \sim d_\mu^\star}\left[\mathcal{D}_h(\pi_s^t, \pi_s^{t'})\right].$$

Similarly, let $\mathcal{D}_t^\star$ denote the expected Bregman divergence between the optimal policy $\pi^\star$ and $\pi^t$, that is,

$$\mathcal{D}_t^\star := \mathbb{E}_{s \sim d_\mu^\star}\left[\mathcal{D}_h(\pi_s^\star, \pi_s^t)\right].$$

Let $\Delta_t := V^\star(\mu) - V^t(\mu)$ be the optimality gap.

We can now state the following important recursion result for the AMPO method.

**Proposition D.6** (Proposition 4.4). *Consider the iterates of Algorithm 1, at each time $t \geq 0$, we have*

$$\mathbb{E}\left[\frac{\mathcal{D}_t^{t+1}}{(1-\gamma)\eta_t} + \nu_\mu(\Delta_{t+1} - \Delta_t) + \Delta_t\right] \leq \mathbb{E}\left[\frac{\mathcal{D}_t^\star}{(1-\gamma)\eta_t} - \frac{\mathcal{D}_{t+1}^\star}{(1-\gamma)\eta_t}\right] + (1+\nu_\mu)\tau.$$

*Proof.* Using Lemma 4.1 with $\bar{\pi} = \pi^t$, $f^\theta = f^{t+1}$, $\eta = \eta_t$, and thus $\tilde{\pi} = \pi^{t+1}$ by Definition 3.1 and Algorithm 1, and $\pi_s = \pi_s^\star$, we have that

$$\langle \eta_t q_s^t, \pi_s^\star - \pi_s^{t+1}\rangle \leq \mathcal{D}_h(\pi^\star, \pi^t) - \mathcal{D}_h(\pi^\star, \pi^{t+1}) - \mathcal{D}_h(\pi^{t+1}, \pi^t),$$

which can be decomposed as

$$\langle \eta_t q_s^t, \pi_s^t - \pi_s^{t+1}\rangle + \langle \eta_t q_s^t, \pi_s^\star - \pi_s^t\rangle \leq \mathcal{D}_h(\pi^\star, \pi^t) - \mathcal{D}_h(\pi^\star, \pi^{t+1}) - \mathcal{D}_h(\pi^{t+1}, \pi^t).$$

Taking expectation with respect to the distribution $d_\mu^\star$ over states and with respect to the randomness of AMPO and dividing both sides by $\eta_t$, we have

$$\mathbb{E}\left[\mathbb{E}_{s \sim d_\mu^\star}\left[\langle q_s^t, \pi_s^t - \pi_s^{t+1}\rangle\right]\right] + \mathbb{E}\left[\mathbb{E}_{s \sim d_\mu^\star}\left[\langle q_s^t, \pi_s^\star - \pi_s^t\rangle\right]\right] \leq \frac{1}{\eta_t}\mathbb{E}[\mathcal{D}_t^\star - \mathcal{D}_{t+1}^\star - \mathcal{D}_t^{t+1}]. \quad (59)$$

We lower bound the two terms on the left hand side of (59) separately. For the first term, we have that

$$\begin{aligned}
\mathbb{E}\left[\mathbb{E}_{s \sim d_\mu^\star}\left[\langle q_s^t, \pi_s^t - \pi_s^{t+1}\rangle\right]\right] &\overset{(56)}{\geq} \left\|\frac{d_\mu^\star}{d_\mu^{t+1}}\right\|_{L_\infty} \mathbb{E}\left[\mathbb{E}_{s \sim d_\mu^{t+1}}\left[\langle q_s^t, \pi_s^t - \pi_s^{t+1}\rangle\right]\right] \\
&= \nu_{t+1}\mathbb{E}\left[\mathbb{E}_{s \sim d_\mu^{t+1}}\left[\langle Q_s^t, \pi_s^t - \pi_s^{t+1}\rangle\right]\right] \\
&\quad + \nu_{t+1}\mathbb{E}\left[\mathbb{E}_{s \sim d_\mu^{t+1}}\left[\langle q_s^t - Q_s^t, \pi_s^t - \pi_s^{t+1}\rangle\right]\right] \\
&\overset{(a)}{=} \nu_{t+1}(1-\gamma)\mathbb{E}\left[V^t(\mu) - V^{t+1}(\mu)\right] \\
&\quad + \nu_{t+1}\mathbb{E}\left[\mathbb{E}_{s \sim d_\mu^{t+1}}\left[\langle q_s^t - Q_s^t, \pi_s^t - \pi_s^{t+1}\rangle\right]\right] \\
&\overset{(57)}{\geq} \nu_{t+1}(1-\gamma)\mathbb{E}\left[V^t(\mu) - V^{t+1}(\mu)\right] - \nu_{t+1}\tau(1-\gamma) \\
&= \nu_{t+1}(1-\gamma)\mathbb{E}\left[\Delta_{t+1} - \Delta_t\right] - \nu_{t+1}\tau(1-\gamma),
\end{aligned}$$

where $(a)$ follows from Lemma D.4. For the second term, we have that

$$\begin{aligned}
\mathbb{E}\left[\mathbb{E}_{s \sim d_\mu^\star}\left[\langle q_s^t, \pi_s^\star - \pi_s^t\rangle\right]\right] &= \mathbb{E}\left[\mathbb{E}_{s \sim d_\mu^\star}\left[\langle Q_s^t, \pi_s^\star - \pi_s^t\rangle\right]\right] + \mathbb{E}\left[\mathbb{E}_{s \sim d_\mu^\star}\left[\langle q_s^t - Q_s^t, \pi_s^\star - \pi_s^t\rangle\right]\right] \\
&\overset{(b)}{=} \mathbb{E}[\Delta_t](1-\gamma) + \mathbb{E}\left[\mathbb{E}_{s \sim d_\mu^\star}\left[\langle q_s^t - Q_s^t, \pi_s^\star - \pi_s^t\rangle\right]\right] \\
&\overset{(c)}{\geq} \mathbb{E}[\Delta_t](1-\gamma) - \tau(1-\gamma),
\end{aligned}$$

where $(b)$ follows from Lemma D.4 and $(c)$ follows similarly to (57), i.e., by applying (54) twice with $(d_\mu^\pi, \pi) = (d_\mu^\star, \pi^\star)$ and $(d_\mu^\pi, \pi) = (d_\mu^\star, \pi^t)$.

Plugging the two bounds in (59), dividing both sides by $(1-\gamma)$ and rearranging, we obtain

$$\mathbb{E}\left[\frac{\mathcal{D}_t^{t+1}}{(1-\gamma)\eta_t} + \nu_{t+1}(\Delta_{t+1} - \Delta_t - \tau) + \Delta_t\right] \leq \mathbb{E}\left[\frac{\mathcal{D}_t^\star}{(1-\gamma)\eta_t} - \frac{\mathcal{D}_{t+1}^\star}{(1-\gamma)\eta_t}\right] + \tau.$$

From Proposition 4.2, we have that $\Delta_{t+1} - \Delta_t - \tau \leq 0$. Consequently, since $\nu_{t+1} \leq \nu_\mu$ by the definition of $\nu_\mu$ in Assumption (A3), one can lower bound the left hand side of the above inequality by replacing $\nu_{t+1}$ by $\nu_\mu$, that is,

$$\mathbb{E}\left[\frac{\mathcal{D}_t^{t+1}}{(1-\gamma)\eta_t} + \nu_\mu(\Delta_{t+1} - \Delta_t - \tau) + \Delta_t\right] \leq \mathbb{E}\left[\frac{\mathcal{D}_t^\star}{(1-\gamma)\eta_t} - \frac{\mathcal{D}_{t+1}^\star}{(1-\gamma)\eta_t}\right] + \tau,$$

which concludes the proof. □

## D.5 Proof of the sublinear convergence analysis

In this section, we derive the sublinear convergence result of Theorem 4.3 with non-decreasing step-size.

*Proof.* Starting from Proposition D.6

$$\mathbb{E}\left[\frac{\mathcal{D}_t^{t+1}}{(1-\gamma)\eta_t} + \nu_\mu\left(\Delta_{t+1} - \Delta_t\right) + \Delta_t\right] \leq \mathbb{E}\left[\frac{\mathcal{D}_t^\star}{(1-\gamma)\eta_t} - \frac{\mathcal{D}_{t+1}^\star}{(1-\gamma)\eta_t}\right] + (1+\nu_\mu)\tau.$$

If $\eta_t \leq \eta_{t+1}$,

$$\mathbb{E}\left[\frac{\mathcal{D}_t^{t+1}}{(1-\gamma)\eta_t} + \nu_\mu\left(\Delta_{t+1} - \Delta_t\right) + \Delta_t\right] \leq \mathbb{E}\left[\frac{\mathcal{D}_t^\star}{(1-\gamma)\eta_t} - \frac{\mathcal{D}_{t+1}^\star}{(1-\gamma)\eta_{t+1}}\right] + (1+\nu_\mu)\tau. \quad (60)$$

Summing up from 0 to $T-1$ and dropping some positive terms on the left hand side and some negative terms on the right hand side, we have

$$\sum_{t<T}\mathbb{E}\left[\Delta_t\right] \leq \frac{\mathcal{D}_0^\star}{(1-\gamma)\eta_0} + \nu_\mu\Delta_0 + T(1+\nu_\mu)\tau \leq \frac{\mathcal{D}_0^\star}{(1-\gamma)\eta_0} + \frac{\nu_\mu}{1-\gamma} + T(1+\nu_\mu)\tau.$$

Notice that $\Delta_0 \leq \frac{1}{1-\gamma}$ as $r(s,a) \in [0,1]$. By dividing $T$ on both side, we yield the proof of the sublinear convergence

$$V^\star(\mu) - \frac{1}{T}\sum_{t<T}\mathbb{E}\left[V^t(\mu)\right] \leq \frac{1}{T}\left(\frac{\mathcal{D}_0^\star}{(1-\gamma)\eta_0} + \frac{\nu_\mu}{1-\gamma}\right) + (1+\nu_\mu)\tau.$$

$\square$

## D.6 Proof of the linear convergence analysis

In this section, we derive the linear convergence result of Theorem 4.3 with exponentially increasing step-size.

*Proof.* Starting from Proposition D.6 by dropping $\frac{\mathcal{D}_t^{t+1}}{(1-\gamma)\eta_t}$ on the left hand side, we have

$$\mathbb{E}\left[\nu_\mu\left(\Delta_{t+1} - \Delta_t\right) + \Delta_t\right] \leq \mathbb{E}\left[\frac{\mathcal{D}_t^\star}{(1-\gamma)\eta_t} - \frac{\mathcal{D}_{t+1}^\star}{(1-\gamma)\eta_t}\right] + (1+\nu_\mu)\tau.$$

Dividing $\nu_\mu$ on both side and rearranging, we obtain

$$\mathbb{E}\left[\Delta_{t+1} + \frac{\mathcal{D}_{t+1}^\star}{(1-\gamma)\nu_\mu\eta_t}\right] \leq \left(1 - \frac{1}{\nu_\mu}\right)\mathbb{E}\left[\Delta_t + \frac{\mathcal{D}_t^\star}{(1-\gamma)\eta_t(\nu_\mu-1)}\right] + \left(1 + \frac{1}{\nu_\mu}\right)\tau.$$

If the step-sizes satisfy $\eta_{t+1}(\nu_\mu - 1) \geq \eta_t\nu_\mu$ with $\nu_\mu \geq 1$, then

$$\mathbb{E}\left[\Delta_{t+1} + \frac{\mathcal{D}_{t+1}^\star}{(1-\gamma)\eta_{t+1}(\nu_\mu-1)}\right] \leq \left(1 - \frac{1}{\nu_\mu}\right)\mathbb{E}\left[\Delta_t + \frac{\mathcal{D}_t^\star}{(1-\gamma)\eta_t(\nu_\mu-1)}\right] + \left(1 + \frac{1}{\nu_\mu}\right)\tau.$$

Now we need the following simple fact, whose proof is straightforward and thus omitted.

Suppose $0 < \alpha < 1, b > 0$ and a nonnegative sequence $\{a_t\}_{t\geq 0}$ satisfies

$$a_{t+1} \leq \alpha a_t + b \qquad \forall t \geq 0.$$

Then for all $t \geq 0$,

$$a_t \leq \alpha^t a_0 + \frac{b}{1-\alpha}.$$

The proof of the linear convergence analysis follows by applying this fact with $a_t = \mathbb{E}\left[\Delta_t + \frac{\mathcal{D}_t^\star}{(1-\gamma)\eta_t(\nu_\mu-1)}\right], \alpha = 1 - \frac{1}{\nu_\mu}$ and $b = \left(1 + \frac{1}{\nu_\mu}\right)\tau.$ $\square$

# E    Discussion of the first step (Line 1) of AMPO – the compatible function approximation framework

Starting from this section, some additional remarks about AMPO are in order. In particular, we discuss in detail the novelty of the first step (Line 1) and the second step (Line 2) of AMPO in this and the next section, respectively. Afterwards, we provide an extensive justification of the assumptions used in Theorem 4.3 in Appendices G to I.

As mentioned in Remark 3.3, Agarwal et al. [1] study NPG with smooth policies through compatible function approximation and propose the following algorithm. Let $\{\pi^\theta : \theta \in \Theta\}$ be a policy class such that $\log \pi^\theta(a \mid s)$ is a $\beta$-smooth function of $\theta$ for all $s \in \mathcal{S}$, $a \in \mathcal{A}$. At each iteration $t$, update

$$\theta^{t+1} = \theta^t + \eta w^t,$$

with

$$w^t \in \operatorname*{argmin}_{\|w\|_2 \leq W} \left\| A^t - w^\top \nabla_\theta \log \pi^t \right\|_{L_2(d_\mu^t \cdot \pi^t)}, \tag{61}$$

where $W > 0$ and $A^t(s,a) = Q^t(s,a) - V^t(s)$ represents the advantage function. While both the algorithm proposed by Agarwal et al. [1] and AMPO involve regression problems, the one in (61) is restricted to linearly approximate $A^t$ with $\nabla_\theta \log \pi^t$, whereas the one in Line 1 of Algorithm 1 is relaxed to approximate $A^t$ with an arbitrary class of functions $\mathcal{F}^\Theta$. Additionally, (61) depends on the distribution $d_\mu^t$, while Line 1 of Algorithm 1 does not and allows off-policy updates involving an arbitrary distribution $v^t$, as $v^t$ is independent of the current policy $\pi^t$.

# F    Discussion of the second step (Line 2) of AMPO – the Bregman projection

As mentioned in Remark 3.4, we can rewrite the second step (Line 2) of AMPO through the following lemma.

**Lemma F.1.** *For any policy $\bar{\pi}$, for any function $f^\theta \in \mathcal{F}^\Theta$ and for $\eta > 0$, we have, for all $s \in \mathcal{S}$,*

$$\tilde{\pi}_s \in \operatorname*{argmin}_{p \in \Delta(\mathcal{A})} \mathcal{D}_h(p, \nabla h^*(\eta f_s^\theta)) \iff \tilde{\pi}_s \in \operatorname*{argmin}_{p \in \Delta(\mathcal{A})} \langle -\eta f_s^\theta + \nabla h(\bar{\pi}_s), p \rangle + \mathcal{D}_h(p, \bar{\pi}_s).$$

Equations (12) and (20) are obtained by choosing $\tilde{\pi}_s = \pi_s^{t+1}$ and $\bar{\pi}_s = \pi_s^t$ for all $s \in \mathcal{S}$, $\eta = \eta_t$, $\theta = \theta^{t+1}$, and by changing the sign of the expression on the right in order to obtain an $\operatorname{argmax}$.

*Proof.* Starting from the definition of $\tilde{\pi}$, we have

$$\begin{aligned}
\tilde{\pi}_s &\in \operatorname*{argmin}_{p \in \Delta(\mathcal{A})} \mathcal{D}_h(p, \nabla h^*(\eta f_s^\theta)) \\
&\in \operatorname*{argmin}_{p \in \Delta(\mathcal{A})} h(p) - h(\nabla h^*(\eta f_s^\theta)) - \langle \nabla h(\nabla h^*(\eta f_s^\theta)), p - \nabla h^*(\eta f_s^\theta) \rangle \\
&\in \operatorname*{argmin}_{p \in \Delta(\mathcal{A})} h(p) - \langle \eta f_s^\theta, p \rangle \\
&\in \operatorname*{argmin}_{p \in \Delta(\mathcal{A})} \langle -\eta f_s^\theta + \nabla h(\bar{\pi}_s), p \rangle + h(p) - h(\bar{\pi}_s) - \langle \nabla h(\bar{\pi}_s), p - \bar{\pi}_s \rangle \\
&\in \operatorname*{argmin}_{p \in \Delta(\mathcal{A})} \langle -\eta f_s^\theta + \nabla h(\bar{\pi}_s), p \rangle + \mathcal{D}_h(p, \bar{\pi}_s), \tag{62}
\end{aligned}$$

where the second and the last lines are obtained using the definition of the Bregman divergence, and the third line is obtained using (4) ($\nabla h(\nabla h^*(x^*)) = x^*$ for all $x^* \in \mathbb{R}^{|\mathcal{A}|}$). $\qquad\square$

Lemmas F.1 and B.1 share a similar result, as they both rewrite the Bregman projection into the MD updates. However, the MD updates in Lemma B.1 are exact, while the MD updates in AMPO involve approximation (Line 1).

Next, we provide an alternative proof for Lemma 4.1 to show that it is the direct consequence of Lemma F.1. The proof will involve the application of the three-point descent lemma [19, Lemma 3.2]. Here we adopt its slight variation by following Lemma 6 in Xiao [93].

**Lemma F.2** (Three-point decent lemma, Lemma 6 in Xiao [93])**.** *Suppose that $\mathcal{C} \subset \mathbb{R}^m$ is a closed convex set, $f : \mathcal{C} \to \mathbb{R}$ is a proper, closed [6] convex function, $\mathcal{D}_h(\cdot, \cdot)$ is the Bregman divergence generated by a mirror map $h$. Denote $\mathrm{rint}\,\mathrm{dom}\,h$ as the relative interior of $\mathrm{dom}\,h$. For any $x \in \mathrm{rint}\,\mathrm{dom}\,h$, let*

$$x^+ \in \arg \min_{u \in \mathrm{dom}\,h \cap \mathcal{C}} \{f(u) + \mathcal{D}_h(u, x)\}.$$

*Then $x^+ \in \mathrm{rint}\,\mathrm{dom}\,h \cap \mathcal{C}$ and for any $u \in \mathrm{dom}\,h \cap \mathcal{C}$,*

$$f(x^+) + \mathcal{D}_h(x^+, x) \le f(u) + \mathcal{D}_h(u, x) - \mathcal{D}_h(u, x^+).$$

We refer to Yuan et al. [98, Lemma 11] for a proof of Lemma F.2.

Lemma 4.1 is obtained by simply applying the three-point descent lemma, Lemma F.2, to (62) with $x^+ = \tilde{\pi}_s$, $f(u) = \langle -\eta f_s^\theta + \nabla h(\bar{\pi}_s), u \rangle$, $u = \pi$ and $x = \bar{\pi}_s$ and rearranging terms.

In contrast, it may not be possible to apply Lemma F.2 to (11), as $\Pi(\Theta)$ is often non-convex.

# G  Discussion on Assumption (A1) – the approximation error

The compatible function approximation approach [1, 61, 15, 21, 2, 98] has been introduced to deal with large state and action spaces, in order to reduce the dimension of the problem and make the computation feasible. As mentioned in the *proof idea* in Page 8, this framework consists in upper-bounding the sub-optimality gap with an optimization error plus an approximation error. Consequently, it is important that both error terms converge to 0 in order to achieve convergence to a global optimum.

Assumptions similar to Assumption (A1) are common in the compatible function approximation literature. Assumption (A1) encodes a form of realizability assumption for the parameterization class $\mathcal{F}^\Theta$, that is, we assume that for all $t \le T$ there exists a function $f^\theta \in \mathcal{F}^\Theta$ such that

$$\left\| f^\theta - Q^t - \eta_t^{-1} \nabla h(\pi^t) \right\|_{L_2(v^t)}^2 \le \varepsilon_{\mathrm{approx}}.$$

When $\mathcal{F}^\Theta$ is a class of sufficiently large shallow neural networks, this realizability assumption holds as it has been shown that shallow neural networks are universal approximators [39]. It is, however, possible to relax Assumption (A1). In particular, the condition

$$\frac{1}{T} \sum_{t<T} \sqrt{\mathbb{E}\left[ \mathbb{E} \left\| f^{t+1} - Q^t - \eta_t^{-1} \nabla h(\pi^t) \right\|_{L_2(v^t)}^2 \right]} \le \sqrt{\varepsilon_{\mathrm{approx}}} \tag{63}$$

can replace Assumption (A1) and is sufficient for the sublinear convergence rate in Theorem 4.3 to hold. Equation (63) shows that the realizability assumption does not need to hold for all $t < T$, but only needs to hold on average over $T$ iterations. Similarly, the condition

$$\sum_{t \le T} \left(1 - \frac{1}{\nu_\mu}\right)^{T-t} \frac{1}{\nu_\mu} \sqrt{\mathbb{E}\left[ \mathbb{E} \left\| f^{t+1} - Q^t - \eta_t^{-1} \nabla h(\pi^t) \right\|_{L_2(v^t)}^2 \right]} \le \sqrt{\varepsilon_{\mathrm{approx}}} \tag{64}$$

can replace Assumption (A1) and is sufficient for the linear convergence rate in Theorem 4.3 to hold. Additionally, requiring, for all $t < T$,

$$\mathbb{E}\left[ \mathbb{E} \left\| f^{t+1} - Q^t - \eta_t^{-1} \nabla h(\pi^t) \right\|_{L_2(v^t)}^2 \right] \le \frac{\nu_\mu^2}{T^2} \left(1 - \frac{1}{\nu_\mu}\right)^{-2(T-t)} \varepsilon_{\mathrm{approx}} \tag{65}$$

is sufficient for Equation (64) to hold. Equation (65) shows that the error floor in the linear convergence rate is less influenced by approximation errors made in early iterations, which are discounted by the term $\left(1 - \frac{1}{\nu_\mu}\right)$. On the other hand, the realizability assumption becomes relevant once the algorithm approaches convergence, i.e., when $t \simeq T$ and $Q^t \simeq Q^\star$, as the discount term $\left(1 - \frac{1}{\nu_\mu}\right)$ is applied fewer times.

Finally, although Assumption (A1) holds for the softmax tabular policies and for the neural network parameterization, it remains an open question whether Assumption (A1) is necessary to achieve the global optimum convergence, especially when the representation power of $\mathcal{F}^\Theta$ cannot guarantee a small approximation error.

---

[6]A convex function $f$ is proper if $\mathrm{dom}\,f$ is nonempty and for all $x \in \mathrm{dom}\,f$, $f(x) > -\infty$. A convex function is closed, if it is lower semi-continuous.

# H   Discussion on Assumption (A2) – the concentrability coefficients

In our convergence analysis, Assumptions (A2) and (A3) involve the concentrability coefficient $C_v$ and the distribution mismatch coefficient $\nu_\mu$, which are potentially large. We give extensive discussions on them in this and the next section, respectively.

As discussed in Yuan et al. [98, Appendix H], the issue of having (potentially large) concentrability coefficient (Assumptions (A2)) is unavoidable in all the fast linear convergence analysis of approximate PMD due to the approximation error $\varepsilon_{\text{approx}}$ of the $Q$-function [17, 100, 54, 16, 93, 20, 2, 98]. Indeed, in the fast linear convergence analysis of PMD, the concentrability coefficient is always along with the approximation error $\varepsilon_{\text{approx}}$ under the form of $C_v\varepsilon_{\text{approx}}$, which is the case in Theorem 4.3. To not get the concentrability coefficient involved yet maintain the linear convergence of PMD, one needs to consider the exact PMD in the tabular setting [see 93, Theorem 10]. Consequently, the PMD update is deterministic and the full policy space $\Delta(\mathcal{A})^{\mathcal{S}}$ is considered. In this setting, at each time $t$, it exists $\theta^{t+1}$ such that, for any state-action distribution $v^t$,

$$\left\| f^{t+1} - Q^t - \eta_t^{-1}\nabla h(\pi^t) \right\|_{L_2(v^t)}^2 = 0 = \varepsilon_{\text{approx}},$$

and $C_v$ is ignored in the convergence analysis thanks to the vanishing of $\varepsilon_{\text{approx}}$. We note that the PMD analysis in the seminal paper by Agarwal et al. [1] does not use such a coefficient, but a condition number instead. The condition number is controllable to be relatively small, so that the error term in their PMD analysis is smaller than ours. However, their PMD analysis has only a sublinear convergence rate, while ours enjoys a fast linear convergence rate. It remains an open question whether one can both avoid using the concentrability coefficient and maintain the linear convergence of PMD.

Now we compare our concentrability coefficient $C_v$ with others used in the fast linear convergence analysis of approximate PMD [17, 100, 54, 15, 93, 20, 2, 98]. To the best of our knowledge, the previously best-known concentrability coefficient $C_v$ was the one used by Yuan et al. [98, Appendix H]. As they discuss, their concentrability coefficient involved the weakest assumptions on errors among Lan [54], Xiao [93] and Chen and Theja Maguluri [20] by using the $L_2$-norm instead of the $\ell_\infty$-norm over the approximation error $\varepsilon_{\text{approx}}$. Additionally, it did not impose any restrictions on the MDP dynamics compared to Cayci et al. [15], as the concentrability coefficient of Yuan et al. [98] was independent from the iterates.

Indeed, Yuan et al. [98] choose $v^t$ such that, for all $(s,a) \in \mathcal{S} \times \mathcal{A}$,

$$v^t(s,a) = (1-\gamma)\,\mathbb{E}_{(s_0,a_0)\sim\nu}\left[\sum_{t'=0}^{\infty}\gamma^{t'}P(s_{t'}=s, a_{t'}=a \mid \pi^t, s_0, a_0)\right],$$

where $\nu$ is an initial state-action distribution chosen by the user. In this setting, we have

$$v^t(s,a) \geq (1-\gamma)\nu(s,a).$$

From the above lower bound of $v^t$, we obtain that

$$\mathbb{E}_{(s,a)\sim v^t}\left[\left(\frac{d_\mu^\pi(s)\pi(a\mid s)}{v^t(s,a)}\right)^2\right] = \int_{(s,a)\in\mathcal{S}\times\mathcal{A}}\frac{d_\mu^\pi(s)^2\pi(a\mid s)^2}{v^t(s,a)}$$

$$\leq \int_{(s,a)\in\mathcal{S}\times\mathcal{A}}\frac{1}{v^t(s,a)} \leq \frac{1}{(1-\gamma)\min_{(s,a)\in\mathcal{S}\times\mathcal{A}}\nu(s,a)},$$

where the finite upper bound is independent to $t$.

As mentioned right after Assumption (A2), the assumption on our concentrability coefficient $C_v$ is weaker than the one in Yuan et al. [98, Assumption 9], as we have the full control over $v^t$ while Yuan et al. [98] only has the full control over the initial state-action distribution $\nu$. In particular, our concentrability coefficient $C_v$ recovers the previous best-known one in Yuan et al. [98] as a special case. Consequently, our concentrability coefficient $C_v$ becomes the "best" with the full control over $v^t$ when other concentrability coefficients are infinite or require strong assumptions [77].

In general, for the ratio $\mathbb{E}_{(s,a)\sim v^t}\left[\left(\frac{d_\mu^\pi(s)\pi(a|s)}{v^t(s,a)}\right)^2\right]$ to have a finite upper bound $C_v$, it is important that $v^t$ covers well the state and action spaces so that the upper bound is independent to $t$. However, the

upper bound $\frac{1}{(1-\gamma)\min_{(s,a)\in\mathcal{S}\times\mathcal{A}}\nu(s,a)}$ in Yuan et al. [98] is very pessimistic. Indeed, when $\pi^t$ and $\pi^{t+1}$ converge to $\pi^\star$, one reasonable choice of $v^t$ is to choose $v^t \in \{d_\mu^\star \cdot \pi^\star, d_\mu^{t+1} \cdot \pi^{t+1}, d_\mu^\star \cdot \pi^t, d_\mu^{t+1} \cdot \pi^t\}$ such that $C_v$ is close to 1.

We also refer to Yuan et al. [98, Appendix H] for more discussions on the concentrability coefficient.

## I  Discussion on Assumption (A3) – the distribution mismatch coefficients

In this section, we give further insights on the distribution mismatch coefficient $\nu_\mu$ in Assumption (A3). As mentioned right after (A3), we have that

$$\sup_{s\in\mathcal{S}} \frac{d_\mu^\star(s)}{d_\mu^t(s)} \leq \frac{1}{1-\gamma} \sup_{s\in\mathcal{S}} \frac{d_\mu^\star(s)}{\mu(s)} := \nu_\mu',$$

which is a sufficient upper bound for $\nu_\mu$. As discussed in Yuan et al. [98, Appendix H],

$$1/(1-\gamma) \leq \nu_\mu' \leq 1/((1-\gamma)\min_s \mu(s)).$$

The upper bound $1/((1-\gamma)\min_s \mu(s))$ of $\nu_\mu'$ is very pessimistic and the lower bound $\nu_\mu' = 1/(1-\gamma)$ is often achieved by choosing $\mu = d_\mu^\star$.

Furthermore, if $\mu$ does not have full support on the state space, i.e., the upper bound $1/((1-\gamma)\min_s \mu(s))$ might be infinite, one can always convert the convergence guarantees for some state distribution $\mu' \in \Delta(\mathcal{S})$ with full support such that

$$V^\star(\mu) - \mathbb{E}[V^T(\mu)] = \mathbb{E}\left[\int_{s\in\mathcal{S}} \frac{\mu(s)}{\mu'(s)}\mu'(s)\left(V^\star(s) - V^T(s)\right)\right]$$

$$\leq \sup_{s\in\mathcal{S}} \frac{\mu(s)}{\mu'(s)}\left(V^\star(\mu') - \mathbb{E}[V^T(\mu')]\right).$$

Then by the linear convergence result of Theorem 4.3, we only transfer the original convergence guarantee to $V^\star(\mu') - \mathbb{E}[V^T(\mu')]$ up to a scaling factor $\sup_{s\in\mathcal{S}} \frac{\mu(s)}{\mu'(s)}$ with an arbitrary distribution $\mu'$ such that $\nu_\mu'$ is finite.

Finally, if $d_\mu^t$ converges to $d_\mu^\star$ which is the case of AMPO through the proof of our Theorem 4.3, then $\sup_{s\in\mathcal{S}} \frac{d_\mu^\star(s)}{d_\mu^t(s)}$ converges to 1. This might imply superlinear convergence results as discussed in Xiao [93, Section 4.3]. In this case, the notion of the distribution mismatch coefficients $\nu_\mu$ no longer exists for the superlinear convergence analysis.

We also refer to Yuan et al. [98, Appendix H] for more discussions on the distribution mismatch coefficient.

## J  Sample complexity for neural network parameterization

We prove here Corollary 4.5 through a result by Allen-Zhu et al. [3, Theorem 1 and Example 3.1]. We first give a simplified version of this result and then we show how to use it to prove Corollary 4.5.

Consider learning some unknown distribution $\mathcal{D}$ of data points $z = (x, y) \in \mathbb{R}^d \times \mathcal{Y}$, where $x$ is the input point and $y$ is the label. Without loss of generality, assume $\|x\|_2 = 1$ and $x_d = 1/2$. Consider a loss function $L : \mathbb{R}^k \times \mathcal{Y} \to \mathbb{R}$ such that for every $y \in \mathcal{Y}$, the function $L(\cdot, y)$ is non-negative, convex, 1-Lipschitz continuous and $L(0, y) \in [0, 1]$. This includes both the cross-entropy loss and the $L_2$-regression loss (for bounded $\mathcal{Y}$).

Let $g : \mathbb{R} \to \mathbb{R}$ be a smooth activation function such that $g(z) = e^z, \sin(z), \texttt{sigmoid}(z), \tanh(z)$ or is a low degree polynomial.

Define $F^\star : \mathbb{R}^d \to \mathbb{R}^k$ such that $OPT = \mathbb{E}_{\mathcal{D}}[L(F^\star(x), y)]$ is the smallest population error made by a neural network of the form $F^\star = A^\star g(W^\star x)$, where $A^\star \in \mathbb{R}^{k\times p}$ and $W^\star \in \mathbb{R}^{p\times d}$. Assume for simplicity that the rows of $W^*$ have $\ell_2$-norm 1 and each element of $A^*$ is less or equal than 1.

Define a ReLU neural network $F(x, W_0) = A_0\sigma(W_0 x + b_0)$, where $A_0 \in \mathbb{R}^{k \times m}$, $W_0 \in \mathbb{R}^{m \times d}$, the entries of $W_0$ and $b_0$ are i.i.d. random Gaussians from $\mathcal{N}(0, 1/m)$ and the entries of $A$ are i.i.d. random Gaussians from $\mathcal{N}(0, \varepsilon_A)$, for $\varepsilon_A \in (0, 1]$. We train the weights $W$ of this neural network through stochastic gradient descent over a dataset with $N$ i.i.d. samples from $\mathcal{D}$, i.e., we update $W_{t+1} = W_t - \eta g_t$, where $\mathbb{E}[g_t] = \nabla \mathbb{E}_{\mathcal{D}}[L(F(x, W_0 + W_t), y)]$.

**Theorem J.1** (Theorem 1 of Allen-Zhu et al. [3]). *Let $\varepsilon \in (0, O(1/pk))$, choose $\varepsilon_A = \varepsilon/\widetilde{\Theta}(1)$ for the initialization and learning rate $\eta = \widetilde{\Theta}\left(\frac{1}{\varepsilon k m}\right)$. SGD finds a set of parameters such that*

$$\frac{1}{J} \sum_{n=0}^{J-1} \mathbb{E}_{(x,y) \sim \mathcal{D}}\left[L\left(F(x; W^{(0)} + W_t), y\right)\right] \leq OPT + \varepsilon$$

*with probability $1 - e^{-c\log^2 m}$ over the random initialization, for a sufficiently large constant c, with*

$$size\ m = \frac{\texttt{poly}(k,p)}{\texttt{poly}(\varepsilon)} \ \ and\ sample\ complexity\ \ \min\{N, J\} = \frac{\texttt{poly}(k, p, \log m)}{\varepsilon^2}.$$

Theorem J.1 shows that it is possible to achieve the population error OPT by training a two-layer ReLU network with SGD, and quantifies the number of samples needed to do so.

We make the following assumption to address the population error in our setting.

**Assumption J.2.** Let $g : \mathbb{R} \to \mathbb{R}$ be a smooth activation function such that $g(z) = e^z$, $\sin(z)$, $\texttt{sigmoid}(z)$, $\tanh(z)$ or is a low degree polynomial. For all time-steps $t$, we assume that there exists a target network $F^{\star,t} : \mathbb{R}^d \to \mathbb{R}^k$, with

$$F^{\star,t} = (f_1^{\star,t}, \ldots, f_k^{\star,t}) \quad \text{and} \quad f_r^{\star,t}(x) = \sum_{i=1}^{p} a_{r,i}^{\star,t} g(\langle w_{1,i}^{\star,t}, x \rangle) \langle w_{2,i}^{\star,t}, x \rangle$$

where $w_{1,i}^{\star,t} \in \mathbb{R}^d$, $w_{2,i}^{\star,t} \in \mathbb{R}^d$, and $a_{r,i}^{\star,t} \in \mathbb{R}$, such that

$$\mathbb{E}\big[\, \big\|F^{\star,t} - Q^t - \eta_t^{-1}\nabla h(\pi^t)\big\|_{L_2(v^t)}^2\,\big] \leq OPT.$$

We assume for simplicity $\|w_{1,i}^{\star,t}\|_2 = \|w_{2,i}^{\star,t}\|_2 = 1$ and $|a_{r,i}^{\star,t}| \leq 1$.

Assumptions similar to Assumption J.2 have already been made in the literature, such as the bias assumption in the compatible function approximation framework studied by [1]. The term $OPT$ represents the minimum error incurred by a target network parameterized as $F^{\star,t}$ when solving the regression problem in Line 1 of Algorithm 1.

We are now ready to prove Corollary 4.5, which uses Algorithm 4 to obtain an unbiased estimate of the current Q-function. We assume to be in the same setting as Theorem J.1

*Proof of Corollary 4.5.* We aim to find a policy $\pi^T$ such that

$$V^\star(\mu) - \mathbb{E}\left[V^T(\mu)\right] \leq \varepsilon. \tag{66}$$

Suppose the total number of iterations, that is policy optimization steps, in AMPO is $T$. We need the bound in Assumption (A1) to hold for all $T$ with probability $1 - e^{-c\log^2 m}$, which means that at each iteration the bound should hold with probability $1 - T^{-1}e^{-c\log^2 m}$. Through Algorithm 4, the expected number of samples needed to obtain an unbiased estimate of the current Q-function is $(1-\gamma)^{-1}$. Therefore, using Theorem J.1, at each iteration of AMPO we need at most

$$\frac{\texttt{poly}(k, p, \log m, \log T)}{\varepsilon_{\text{approx}}^2(1-\gamma)}$$

samples for SGD to find parameters that satisfy Assumption (A1) with probability $1 - T^{-1}e^{-c\log^2 m}$. To obtain (66), we need

$$\frac{1}{1-\gamma}\left(1 - \frac{1}{\nu_\mu}\right)^T \left(1 + \frac{\mathcal{D}_0^\star}{\eta_0(\nu_\mu - 1)}\right) \leq \frac{\varepsilon}{2} \quad \text{and} \quad \frac{2(1 + \nu_\mu)\sqrt{C_v \varepsilon_{\text{approx}}}}{1-\gamma} \leq \frac{\varepsilon}{2}. \tag{67}$$

**Algorithm 4:** Sampler for an unbiased estimate $\widehat{Q^t}(s,a)$ of $Q^t(s,a)$

---

**Input:** Initial state-action couple $(s_0, a_0)$, policy $\pi^t$, discount factor $\gamma \in [0, 1)$

1   Initialize $\widehat{Q^t}(s_0, a_0) = r(s_0, a_0)$, the time step $n = 0$.
2   **while** *True* **do**
3      **With probability** $\gamma$**:**
4         Sample $s_{n+1} \sim P(\cdot \mid s_n, a_n)$
5         Sample $a_{n+1} \sim \pi^t(\cdot | s_{n+1})$
6         $\widehat{Q^t}(s_0, a_0) \leftarrow \widehat{Q^t}(s_0, a_0) + r(s_{n+1}, a_{n+1})$
7         $n \leftarrow n + 1$
8      **Otherwise with probability** $(1 - \gamma)$**:**
9         **break**                               ▷ Accept $\widehat{Q}_{s_h, a_h}(\theta)$

**Output:** $\widehat{Q^t}(s_0, a_0)$

---

Solving for $T$ and $\varepsilon_{\mathrm{approx}}$ and multiplying them together, we obtain the sample complexity of AMPO, that is

$$\widetilde{\mathcal{O}}\left( \frac{\mathrm{poly}(k, p, \log m) C_v^2 \nu_\mu^5}{\varepsilon^4 (1-\gamma)^6} \right).$$

Due to the statement of Theorem J.1, we cannot guarantee the approximation error incurred by the learner network to be smaller than $OPT$. Consequently, we have that

$$\varepsilon \geq \frac{4(1 + \nu_\mu)\sqrt{C_v OPT}}{1 - \gamma}.$$

A similar bound can be applied to any proof that contains the bias assumption introduced by [1].   □

We can obtain an improvement over Corollary 4.5 by using the relaxed assumptions in Appendix G, in particular using the condition in (65).

**Corollary J.3.** *In the setting of Theorem 4.3, replace Assumption* (A1) *with the condition*

$$\mathbb{E}\left[ \mathbb{E}\left\| f^{t+1} - Q^t - \eta_t^{-1} \nabla h(\pi^t) \right\|_{L_2(v^t)}^2 \right] \leq \frac{\nu_\mu^2}{T^2} \left( 1 - \frac{1}{\nu_\mu} \right)^{-2(T-t)} \varepsilon_{\mathrm{approx}}, \tag{68}$$

*for all $t < T$. Let the parameterization class $\mathcal{F}^\Theta$ consist of sufficiently wide shallow ReLU neural networks. Using an exponentially increasing step-size and solving the minimization problem in Line 1 with SGD as in* (19)*, the number of samples required by AMPO to find an $\varepsilon$-optimal policy with high probability is $\widetilde{\Theta}(C_v^2 \nu_\mu^4 / \varepsilon^4 (1-\gamma)^6)$.*

*Proof.* The proof follow that of Corollary 4.5. Using Theorem J.1, at each iteration $t$ of AMPO, we need at most

$$\frac{T^2}{\nu_\mu^2} \left( 1 - \frac{1}{\nu_\mu} \right)^{2(T-t)} \frac{\mathrm{poly}(k, p, \log m, \log T)}{\varepsilon_{\mathrm{approx}}^2 (1-\gamma)}$$

samples for SGD to find parameters that satisfy condition (68) with probability $1 - T^{-1}e^{-c\log^2 m}$. Summing over $T$ total iterations of AMPO we obtain that the total number of samples needed is

$$\sum_{t \leq T} \frac{T^2}{\nu_\mu^2} \left(1 - \frac{1}{\nu_\mu}\right)^{2(T-t)} \frac{\texttt{poly}(k, p, \log m, \log T)}{\varepsilon_{\text{approx}}^2(1-\gamma)}$$

$$= \frac{T^2}{\nu_\mu^2} \left(1 - \frac{1}{\nu_\mu}\right)^{2T} \frac{\texttt{poly}(k, p, \log m, \log T)}{\varepsilon_{\text{approx}}^2(1-\gamma)} \sum_{t \leq T} \left(1 - \frac{1}{\nu_\mu}\right)^{-2t}$$

$$= \frac{T^2}{\nu_\mu^2} \left(1 - \frac{1}{\nu_\mu}\right)^{2T} \frac{\texttt{poly}(k, p, \log m, \log T)}{\varepsilon_{\text{approx}}^2(1-\gamma)} \frac{\left(\left(1 - \frac{1}{\nu_\mu}\right)^{-2(T+1)} - 1\right)}{\left(\left(\frac{1}{1-\nu_\mu}\right)^{-2} - 1\right)}$$

$$\leq \mathcal{O}\left(\frac{T^2}{\nu_\mu^2} \frac{\texttt{poly}(k, p, \log m, \log T)}{\varepsilon_{\text{approx}}^2(1-\gamma)}\right)$$

Replacing $T$ and $\varepsilon_{\text{approx}}$ with the solutions of (67) gives the result. $\qquad\square$

At this stage, it is important to note that choosing a method different from the one proposed by Allen-Zhu et al. [4] to solve Line 1 in Algorithm 1 of our paper with neural networks can lead to alternative, and possibly better, sample complexity results for AMPO. For example, we can obtain a sample complexity result for AMPO that does not involve a target network using results from [39] and [16], although this requires introducing more notation and background results compared to Corollary 4.5 (since in [16] they employ a temporal-difference-based algorithm, that is Algorithm 3 in their work, to obtain a neural network estimate $\widehat{Q}^t$ of $Q^t$, while in [39] they provide a method based on Fourier transforms to approximate a target function through shallow ReLU networks). We outline below the steps in order to do so (and additional details including the precise statements of the results we use and how we use them are provided thereafter for the sake of completeness).

**Step 1)** We first split the approximation error in Assumption (A1) into a critic error $\mathbb{E}[\sqrt{\|\widehat{Q}^t - Q^t\|_{L_2(v^t)}^2}] \leq \varepsilon_{\text{critic}}$ and an actor error $\mathbb{E}[\sqrt{\|f^{t+1} - \widehat{Q}^t - \eta_t^{-1}\nabla h(\pi^t)\|_{L_2(v^t)}^2}] \leq \varepsilon_{\text{actor}}$. In this case, the linear convergence rate in our Theorem 4.3 becomes

$$V^\star(\mu) - \mathbb{E}\left[V^T(\mu)\right] \leq \frac{1}{1-\gamma}\left(1 - \frac{1}{\nu_\mu}\right)^T \left(1 + \frac{\mathcal{D}_0^\star}{\eta_0(\nu_\mu - 1)}\right) + \frac{2(1+\nu_\mu)\sqrt{C_v}(\varepsilon_{\text{critic}} + \varepsilon_{\text{actor}})}{1-\gamma}.$$

[We can obtain this alternative statement by modifying the passages in Appendix D.2. In particular, writing $f^{t+1} - Q^t - \eta_t^{-1}\nabla h(\pi^t) = (f^{t+1} - \widehat{Q}^t - \eta_t^{-1}\nabla h(\pi^t)) + (Q^t - \widehat{Q}^t)$ and bounding the two terms with the same procedure in Appendix D.2 leads to this alternative expression for the error.]

We will next deal with the critic error and actor error separately.

**Step 2)** Critic error. Under a realizability assumption that we provide below along with the statement of the theorem (Assumption 2 in [16]), Theorem 1 from [16] gives that the sample complexity required to obtain $\mathbb{E}[\sqrt{\|\widehat{Q}^t - Q^t\|_{L_2(d_\mu^t \cdot \pi^t)}^2}] \leq \varepsilon$ is $\widetilde{O}(\varepsilon^{-4}(1-\gamma)^{-2})$, while the required network width is $\widetilde{O}(\varepsilon^{-2})$.

**Step 3)** Actor error. Using Theorem E.1 from [39], we obtain that $\mathbb{E}[\sqrt{\|f^{t+1} - \widehat{Q}^t - \eta_t^{-1}\nabla h(\pi^t)\|_{L_2(v^t)}^2}]$ can be made arbitrarily small by tuning the width of $f^{t+1}$, without using further samples.

**Step 4)** Replacing Equation (67) with the sample complexity of the critic, we obtain the following corollary on the sample complexity of AMPO, which does not depend on the error made by a target network.

**Corollary J.4.** *In the setting of Theorem 4.3, let the parameterization class $\mathcal{F}^\Theta$ consist of sufficiently wide shallow ReLU neural networks. Using an exponentially increasing step-size and using the techniques above to update $f^\theta$, the number of samples required by AMPO to find an $\varepsilon$-optimal policy with high probability is $\widetilde{\mathcal{O}}(C_v^2 \nu_\mu^5 / \varepsilon^4 (1-\gamma)^7)$.*

To the best of our knowledge, this result improves upon the previous best result on the sample complexity of a PG method with neural network parameterization [16], i.e., $\widetilde{\mathcal{O}}(C_v^2/\varepsilon^6(1-\gamma)^9)$.

We now provide the statements of the aforementioned results we used.

**Recalling Theorem 1 in [16] and its assumptions.** Consider the following space of mappings:

$$\mathcal{H}_{\bar{\nu}} = \{v : \mathbb{R}^d \to \mathbb{R}^d : \sup_{w \in \mathbb{R}^d} \|v(w)\|_2 \leq \bar{\nu}\},$$

and the function class:

$$\mathcal{F}_{\bar{\nu}} = \left\{ g(\cdot) = \mathbb{E}_{w_0 \sim \mathcal{N}(0, I_d)}[\langle v(w_0), \cdot \rangle \mathbb{I}\{\langle w_0, \cdot \rangle > 0\}] : v \in \mathcal{H}_{\bar{\nu}} \right\}.$$

Consider the following realizability assumption for the Q-function.

**Assumption J.5** (Assumption 2 in [16]). For any $t \geq 0$, we assume that $Q^t \in \mathcal{F}_{\bar{\nu}}$ for some $\bar{\nu} > 0$.

**Theorem J.6** (Theorem 1 in [16]). *Under Assumption 2 in [16], for any error probability $\delta \in (0, 1)$, let*

$$\ell(m', \delta) = 4\sqrt{\log(2m' + 1)} + 4\sqrt{\log(T/\delta)},$$

*and $R > \bar{\nu}$. Then, for any target error $\varepsilon > 0$, number of iterations $T' \in \mathbb{N}$, network width*

$$m' > \frac{16\left(\bar{\nu} + \left(R + \ell(m', \delta)\right)\left(\bar{\nu} + R\right)\right)^2}{(1-\gamma)^2 \varepsilon^2},$$

*and step-size*

$$\alpha_C = \frac{\varepsilon^2 (1 - \gamma)}{(1 + 2R)^2},$$

*Algorithm 3 in [16] yields the following bound:*

$$\mathbb{E}\left[\sqrt{\|\widehat{Q}^t - Q^t\|_{L_2(d_\mu^t \cdot \pi^t)}^2} \mathbb{I}_{A_2}\right] \leq \frac{(1 + 2R)\bar{\nu}}{\varepsilon(1 - \gamma)\sqrt{T'}} + 3\varepsilon,$$

*where $A_2$ holds with probability at least $1 - \delta$ over the random initializations of the critic network $\widehat{Q}^t$.*

As indicated in [16], a consequence of this result is that in order to achieve a target error less than $\varepsilon > 0$, a network width of $m' = \widetilde{O}\left(\frac{\bar{\nu}^4}{\varepsilon^2}\right)$ and iteration complexity $O\left(\frac{(1+2\bar{\nu})^2 \bar{\nu}^2}{(1-\gamma)^2 \varepsilon^4}\right)$ suffice.

The statement of Theorem 1 in [16] can be readily applied to obtain the sample complexity of the critic.

**Recalling Theorem E.1 in [39] and its assumptions** Let $g : \mathbb{R}^n \to \mathbb{R}$ be given and define the modulus of continuity $\omega_g$ as

$$\omega_g(\delta) := \sup_{x, x' \in \mathbb{R}^n} \{g(x) - g(x') : \max(\|x\|_2, \|x'\|_2) \leq 1 + \delta, \|x - x'\|_2 \leq \delta\}.$$

If $g$ is continuous, then $\omega_g$ is not only finite for all inputs, but moreover $\lim_{\delta \to 0} \omega_g(\delta) \to 0$.

Denote $\|p\|_{L_1} = \int |p(w)| dw$. Define a sample from a signed density $p : \mathbb{R}^{n+1} \to \mathbb{R}$ with $\|p\|_{L_1} < \infty$ as $(w, b, s)$, where $(w, b) \in \mathbb{R}$ is sampled from the probability density $|p|/\|p\|_{L_1}$ and $s = sign(p(w, b))$

**Theorem J.7** (Theorem E.1 in [39]). *Let $g : \mathbb{R}^n \to \mathbb{R}$, $\delta > 0$ and $\omega_g(\delta)$ be as above and define for $x \in \mathbb{R}^n$*

$$M := \sup_{\|x\| \leq 1 + \delta} |g(x)|, \qquad g_{|\delta}(x) = f(x) \mathbb{I}[\|x\| \leq 1 + \delta], \qquad \alpha := \frac{\delta}{\sqrt{\delta} + \sqrt{2 \log(2M/\omega_g(\delta))}}.$$

*Let $G_\alpha$ be a gaussian distribution on $\mathbb{R}^n$ with mean $0$ and variance $\alpha^2 \mathcal{I}$. Define the Gaussian convolution $l = g_{|\delta} * G_\alpha$ with Fourier transform $\widehat{l}$ satisfying radial decomposition $\widehat{l}(w) = |\widehat{l}(w)| \exp(2\pi i \theta_h(w))$. Let $P$ be a probability distribution supported on $\|x\| \leq 1$. Additionally define*

$$c := g(0)g(0) \int |\widehat{l}(w)| \big[ \cos(2\pi(\theta_l(w) - \|w\|_2)) - 2\pi\|w\|_2 \sin(2\pi(\theta_l(w) - \|w\|_2)) \big] dw$$

$$a = \int w|\widehat{l}(w)| dw$$

$$r = \sqrt{n} + 2\sqrt{\log \frac{24\pi^2(\sqrt{d}+7)^2\|g_{|\delta}\|_{L_1}}{\omega_g(\delta)}}$$

$$p := 4\pi^2|\widehat{l}(w)| \cos(2\pi(\|w\|_2 - b))\mathbb{I}[|b| \le \|w\| \le r],$$

*and for convenience create fake (weight, bias, sign) triples*

$$(w, b, s)_{m+1} := (0, c, m\,sign(c)), \quad (w, b, s)_{m+2} := (a, 0, m), \quad (w, b, s)_{m+3} := (-a, 0, -m).$$

*Then*

$$|c| \le M + 2\sqrt{n}\|g_{|\delta}\|_{L_1}(2\pi\alpha^2)^{-d/2},$$

$$\|p\|_{L_1} \le 2\|g_{|\delta}\|_{L_1}\sqrt{\frac{(2\pi)^3 n}{(2\pi\alpha^2)^{n+1}}},$$

*and with probability at least $1 - 3\lambda$ over a draw of $((s_j, w_j, b_j))_{j=1}^m$ from $p$*

$$\sqrt{\left\|g - \frac{1}{m}\sum_{j=1}^{m+3} s_j\sigma(\langle w_j, x\rangle + b_j)\right\|_{L(P)}} \le 3\omega_g(\delta) + \frac{r\|p\|_{L_1}}{\sqrt{m}}\left[1 + \sqrt{2\log(1/\lambda)}\right].$$

We can then characterize the error of the actor by choosing $x = (s, a)$, $g = \widehat{Q}^t + \eta_t^{-1}\nabla h(\pi^t)$, and $f^{t+1} = \frac{1}{m}\sum_{j=1}^{m+3} s_j\sigma(\langle w_j, x\rangle + b_j)$. We can then make the actor error arbitrarily small by tuning the network width $m$ and $\delta$ (note that, since both $\widehat{Q}^t$ and $f^t$ are continuous neural networks, $g$ is a continuous function).

