# OpenReview forum: "A Novel Framework for Policy Mirror Descent with General Parameterization and Linear Convergence"
_NeurIPS.cc/2023/Conference — NeurIPS 2023 poster_

### Official Review · Reviewer_WBF5 · 2023-07-03

**Soundness:** 2 fair
**Presentation:** 3 good
**Contribution:** 2 fair
**Rating:** 6
**Confidence:** 4

**Summary:**

This paper proposes a policy update scheme based on mirror descent for general function approximation. The paper also provides corresponding theoretical analysis on the sub-optimality gap and computation cost. Overall, the reviewer find the theory part not strong enough to prove the advantage of the proposed new algorithm or contribute any new insight to the community.

**Strengths:**

-

**Weaknesses:**

1. The main results in Theorem 4.3 is not really a meaningful upper bound of the sub-optimality gap of the learned policy since, for a general function approximation class, we do not know how large the term involving $\epsilon_{approx}$, $v_\mu$, and $C_v$ is. More detailed characterization of their scales (how fast do they converge to zero) is necessary to make the current analysis meaningful.
2. It is weird to make assumptions on $\pi^t$ in A1, A2, A3 when you are exactly analyzing the convergence of $\{\pi^t\}$, since we do not know where the policy iterate $\pi^t$ will go to. A more natural way to do this is to assume the conditions hold for all policies in the policy class. (But then the authors need to be more cautious about the assumptions.)
3. No new insights found in the current framework. The main techniques (mirror decent for policy optimization) are already widely used in previous reinforcement learning theory studies. The proposed algorithm seems another form or a very close variant of proximal policy gradient (PPO).

**Questions:**

-

**Limitations:**

-

---

> ### Author Rebuttal · Authors · 2023-08-09
>
> Please find below an answer to the issues you have raised regarding the significance and soundness of our work. Due to space constraints, please find the bibliography for this reply in the reply to Reviewer qVps.
>
> **Characterization of $\epsilon_\mathrm{approx}$, $\nu_\mu$ and $C_v$ in Theorem 4.3.**
> Upper bounds involving terms similar to $\epsilon_\mathrm{approx}$, $\nu_\mu$ and $C_v$ have been established for the linear convergence of natural policy gradient (NPG) with log-linear policies in Theorem 1, 3 and 4 in [8], and for the linear convergence of NPG with tabular softmax policies in Theorem 10 and 14 in [7]. The particularity of Theorem 4.3 is that it unifies and extends (in terms of applications to general parametrization and arbitrary mirror maps) the current theory of policy mirror descent (PMD) analysis. Notably, Theorem 4.3 recovers  the best known linear convergence results of [7] and [8], as indicated in Lines 298-300, in Lines 304-306 and in Table 2.
>
> In order to shed a clearer light on how to interpret Theorem 4.3, we provide below the characterizations that we have obtained for the approximation error $\epsilon_\mathrm{approx}$, the distribution mismatch coefficient $\nu_\mu$, and the concentrability coefficient $C_v$ and put them into perspective with the existing literature. For additional discussion, please refer to Lines 252-261, Lines 270-273, Lines 280-284, Remark 3.3, and Appendices A.3, E, and F of our paper.
>
> **1.** The approximation error $\epsilon_\mathrm{approx}$, which is defined in Assumption (A1) as the error incurred when solving the regression problem in Line 1 of Algorithm 1, characterizes how well the function $f^\theta$ approximates the $Q$-function. This approximation error follows from the conventional compatible function approximation analysis by [1], which is, to the best of our knowledge, one of the current best known proof techniques for policy gradient analysis. Together with $\epsilon_\mathrm{approx}$, this proof technique has been extensively analysed in the literature [8,9,10,11].
>
> Assumption (A1) for $\epsilon_\mathrm{approx}$ is in fact weaker than Assumption 6.1 made in the seminal paper [1], which is a very common assumption in the literature and can be expressed as
> $$
> \mathbb{E}[\|Q\^t-w\^\top\nabla\_\theta\log\pi\^t\|\_{L_2(d\^t\_\mu\cdot\pi\^t)}] \leq \epsilon\_\mathrm{approx},
> $$
> because we do not contrain the approximation of the $Q$ function to be linear in $\nabla_\theta\log\pi^t$.
>
> Furthermore, we show that Assumption (A1) holds when the parametrization function $f^\theta$ is parametrized as a shallow ReLU network in Appendix G. This result allows us to obtain the sample complexity of AMPO for the important case of neural network parametrization.
>
> **2.** As to the concentrability coefficient $C_v$, terms of this nature are unavoidable in all the fast linear convergence analysis of approximate PMD (see Appendix F for citations). Moreover, our concentrability coefficient $C_v$ is weaker than the one in Assumption 9 in [8] which was previously the ''best'' known concentrability coefficient among the works cited in Lines 1128-1130.
> This is because we have full control over $v^t$ in Assumption (A2), while [8] only has full control over the initial state-action distribution.
> Lastly, we provide an upper bound for $C_v$, that is
>
> $$\mathbb{E}\_{(s,a)\sim v\^t}\bigg[\left(\frac{d\_\mu\^\pi(s)\pi(a|s)}{v\^t(s,a)}\right)\^2\bigg] \leq \frac{1}{(1-\gamma)\min\_{(s,a) \in \mathcal{S} \times \mathcal{A}}\nu(s,a)},$$
>
> where $\nu$ is the initial state-action distribution. However, this upper bound is very pessimistic. When $\pi^t$ and $\pi^{t+1}$ converge to $\pi^\star$, we can choose $v^t \in \{d_\mu^\star\cdot\pi^\star, d_\mu^{t+1}\cdot\pi^{t+1}, d_\mu^\star\cdot\pi^t, d_\mu^{t+1}\cdot\pi^t\}$ such that $C_v$ is close to $1$.
>
> **3.** As to the distribution mismatch coefficient $\nu_\mu$, it is a standard term among the works cited in Lines 281-283. We provide an upper bound for $\nu_\mu$, that is
> $$\max_{s\in\mathcal{S}}\frac{d_\mu^\star(s)}{d_\mu^t(s)}\leq\frac{1}{1-\gamma}\max_{s\in\mathcal{S}}\frac{d^\star_\mu(s)}{\mu(s)}:= \nu_\mu'.$$
> We also provide lower and upper bounds for $\nu_\mu'$, i.e.
> $$1/(1-\gamma) \leq \nu_\mu' \leq 1/((1-\gamma)\min_s\mu(s)).$$
> The upper bound $1/((1-\gamma)\min_s\mu(s))$ of $\nu_\mu'$ is very pessimistic, while the lower bound $\nu_\mu' = 1/(1-\gamma)$ can be achieved by choosing $\mu = d_\mu^\star$. In Appendix F, we also cover the case where $\mu$ does not have full support on the state space. Overall, $\nu_\mu$ is finite and can be reasonably small.
>
> **Time-step dependence of the assumptions.** As we argued in our above reply **Characterization of $\epsilon_\mathrm{approx}$, $\nu_\mu$ and $C_v$ in Theorem 4.3**, assumptions on $\pi^t$ regarding the approximation error, the concentrability coefficient, and the distribution mismatch coefficient are widely adopted in previous literature [1,7,8,9,10,11]. Our Assumptions (A2) and (A3) are already the weakest among them and we provide upper bounds that do not depend on the time-step $t$.
>
> **Literature.** The application of mirror descent to policy optimization is a vast and growing body literature regarding the design of new algorithms and the study of their theoretical properties. Within this literature, our work distinguishes itself by being the first to proposes performing the update in the dual space rather than in the policy space and obtaining the policy through a Bregman projection. Thanks to this idea, we are able to provide, to the best of our knowledge, the first result that establishes linear convergence for a policy gradient-based method involving general policy parametrization. Our works paves the way for a new direction of research that investigates the properties of specific mirror maps in the general parametrization setting.

---

> > ### Comment · Reviewer_WBF5 · 2023-08-20
> >
> > Thanks very much for authors' response. I am convinced by the explanation for the approximation error $\epsilon_{approx}$ and thus will raise my score to accept.

---

### Official Review · Reviewer_zS6s · 2023-07-20

**Soundness:** 2 fair
**Presentation:** 3 good
**Contribution:** 3 good
**Rating:** 7
**Confidence:** 4

**Summary:**

While policy mirror descent methods have been theoretically analyzed in many past works to establish strong convergence guarantees, questions remain regarding the implementation of function approximation in this class of algorithms. The work analyzes general approximators with PMD, obtaining linear convergence guarantees up to standard approximation error. The theoretical approach also yields past algorithms as special cases, and the authors analyze the neural network approximation case as a corollary of their result. The theory also incorporates slightly weaker assumptions/constants in the bounds compared to past work.

**Strengths:**

I think the theoretical contributions of the paper are interesting.

- The paper contributes to understanding how theoretically sound function approximation could be used in the case of policy mirror descent based methods, which already admit strong linear convergence guarantees.
- The paper is well-written, and the main statements are rigorous and well-defined.
- The theory unifies many past policy-based approaches under a framework, which is a meaningful contribution to RL theory.
- AMPO, while having theoretical guarantees, seems to be practically relevant and straightforward to implement.
- The authors provide a comprehensive comparison with existing results, and their algorithm is well-motivated with examples. The assumptions they introduce seem to be weaker than certain previous results,

**Weaknesses:**

- Certain parts of the theoretical results (Sections 3.1, 4.2) are adaptations of existing results, and in the case of Section 4.2, it is not clear how the paper uses certain assumptions/quantities in the referred work in their context.
- The particular assumption (A1) and the final result incorporating a $\mathcal{O}(\sqrt{\varepsilon_{approx}})$ bias could be compared to existing results on approximate PMD, see Q4 below.
- No experimental results: I think this is a minor weakness. As the main contribution of the paper is theoretical, benchmarking scores with well-tuned existing algorithms could be in my opinion irrelevant. However, since the starting point of the paper was the practical success of PO with function approximation, at least the practical implementation of Algorithm 1 could be demonstrated in some cases.

**Questions:**

Q1: The $\omega$-potential mirror maps seem to induce a wide range of mirror maps that have a sum of functions of components structure. Are there relevant cases where the $\omega$-potential mirror map condition is not satisfied?

Q2: In the case of two-layer neural network parameterization (Section 4.2), I found it difficult to compare the assumptions in the paper and the reference work by Allen-Zhu et al, 2019a. The referred work includes a term $OPT$ that depends on the population error (i.e., best achievable error with a finite width neural network). Why can this term be ignored in the Section G, when quantifying $\varepsilon_{approx}$? While neural networks are universal approximators, I couldn’t follow how large the width $m$ would need to be to control $OPT$, which in turn affects the sample complexity. For example, could it be the case that achieving $OPT < \delta$ requires $m \sim \exp{\delta^{-1}}$, introducing an additional polynomial term in Corollary 4.4? Or is Corollary 4.4 stated up to an error of $OPT$? A related assumption that was not clear to me was the realizability of the target Q-values (potentially in the infinite width limit), which might need to be introduced to be able to control $OPT$ to be arbitrarily small. If this assumption is indeed necessary, it should be stated clearly.

Q3: As a related question, does the assumption that $F^*$ admitted $A^*, W^*$ with $W^*$ having row norms less than 1 introduce a smoothness assumption on the target Q function?

Q4: As a general question, PMD has been analyzed in the case where policy evaluation or the optimization subproblem can not be computed exactly but approximately, for instance in [Wenhao Zhan, 2021], Section 3.2, Theorem 2, and potentially others. These results also potentially incorporate approximation error, similar to (A1), is there a fundamental difference beyond the incorporation of the particular mirror map? As far as I understand, Theorem 2 of [Wenhao Zhan, 2021] also incorporates linear convergence up to a non-vanishing approximation error.

I am happy to change my score in case the authors clarify the above points.

Minor comments:
Proposition 4.2 could include the (stronger) positive term stated in the appendix as well to ease reading, since (as stated) it seems improvement is not necessarily guaranteed.

Line 985-986, missing $h$ in the equation.

**Limitations:**

The paper makes the assumptions of the theory clear. Some comments regarding assumptions that were not clear are mentioned in Q1-3 above.

---

> ### Author Rebuttal · Authors · 2023-08-08
>
> Thank you for your positive and insightful remarks. Please find the answers to your questions in Experimental results for AMPO and
> below.
>
> **Q1: $\omega$-potential class.** We are aware of two previously used mirror maps that cannot be recovered using $\omega$-potentials: $h(x) = \frac{1}{2}x^\top A x$, for some matrix $A$, which generates the Mahalanobis distance, and $p$-norms, i.e. $h(x) = \|x\|_p^2$. Note that the case where $h(x) = \|x\|_p^p$ can be recovered. We will add these observations in our revision.
>
> **Q2: Appendix G.** We agree with the reviewer that the exposition in Appendix G, especially regarding how to use the results from [2], could have been clearer. We will add the following clarifications in the proof of Corollary 4.4 in the revised version of the paper.
>
> *Assumption* Let $g:\mathcal{R}\rightarrow\mathcal{R}$ be a smooth activation function such that $g(z) = e^z,~ \sin(z)$, $\mathtt{sigmoid}(z)$, $\tanh(z)$ or is a low degree polynomial. For all time-steps $t$, we assume that there exists a target network $F^{\star,t}:\mathcal{R}^d\rightarrow\mathcal{R}^k$, with
>
> $$
> F\^{\star,t} = (f\^{\star,t}\_1, \dots, f\^{\star,t}\_k)\quad \text{and} \quad f\^{\star,t}\_r(x)= \sum\_{i=1}\^p a\^{\star,t}\_{r,i}g(\langle w\^{\star,t}\_{1,i},x\rangle)\langle w\^{\star,t}\_{2,i},x\rangle
> $$
>
> where $w\^{\star,t}\_{1,i} \in \mathbb{R}\^d$, $w\^{\star,t}\_{2,i} \in \mathbb{R}\^d$, and $a\^{\star,t}\_{r,i} \in \mathbb{R}$, such that
> $$\mathbb{E}\big[\left\lVert F\^{\star,t}- Q\^t-\eta\_t\^{-1}\nabla h(\pi\^t) \right\rVert\_{L\_2(v\_t)}\^2\big] \leq OPT.$$
> We assume for simplicity $\|w\^{\star,t}\_{1,i}\|\_2 = \|w\^{\star,t}\_{2,i}\|\_2 = 1$ and $|a\^{\star,t}\_{r,i}| \leq 1$.
>
> Compared to [2], this assumption represents a particular case of the class of loss functions for which their results hold. In the RL setting, assumptions of this type have already been made in the literature, such as the bias assumption in the compatible function approximation framework studied by [1]. The term $OPT$ represents the minimum error incurred by a target network parametrized as $F^{\star,t}$ when solving the regression problem in Line 1 of Algorithm 1.
>
> In the proof of Corollary 4.4, we will indicate that, due to the statement of Theorem 1 by [2], which we provide in Appendix G, we cannot guarantee the approximation error incurred by the learner network to be smaller than $OPT$. Consequently, since in the proof of Corollary 4.4 we require
> $$
> \frac{2(1+\nu_\mu)\sqrt{C_v \varepsilon_\mathrm{approx}}}{1-\gamma}\leq\frac{\varepsilon}{2},
> $$
> where $\varepsilon$ upper-bounds the difference between the values of the optimal policy and the last policy, we also have that
> $$
> \varepsilon\geq\frac{4(1+\nu_\mu)\sqrt{C_v OPT}}{1-\gamma}.
> $$
> A similar bound can be applied to any proof that contains the bias assumption introduced by [1].
>
> Regarding how to tune the width of the target network $F^{\star, t}$ in order to control $OPT$, [3] have shown that polynomial width suffices to approximate target functions arbitrarily well, given some regularity conditions on the target function. However, the result form [3] holds for ReLU networks, while we are not aware of results regarding the approximation properties of neural networks with smooth activation functions, that is the class to which $F^{\star, t}$ belongs. We leave addressing this gap in the literature as future work.
>
> **Q3: Assumption on bounded norms of the parameters.** The condition $\|w\^{\star,t}\_{1,i}\|\_2 = \|w\^{\star,t}\_{2,i}\|\_2 = 1$ and $|a\^{\star,t}\_{r,i}| \leq 1$ in the previous comment is only stated for simplicity and can be relaxed. We summarize here the discussion from footnote 6 in page 5 of [2] on this matter. For general $\|w^*\_{1,i}\|\_2 \leq B$, $\|w^*\_{2,i}\|\_2 \leq B$, $|a^*\_{r,i}| \leq B$, the scaling factor $B$ can be absorbed into the activation
> function $g'(x) = g(Bx)$. For the activation functions that we consider in Appendix G, that is the ones considered in Example 2.1 of [2], the sample complexity result described in Theorem G.1 of our paper is not affected by changes on the scaling factor.
>
> **Incorporating Assumption A1 in previous results.** Using the AMPO formulation of the parametrization function update in the dual space and the policy projection step, we can take advantage of Lemma 4.1 and the proof technique from [4] to make Assumption (A1) a mean square error assumption. If we were to use the proof technique from [Wenhao Zhan, 2021], we would need Assumption (A1) to hold in $\ell_\infty$-norm, due to the passages in their Appendix B.2. That is, we would have to assume that there exists $\varepsilon_\mathrm{approx}\geq 0$ such that, for all times $t\geq 0$,
>
> $$
> \mathbb{E}\big[\left\lVert f\^{t+1}- Q\^t-\eta\_t\^{-1}\nabla h(\pi\^t)\right\rVert\_\infty\big] \leq \varepsilon\_\mathrm{approx}
> $$
>
> which is a stronger assumption than our Assumption (A1), especially in contexts where the state and action spaces are very large.
>
> **Proposition 4.2.** We will write the tighter bound for Proposition 4.2 in the main body of revised version of the paper. We thank the reviewer for this suggestion.
>
>
>
> 1. Alekh Agarwal, Sham M. Kakade, Jason D. Lee, and Gaurav Mahajan. On the theory of policy gradient methods: Optimality, approximation, and distribution shift. Journal of Machine Learning Research, 2021.
> 2. Zeyuan Allen-Zhu, Yuanzhi Li, and Yingyu Liang. Learning and generalization in overparameterized neural networks, going beyond two layers. Advances in Neural Information Processing Systems, 2019.
> 3. Ziwei Ji, Matus Telgarsky, and Ruicheng Xian. Neural tangent kernels, transportation mappings, and universal approximation. In International Conference on Learning Representations, 2019.
> 4. Lin Xiao. On the convergence rates of policy gradient methods. Journal of Machine Learning Research, 2022.

---

> > ### Comment · Reviewer_zS6s · 2023-08-14
> >
> > I thank the authors for their extensive responses; most of my questions have been addressed, and I don't have additional comments reading other reviews. I will increase my score once the arguments regarding NN approximation (Appendix G) are clarified, as I found it difficult to follow the rebuttal of this question.
> >
> > I could interpret the response in several different ways, could the authors kindly clarify if I understand them correctly?
> >
> > 1. **The authors will introduce an explicit non-vanishing $OPT$ term to Corollary 4.4:** I think this option makes sense, as it seems without additional assumptions (the authors mention also in their rebuttal) $OPT$ can not be controlled. I also agree that this $OPT$ approximation error is also outside the scope of the work.
> >
> > 2. **The statement "$\varepsilon$-optimal policy" in Corollary 4.4 is stated up to $\varepsilon_{approx}$ error**: This would also be logically consistent if $OPT$ can not be controlled directly, but should be made explicit.
> >
> > 3. **The assumption of bounded approximation error (A1) readily implies the stated result of Corollary 4.4**: I think this might be wrong, as to obtain an $\varepsilon$-optimal policy using theorem 4.3, up to constants we would need $\varepsilon_{approx} \leq \varepsilon^2$, or consequently $OPT \leq \varepsilon^2$. However, both $OPT$ and $m$ is affected by parameter $p$ according to the cited Theorem G.1 and requiring $OPT \leq \varepsilon^2$ might (at least without further proof) not be consistent with $p = log(\varepsilon^{-1})$ for arbitrary smooth functions. In other words, as the authors state in the rebuttal,
> > > However, the result form [3] holds for ReLU networks, while we are not aware of results regarding the approximation properties of neural networks with smooth activation functions.
> >
> > but the width of the target network does affect the sample complexity as Theorem 1 suggests, hence the sample complexity of Corollary 4.4 would need to incorporate a function whose growth rate is unknown. In fact, upon a more careful read, the sample complexity bound by Theorem G.1 incorporates a polynomial dependency on $p$ which is the width of the "target" network inducing error $OPT$, so unlike what I suggested in my original review, an additional $poly(\varepsilon^{-1})$ term might be missing in the sample complexity. This makes it especially difficult to compare existing bounds for policy learning with neural networks.

---

> > > ### Author Response · Authors · 2023-08-15
> > > **Additional reply**
> > >
> > > Thank you for your additional insightful comments and for providing us with the opportunity to further clarify our work.
> > > - In our rebuttal, we meant exactly what you wrote in 1., that is, we will introduce an explicit non-vanishing $OPT$ term to Corollary 4.4 in the revised version of our paper.
> > >
> > > - We chose to use Theorem 1 from [2] in our case-study of neural network parametrization because it can be readily applied to our setting with little notation or prior knowledge. However, as you correctly pointed out, there is a dependency on the width of the target network that may impact the sample complexity in Corollary 4.4, even after introducing an explicit non-vanishing $OPT$ term. While our goal for Corollary 4.4 was to showcase how the application of existing results for neural networks can lead to the sample complexity of AMPO, it is true that the dependency on the width of the target network makes it difficult to compare this result with existing sample complexity results for policy learning with neural networks.
> > >
> > > At this stage, it is important to note that choosing a method different from the one proposed by [2] to solve Line 1 in Algorithm 1 of our paper with neural networks can lead to alternative, and possibly better, sample complexity results for AMPO. For example, we can obtain a sample complexity result for AMPO that does not involve a target network using results from [3] and [5], although this requires introducing more notation and background results compared to Corollary 4.4 (since in [5] they employ a temporal-difference-based algorithm, that is Algorithm 3 in their work, to obtain a neural network estimate $\widehat{Q}^t$ of $Q^t$ while in [3] they provide a method based on Fourier transforms to approximate a target function through shallow ReLU networks). We outline below the steps in order to do so (and additional details including the precise statements of the results we use and how we use them are provided thereafter for the sake of completeness).
> > >
> > >
> > > **Step 1)** We first split the approximation error in Assumption (A1) into a critic error $\mathbb{E}[\sqrt{\|\widehat{Q}\^t- Q^t\|\_{L_2(v\^t)}\^2}] \leq \varepsilon_\text{critic}$ and an actor error $\mathbb{E}[\sqrt{\|f\^{t+1}- \widehat{Q}\^t-\eta_t\^{-1}\nabla h(\pi^t)\|\_{L_2(v^t)}\^2}] \leq \varepsilon\_\text{actor}$. In this case, the linear convergence rate in our Theorem 4.3 becomes
> > > $$
> > > V^\star(\mu)-\mathbb{E}\left[V^T(\mu)\right]\leq\frac{1}{1-\gamma}\bigg(1-\frac{1}{\nu_\mu}\bigg)^T\bigg(1+\frac{\mathcal{D}^\star_0}{\eta_0( \nu_\mu-1)}\bigg)+\frac{2(1+\nu_\mu)\sqrt{C_v} (\varepsilon_\text{critic}+\varepsilon_\text{actor})}{1-\gamma}.
> > > $$
> > > [We can obtain this alternative statement by modifying the passages in Appendix D.2 of our paper. In particular, writing $f^{t+1}- Q^t-\eta_t^{-1}\nabla h(\pi^t) = (f^{t+1}- \widehat{Q}^t-\eta_t^{-1}\nabla h(\pi^t)) + (Q^t-\widehat{Q}^t)$ and bounding the two terms with the same procedure in Appendix D.2 leads to this alternative expression for the error.]
> > >
> > > We will next deal with the critic error and actor error separately.
> > >
> > > **Step 2)** Critic error. Under a realizability assumption that we provide below along with the statement of the theorem (Assumption 2 in [5]), Theorem 1 from [5] gives that the sample complexity required to obtain $\mathbb{E}[\sqrt{\|\widehat{Q}\^t- Q^t\|\_{L_2(d^t_\mu\cdot\pi^t)}\^2}] \leq \varepsilon$ is $\widetilde{O}(\varepsilon\^{-4}(1-\gamma)\^{-2})$, while the required network width is $\widetilde{O}(\varepsilon\^{-2})$.
> > >
> > > **Step 3)** Actor error. Using Theorem E.1 from [3], we obtain that $\mathbb{E}[\sqrt{\|f^{t+1}- \widehat{Q}^t-\eta_t^{-1}\nabla h(\pi^t)\|_{L_2(v^t)}^2}]$ can be made arbitrarily small by tuning the width of $f^{t+1}$, without using further samples.
> > >
> > > **Step 4)** Replacing line 1206 and Equation (52) of our paper with the sample complexity of the critic, we obtain that the sample complexity of AMPO is $\widetilde{O}(C_v^2 \nu_\mu^5/\varepsilon^4(1-\gamma)^7)$, which does not depend on the error made by a target network.
> > >
> > > We thank you for bringing this matter to our attention. In the revision of our paper, we will do the following.
> > > 1. Add at the end of Corollary 4.4, that is Line 342, ``, where $\varepsilon$ has to be larger than a fixed and non-vanishing error floor''.
> > > 2. Replace lines 343-347 with ``We provide a proof for Corollary 4.4 and an expression for the error floor in Appendix G. Note that in the case considered here, the sample complexity might be impacted by an additional $poly(\varepsilon^{-1})$ term. We refer to Appendix G for more details and a derivation which does not include an additional $poly(\varepsilon^{-1})$ term, enabling comparison with prior works.''.
> > > 3. Add to the appendix the sample complexity result that does not involve a target network.
> > >
> > > [5] Semih Cayci, Niao He, and R Srikant. Finite-time analysis of entropy-regularized neural natural actor-critic algorithm. arXiv preprint arXiv:2206.00833, 2022.

---

> > > > ### Author Response · Authors · 2023-08-15
> > > > **Continuing the reply**
> > > >
> > > > We now provide the statements of the aforementioned results we used.
> > > >
> > > > **Recalling Theorem 1 in [5] and its assumptions.**
> > > > Consider the following space of mappings:
> > > > $$
> > > > \mathcal{H}\_{{\bar{\nu}}} = \\{v:\mathbb{R}\^d\rightarrow\mathbb{R}\^d: \sup\_{w\in\mathbb{R}\^d}\\|v(w)\\|\_2 \leq {\bar{\nu}}\\},
> > > > $$
> > > > and the function class:
> > > > $$
> > > > \mathcal{F}\_{{\bar{\nu}}} = \Big\\{ g(\cdot) = \mathbb{E}\_{w\_0\sim\mathcal{N}(0,I\_d)}[\langle v(w\_0), \cdot \rangle \mathbb{I}\{\langle w_0, \cdot \rangle > 0\}]: v\in\mathcal{H}\_{{\bar{\nu}}} \Big\\}.
> > > > $$
> > > >
> > > > Consider the following realizability assumption for the Q-function.
> > > >
> > > > *Assumption 2 in [5]*
> > > > For any $t\geq 0$, we assume that $Q^t \in \mathcal{F}_{{\bar{\nu}}}$ for some ${\bar{\nu}} > 0$.
> > > >
> > > > *Theorem 1 in [5]*
> > > > Under Assumption 2 in [2], for any error probability $\delta \in (0, 1)$, let $$\ell(m',\delta) = 4\sqrt{\log(2m'+1)}+4\sqrt{\log(T/\delta)},$$ and $R > {\bar{\nu}}$. Then, for any target error $\varepsilon > 0$, number of iterations $T' \in \mathbb{N}$, network width $$m' > \frac{16\Big({\bar{\nu}} + \big(R+\ell(m',\delta)\big)\big({\bar{\nu}}+R\big)\Big)^2}{(1-\gamma)^2\varepsilon^2},$$ and step-size $$\alpha_C = \frac{\varepsilon^2(1-\gamma)}{(1+2R)^2},$$  Algorithm 3 in [5] yields the following bound:
> > > > $$
> > > > \mathbb{E}\Big[\sqrt{\|\widehat{Q}\^t- Q^t\|\_{L_2(d^t_\mu\cdot\pi^t)}^2}\mathbb{I}_{A_2}\Big] \leq \frac{(1+2R){\bar{\nu}}}{\varepsilon(1-\gamma)\sqrt{T'}} + 3\varepsilon,
> > > > $$
> > > >
> > > > where $A_2$ holds with probability at least $1-\delta$ over the random initializations of the critic network $\widehat{Q}^t$. As indicated in [5], a consequence of this result is that in order to achieve a target error less than $\varepsilon > 0$, a network width of $m' = \widetilde{O}\Big(\frac{{\bar{\nu}}^4}{\varepsilon^2}\Big)$ and iteration complexity $O\Big(\frac{(1+2{\bar{\nu}})^2{\bar{\nu}}^2}{(1-\gamma)^2\varepsilon^4}\Big)$ suffice.
> > > >
> > > > The statement of Theorem 1 in [5] can be readily applied to obtain the sample complexity of the critic.
> > > >
> > > > **Recalling Theorem E.1 in [3] and its assumptions** Let $g:\mathbb{R}^n\rightarrow \mathbb{R}$ be given and define the modulus of continuity $\omega_g$ as
> > > > $$
> > > > \omega_g(\delta):=\sup\_{x,x'\in\mathbb{R}\^n}\\{g(x)-g(x'):\max(\\|x\\|\_2,\\|x'\\|\_2)\leq 1+\delta,\\|x-x'\\|\_2\leq\delta \\}.
> > > > $$
> > > > If $g$ is continuous, then $\omega_g$ is not only finite for all inputs, but moreover $\lim_{{\delta \to 0}} \omega_g (\delta) \to 0$.
> > > >
> > > >
> > > > Denote $\\|p\\|\_{L_1} = \int|p(w)|dw$. Define a sample from a signed density $p:\mathbb{R}^{n+1}\rightarrow\mathbb{R}$ with $\\|p\\|\_{L_1} < \infty$ as $(w,b,s)$, where $(w,b)\in\mathbb{R}$ is sampled from the probability density $|p|/\\|p\\|\_{L_1}$ and $s = sign(p(w,b))$
> > > >
> > > > *Theorem E.1 in [3]*
> > > >     Let $g:\mathbb{R}^n\rightarrow \mathbb{R}$, $\delta > 0$ and $\omega_g(\delta)$ be as above and define for $x\in\mathbb{R}^n$
> > > > $$
> > > > M := \sup_{\\|x\\|\leq 1+\delta} |g(x)|, \qquad g_{|\delta}(x)= f(x)\mathbb{I}[\\|x\\|\leq 1+\delta], \qquad\alpha := \frac{\delta}{\sqrt{\delta}+\sqrt{2\log(2M/\omega_g(\delta))}}.
> > > > $$
> > > > Let $G_\alpha$ be a gaussian distribution on $\mathbb{R}^n$ with mean $0$ and variance $\alpha^2 \mathcal{I}$. Define the Gaussian convolution $l = g_{|\delta} * G_\alpha$ with Fourier transform $\widehat{l}$ satisfying radial decomposition $\widehat{l}(w)= |\widehat{l}(w)|\exp(2\pi i \theta_h(w))$. Let $P$ be a probability distribution supported on $\\|x\\|\leq 1$. Additionally define
> > > >
> > > > $$
> > > > c := g(0)g(0) \int|\widehat{l}(w)|\big[\cos(2\pi(\theta\_l(w)-\\|w\\|\_2))-2\pi\\|w\\|_2\sin(2\pi(\theta\_l(w)-\\|w\\|_2))\big]dw
> > > > $$
> > > > $$
> > > > a := \int w|\widehat{l}(w)|dw
> > > > $$
> > > > $$
> > > > r := \sqrt{n}+2\sqrt{\log\frac{24\pi^2(\sqrt{d}+7)\^2\\|g\_{|\delta}\\|\_{L_1}}{\omega\_g(\delta)}}
> > > > $$
> > > > $$
> > > > p := 4\pi^2|\widehat{l}(w)|\cos(2\pi(\\|w\\|\_2-b))\mathbb{I}[|b|\leq\\|w\\|\leq r],
> > > > $$
> > > >
> > > > and for convenience create fake (weight, bias, sign) triples
> > > > $$
> > > > (w,b,s)\_{m+1}:=(0,c,m\\, sign(c)), \quad (w,b,s)\_{m+2}:=(a,0,m), \quad (w,b,s)\_{m+3}:=(-a,0,-m).
> > > > $$
> > > > Then
> > > > $$
> > > > |c|\leq M +2\sqrt{n}\\|g\_{|\delta}\\|\_{L_1}(2\pi\alpha^2)\^{-d/2},
> > > > $$
> > > > $$
> > > > \\|p\\|\_{L_1}\leq2\\|g\_{|\delta}\\|\_{L_1}\sqrt{\frac{(2\pi)^3 n}{(2\pi\alpha^2)\^{n+1}}},
> > > > $$
> > > >     and with probability at least $1-3\lambda$ over a draw of $((s_j, w_j, b_j))\_{j=1}^m$ from $p$
> > > > $$
> > > >         \sqrt{\Big\\|g-\frac{1}{m}\sum_{j=1}^{m+3} s_j\sigma(\langle w_j, x\rangle+ b_j)\Big\\|\_{L(P)}} \leq 3\omega_{g}(\delta)+\frac{r \|p\|_{L_1}}{\sqrt{m}}\left[1+\sqrt{2\log(1/\lambda)}\right].
> > > > $$
> > > >
> > > > We can then characterize the error of the actor by choosing $x = (s,a)$, $g = \widehat{Q}^t + \eta_t^{-1}\nabla h(\pi^t)$, and $f^{t+1} = \frac{1}{m}\sum_{j=1}^{m+3} s_j\sigma(\langle w_j, x\rangle+ b_j)$. We can then make the actor error arbitrarily small by tuning the network width $m$ and $\delta$ (note that, since both $\widehat{Q}^t$ and $f^t$ are continuous neural networks, $g$ is a continuous function).

---

> > > > > ### Comment · Reviewer_zS6s · 2023-08-18
> > > > >
> > > > > Thank you for the comprehensive response, I think I agree with the currently presented changes which would fix the error in the original submission.
> > > > >
> > > > > I agree that the error is in the neural network "case study" of the proposed method and is not the core contribution. Since the authors proposed a valid fix, and they have comprehensively answered my remaining questions (for instance providing a working implementation) I will raise my score.

---

### Official Review · Reviewer_oNAx · 2023-07-21

**Soundness:** 3 good
**Presentation:** 3 good
**Contribution:** 3 good
**Rating:** 7
**Confidence:** 4

**Summary:**

This paper extends the recently introduced policy mirror descent method of tabular setting to function approximation setting. The essential development seems to reside in exploiting the mirror descent update in the dual space and projecting the dual representation of the policy into a "realizable" representation that is consistent with the function approximation used. The author recovers both sublinear and linear convergence when such projection error can be controlled. The resulting method is thus implementable with general parameterization of the policy.

**Strengths:**

1. I enjoy the simplicity of this idea. It is well known that when using general function approximation of the policy it becomes unclear how basic three-point lemma holds as we are performing descent in the parameter space, not the policy space. This paper uses the well-known interpretation of the mirror descent method and suggests to first project the dual representation of the updated policy into the one that is representable by the function approximator. Consequently, one is directly doing approximate descent directly in the policy space and the existing framework of policy mirror descent can be adapted.

2. I also appreciate discussions on the potential choices of distance-generating functions. Example 3.5 and 3.6 appears to be new to me. Theorem 4.5 on how to control the policy projection error with neural network parameterization seems to be useful.

3. Minor comment: using Tsallis divergence as the distance-generating function has been discussed explicitly in arXiv preprint arXiv:2303.04386, 2023.

**Weaknesses:**

1. It appears to me that Lemma 4.1 can be understood as using the projected dual variables $f^{\theta}_s - \nabla h(\overline{\pi}_s)/\eta$ instead of $Q_s$ when doing the policy mirror descent update, especially in the analysis $\overline{\pi}$ is set to be the last policy.  In this sense, Lemma 4.1 seems to be a fairly standard result. This can be seen from
\begin{align}
 \tilde{\pi}_s =  \mathrm{argmin}  <- \eta f^{\theta}_s + \nabla h(\overline{\pi}_s), \pi> + h(\pi) - <\nabla h(\overline{\pi}_s),\pi>
\end{align}
and consequently
\begin{align}
 \tilde{\pi}_s =  \mathrm{argmin}  <- \eta f^{\theta}_s + \nabla h(\overline{\pi}_s), \pi> + D(\pi, \overline{\pi}_s).
\end{align}
Given the above observation standard descent lemma can be applied and yields Lemma 4.1.

2. I personally would much appreciate the perspective of this paper -- the projection step of the policy -- instead of the technicality/importance of Lemma 4.1 that the paper is currently promoting. Kindly correct me if I have missed anything.



**Questions:**

I enjoy reading this paper with great clarity in presenting its essential idea. I have worked on this topic for a while and therefore I do not have additional technical questions.

**Limitations:**

Yes, the authors have adequately addressed the limitations and potential future improvement.

---

> ### Author Rebuttal · Authors · 2023-08-08
>
>
> Thank you for your positive and insightful remarks. Please find the answers to your questions in Experimental results for AMPO and
> below.
>
> **Tsallis divergence.** We thank the reviewer for pointing out the missing prior work [1] regarding the use of Tsallis entropy. We include it in Line 218 in the revised version of the paper.
>
> **Alternative proof of Lemma 4.1.** We are indebted to the reviewer for providing the alternative insight on $\tilde{\pi}_s$ in Lemma 4.1. We agree with the reviewer and provide the complete derivation here.
>
> $$
> \begin{aligned}
> \tilde{\pi}\_s &= \arg\min\_{\pi\' \in \Delta(\mathcal{A})} \mathcal{D}\_h(\pi\', \nabla h^*(\eta f\_s\^\theta)) \\\\
> &= \arg\min\_{\pi' \in \Delta(\mathcal{A})} h(\pi') - h(\nabla h^*(\eta f\_s\^\theta)) - <\nabla h(\nabla h^*(\eta f\_s\^\theta)), \pi' - \nabla h^*(\eta f\_s\^\theta)> \\\\
> &= \arg\min\_{\pi' \in \Delta(\mathcal{A})} h(\pi') - <\eta f\_s\^\theta, \pi'> \\\\
> &= \arg\min\_{\pi' \in \Delta(\mathcal{A})} <-\eta f\_s\^\theta + \nabla h(\bar{\pi}\_s), \pi'> + h(\pi') - h(\bar{\pi}\_s) - <\nabla h(\bar{\pi}\_s), \pi'> \\\\
> &= \arg\min\_{\pi' \in \Delta(\mathcal{A})} <-\eta f\_s\^\theta + \nabla h(\bar{\pi}\_s), \pi'> + \mathcal{D}\_h(\bar{\pi}\_s, \pi'), \quad \quad \quad (*)
> \end{aligned}
> $$
>
>
>
>
> where the second and the last lines are obtained by the definition of the Bregman divergence, and the third line is obtained by Equation (4): $\nabla h(\nabla h^*(x^*)) = x^*$ for all $x^* \in \mathbb{R}^{|\mathcal{A}|}$. Consequently, we can apply the three-point descent lemma in [2] to prove Lemma 4.1.
> We will include this proof in the revised version of the paper in Appendix D.1.
>
> **Emphasis of the projection step.** Based on the previous point, we agree with the reviewer that the key contribution of the paper is the idea of performing the parametrization function update in the dual space rather than in the policy space and obtaining the updated policy through the projection step.
> We will replace the comment about extending the three-point descent lemma from convex to non-convex functions, which corresponds to Lines 233-237, to better emphasize the contribution in the revised version of the paper.
>
> Please note that the comments made in Lines 174-184 still hold. In particular, our formulations of the update and the projection step are necessary to apply Lemma 4.1, which on the contrary cannot be applied to previous updates such as Equation (11). This improvement represents an important technical contribution that allows to extend the analysis from the tabular ([2]) and linear parametrization ([3]) settings to the general policy parametrization setting and we will make sure to highlight it in our revision.
>
> 1. Yan Li and Guanghui Lan. Policy Mirror Descent Inherently Explores Action Space, 2023.
> 2. Lin Xiao. On the convergence rates of policy gradient methods. Journal of Machine Learning Research, 2022.
> 3. Rui Yuan, Simon Shaolei Du, Robert M. Gower, Alessandro Lazaric, and Lin Xiao. Linear convergence of natural policy gradient methods with log-linear policies. In International Conference on Learning Representations, 2023.

---

> > ### Comment · Reviewer_oNAx · 2023-08-11
> > **Thank you for the rebuttal**
> >
> > I have read through the authors' rebuttal and appreciate their effort in the experiments and their additional explanations. I personally enjoy the paper and believe it should be included in the conference. I will increase my score if Reviewer WBF5 does not.

---

> > > ### Author Response · Authors · 2023-08-15
> > > **Thank you**
> > >
> > > Thank you for your positive feedback, we are grateful for your support.

---

> > > > ### Comment · Reviewer_oNAx · 2023-08-18
> > > >
> > > > I increased my score to 7 as I believe this paper should be included in the conference, and Reviewer WBF5 has not responded to the rebuttal.

---

### Official Review · Reviewer_rns2 · 2023-07-27

**Soundness:** 2 fair
**Presentation:** 3 good
**Contribution:** 3 good
**Rating:** 5
**Confidence:** 2

**Summary:**

This is a theoretical work and the main claim is the proof of linear convergence rate for policy mirror descent (PMD) algorithms with general policy parametrization. This contrasts with previous results (Lan 2022, Xiao 2022, Cen et al 2021, Zhan et al. 2021,  Cayci et al. 2021, etc.) that proved convergence rates for either tabular or other specific policy parametrizations. The work develops the proof by creating a framework that incorporates general policy parameterization into PMD: Approximate Mirror Policy Optimization (AMPO). The analysis performs a non-trivial extension of the three-point descent lemma (Chen and Teboulle, 1993, Lemma 3.2) and uses the proof techniques of Xiao et al. (2022) for PMD.

**Post rebuttal update**

The authors did several changes to answer my review, e.g., they implemented their algorithm and ran some experiments.
I raised my score by 1 point due to including an implementation of the algorithm. The experiments did not fully convince me, so I did not raise my score further (see my discussion with the reviewers for more details). My impression is that the contribution is mainly theoretical, and it fits into the literature on theoretical works in this topic. Looking at the constants, and discussing with the authors, I also had the impression that there is a gap in the theory and practice in this domain. It seems that none of the previous works performed experiments to validate the theory either, and replacing the constants in the bounds with reasonable values makes the bounds vacuous. This is not really a problem of the current work, as previous theoretical works in this domain seemed similar. I would encourage the authors to critically consider these points, and discuss limitations of the theory, unless the experiments strongly support the theory.

**Strengths:**

- The paper is well-written and explains how the contributions relate to previous work.
- It appears rigorous.
- Previous proofs had quite restrictive settings, whereas the submitted paper's theorems allow for neural network parameterizations of the policies, so this is quite important for more realistic theory, and seems like a substantial contribution to this line of work.

**Weaknesses:**

- There are no experiments. It may be good to have to some illustrative experiments that allow understanding the contribution at a glance.
- The paper is dense and all of the proofs are in the appendix. Given that reviewers are not obligated to read the appendix, and that no evidence or intuitively clear reasons for why the theory is true is provided in the main paper, it is difficult to judge the correctness of the work without reviewers going beyond what is expected of them (although the work does recover the best known rates in the tabular setting as well as several other previous restricted settings, which is promising). The main related past works were all published at journals, so I wonder whether submitting to a journal may be a better option. I think that conference theory papers benefit from intuitive examples and experiments in the main part of the paper.

**Questions:**

I would suggest adding experiments and figures that demonstrate how the work improves over previous bounds.

**Limitations:**

Yes, adequately addressed.

---

> ### Author Rebuttal · Authors · 2023-08-08
>
>
> Thank you for your review and remarks. Please find the answers to your questions in Experimental results for AMPO and below.
>
> **Interpretation of the theory.**
> An interpretation of our theory can be provided by connecting AMPO to the Policy Iteration algorithm, which shares the linear convergence of AMPO. To see this, we first rewrite the Bregman projection step of AMPO (Line 2 of Algorithm 1) as
> $$
> \pi\^{t+1}\_s= \arg\min_{\pi \in \Delta(\mathcal{A})} \langle -\eta_t f_s^{\theta^{t+1}} + \nabla h(\pi^t_s), \pi \rangle + \mathcal{D}_h(\pi, \pi^t_s), \quad \forall s \in \mathcal{S},
> $$
>
> $$
> \qquad = \arg\min_{\pi \in \Delta(\mathcal{A})} \langle - f_s^{\theta^{t+1}} + \frac{1}{\eta_t}\nabla h(\pi^t_s), \pi \rangle + \frac{1}{\eta_t}\mathcal{D}_h(\pi, \pi^t_s), \quad \forall s \in \mathcal{S},
> $$
>
> as suggested by Reviewer **oNAx** in Equation ($*$) for the **alternative proof of Lemma 4.1**. Secondly, we have$
> f^\{t+1}\_s - \frac{1}{\eta_t}\nabla h(\pi\^t\_s) \approx Q\^{\pi\^t}\_s
> $
>  by solving Line 1 of Algorithm 1 (see Line 239 as well).
> When the step-size $\eta_t \rightarrow \infty$, and  $1/\eta_t \rightarrow 0$, the above AMPO policy update viewpoint thus becomes
>
> $$
> \pi\^{t+1}\_s = \arg\min_{\pi \in \Delta(\mathcal{A})} <-Q^{\pi^t}_s, \pi>, \quad \forall s \in \mathcal{S},
> $$
>
> $$
> \qquad= \arg\max_{\pi \in \Delta(\mathcal{A})} < Q^{\pi^t}_s, \pi>, \quad \forall s \in \mathcal{S},
> $$
>
> which is the Policy Iteration algorithm.
> Here we ignore the Bregman divergence term $\mathcal{D}_h(\pi, \pi^t_s)$ as it is multiplied by a term that goes to 0. So AMPO behaves more and more like Policy Iteration with large enough step size and is able to converge linearly as Policy Iteration does. We will add this interpretation in the revised version of the paper.
>
> **Proof sketch.** Thank you for your suggestion, we will provide a proof sketch in the revised version of the main paper. In a nutshell, the convergence rates of AMPO are obtained by building on Lemma 4.1 (please refer to Appendix D.1 for its proof) and leveraging the policy mirror descent (PMD) proof techniques of [1]. More specifically, following the conventional compatible function approximation approach in [2], the idea is to write the global optimum convergence results in an additive form, that is
> $$\mbox{sub-optimality gap} \leq \mbox{optimization error} + \mbox{approximation error}.$$
> The separation between the two errors is allowed by Lemma 4.1, while the optimization error is bounded through the proof technique of [1] and the approximation error is characterized by Assumption (A1). Once Lemma 4.1 is established, the proof can be decomposed into three main steps.
>
> *Step 1.* Using Lemma 4.1 with $\bar{\pi} = \pi^t$, $f^\theta=f^{t+1}$, $\eta =\eta_t$, $\tilde{\pi} = \pi^{t+1}$ , and $\pi_s = \pi^t_s$, we obtain that
>
> $$
> \langle \eta_t f\^{t+1}_s - \nabla h(\pi\^t\_s),\pi\^{t+1}_s - \pi\^t\_s\rangle \geq 0,
> $$
>
> which characterizes the improvement of the updated policy.
>
> *Step 2.* Assumption (A1), Step 1, the performance difference lemma (Lemma 1 in [1]), and Lemma 4.1 with $\bar{\pi} = \pi^t$, $f^\theta=f^{t+1}$, $\eta =\eta_t$, $\tilde{\pi} = \pi^{t+1}$, and $\pi_s = \pi^\star_s$ permit us to obtain the following proposition.
>
> *Proposition D.6.*
> Let $\Delta_t := V^\star(\mu) - V^t(\mu)$. For the iterates of Algorithm 1, at each time $t\geq0$, we have
> $$
> \mathbb{E}\left[\nu\_\mu\left(\Delta\_{t+1}-\Delta\_t\right)+\Delta\_t\right]\leq\mathbb{E}\left[\frac{\mathbb{E}\_{s \sim d\_\mu\^\star}\left[\mathcal{D}\_h(\pi\^\star\_s,\pi\^t\_s)\right]}{(1-\gamma)\eta\_t} -\frac{\mathbb{E}\_{s \sim d\_\mu\^\star}\left[\mathcal{D}\_h(\pi\^\star\_s,\pi\^{t+1}\_s)\right]}{(1-\gamma)\eta_t}\right]+(1+\nu\_\mu)\frac{2\sqrt{C\_v \varepsilon\_\mathrm{approx}}}{1-\gamma}.
> $$
>
> *Step 3.* Proposition D.6 leads to the sublinear convergence rate using a telescoping sum, and to the linear convergence rate by  rearranging into
>
> $$
> \mathbb{E}\left[\Delta\_{t+1} + \frac{\mathbb{E}\_{s \sim d\_\mu\^\star}\left[\mathcal{D}\_h(\pi\^\star\_s,\pi\^{t+1}\_s)\right]}{(1-\gamma) \nu\_\mu\eta\_t}\right] \leq \left(1-\frac{1}{ \nu\_\mu}\right)\mathbb{E}\left[\Delta\_t+\frac{\mathbb{E}\_{s \sim d\_\mu\^\star}\left[\mathcal{D}\_h(\pi\^\star\_s,\pi\^t\_s)\right]}{(1-\gamma)\eta\_t( \nu\_\mu-1)}\right] + \left(1+\frac{1}{\nu\_\mu}\right)\frac{2\sqrt{C_v \varepsilon\_\mathrm{approx}}}{1-\gamma},
> $$
> which then induces a recursion.
>
> 1. Lin Xiao. On the convergence rates of policy gradient methods. Journal of Machine Learning Research, 2022.
> 2. Alekh Agarwal, Sham M. Kakade, Jason D. Lee, and Gaurav Mahajan. On the theory of policy gradient methods: Optimality, approximation, and distribution shift. Journal of Machine Learning Research, 2021.

---

> > ### Comment · Reviewer_rns2 · 2023-08-15
> >
> > Thanks for the rebuttal, I appreciate the incorporation of some suggestions.
> >
> > Regarding the experimental work:
> >
> > The demonstration of an actual implementation is useful as it makes
> > clear that the algorithm can be implemented, and may have practical
> > utility.
> >
> > The experimental results looking at the bounds were not that convincing to me though, as you tune the constants to make the curves
> > fit the data, and also because the curves are quite noisy. Probably
> > it's not easy to create thorough experimental validations during the
> > short time-period of the rebuttal, but if you have some falsifiable
> > predictions that can be made with your theory, then check these by
> > experiments, it would be convincing to me.
> >
> > I raise my score by 1 point due to including an implementation of the
> > algorithm.

---

> > > ### Author Response · Authors · 2023-08-17
> > > **Additional reply**
> > >
> > >
> > > Thank you for recognizing the potential practical utility of our work and for your additional comments. We provide a reply below.
> > >
> > > **Convergence rate charts.** The main purpose of the numerical experiments we performed was to validate the **rate** of decay of the bounds prescribed by our theoretical results (i.e. sub-linear and linear convergence, c.f. Theorem 4.3). As we focused on confirming these decay **rates**, we followed the established approach to tune constant factors to ensure a better alignment between the curves from our numerical experiments and our theoretical bounds, as you emphasized in your comments.
> > >
> > > **Noise reduction.** We increased the number of runs of AMPO that we perform to 50. This change greatly diminishes the noise in our charts. Unfortunately the rebuttal guidelines prevent us from adding additional plots here or to use links to external websites. We can now readily provide charts illustrating the median (represented by a solid line) alongside the 25% and 75% quantiles (depicted as lower and upper bands in a shaded regions). Our updated simulations confirm that the median results from our tests showcase a decay that is consistent with both sub-linear and linear rates.
> > >
> > > **Further testing if our convergence results agree with empirical findings.** We further checked the validity of our assumptions and of our convergence rates by verifying that linear convergence cannot be guaranteed if we break our assumptions. To do so, we modified the environment so that it would always start from the same state, making the starting state distribution deterministic. We then can no longer satisfy (A3) (since we cannot bound the distribution mismatch coefficient by $1/(1-\gamma)\min_s\mu(s)$, where $\mu$ is the starting state distribution, as we do in Appendix F). After this modification, we plot the average convergence speed of AMPO with $\phi(x) = \sinh(x)$ and increasing step-size over 50 runs. The convergence speed of AMPO appears to stop being linear after the first 10 000 iterations, with the average return only getting worse. We are happy to include these experiments in our revised paper should you want us to.

---

> > > > ### Comment · Reviewer_rns2 · 2023-08-18
> > > >
> > > > I am not convinced by the convergence rate charts because it is not that clear that the fitted lines correctly show the rate.
> > > >
> > > > For example, for comparing the O(1/T) and O(1/sqrt(T)) graphs, if you plot it as a loglog plot, the lines should be linear, and the convergence rate can be computed as the slope of the chart. Probably this slope will not be constant for the experiment, due to the approximation error and transient effects, so it is not clear where you should compute the slope at. However, if the slope were constant and roughly matched your predicted O(1/T) rate it would be good evidence, but it doesn't seem like it does.
> > > >
> > > > In the plots with the linear convergence rate, how did you compute the slope of the line? My understanding is that this was not computed from your bound, but the slope of the line was fit to the data. It's not clear that this line is the best fit. For example, in Figure 2 (b), a curved line would fit the data better. It also seems the line won't match any more if you continue training past 15000 steps. If you compute the slope of the line by estimating the constants using the theory in your bound, it would be more surprising when the bound fits the data, and it would be more convincing.
> > > >
> > > > The further results you describe also seem preliminary. My impression was that the other methods will also stop improving if you increase the training time. And your bound has the approximation error as a hard limit on the convergence, so we would not expect the error to go down indefinitely at a high rate (and the speed decreasing after 10000 steps as you described does not seem to give strong evidence).
> > > >
> > > > I think it is not that easy to produce experiments that would convince me due to the fairly general assumptions in the theory, and due to the short time for the rebuttal. One idea might be to change the $\gamma$ constant, and see whether the change in convergence rate matches your theoretical prediction (though I guess it would also affect how the other constants change, so it won't be surprising if it doesn't). I realize the work is theoretical. The main contributions appear to be the technical details in the proofs in the Appendix, which I haven't checked in detail. As there were no experiments, I was also not convinced of the practical utility of the theory. For these reasons, I am reluctant to further increase the score. I think conference theory papers benefit from clear experiments and intuitive explanations/results in the main part of the paper.

---

> > > > > ### Author Response · Authors · 2023-08-20
> > > > > **Additional reply**
> > > > >
> > > > > **[Existing papers on theory for RL]** As acknowledged by the reviewer, our work contributes to the *theory* of reinforcement learning methods—in particular establishing upper bounds for a family of algorithms parametrized by the choice of a mirror map and a parameterization function. As such, our work fits into an established and growing body of works published in conferences such as NeurIPS, ICML, ICLR, AISTATS that focus *exclusively* on establishing theoretical results and *do not include numerical simulations*. For instance, see [1-14] below. Considering this existing body of literature, we wish to ensure that our work is not penalised due to the absence of experiments in our initial submission, especially because we have now provided an empirical evaluation of AMPO in our rebuttal that validates our theoretical bounds [see next point].
> > > > >
> > > > >
> > > > > **[Plots]** To address the feedback of the reviewer, we have designed experiments to validate the sub-linear and linear convergence rates prescribed by our theoretical results (Theorem 4.3). It is worth noticing that our theoretical results are just about *upper* bounds, and we have no claim of tightness / lower bound. It is also worth reiterating that employing the exact constants derived from our bounds is, in general, *not* needed for the purpose of confirming sub-linear and linear convergence rates. In a log-log plot, for instance, the slope of the bound (that capture the rate of decay) remains unaffected by multiplicative constants, as these constants solely impact the intercept.
> > > > >
> > > > > 1) To validate the sub-linear rate ($O(1/T)$) in the first upper bound in Theorem 4.3, we have plotted in a log-log plot the average error of AMPO over 50 runs with constant step-size and $\phi(x)=e^x$. The error is tightly bounded by a line with slope -1, in this case confirming the tightness of the $O(1/T)$ rate.
> > > > >
> > > > > 2) To validate the linear rate ($O(exp(c T))$) in the second bound in Theorem 4.3, we have plotted in a log-linear plot the average error of AMPO over 50 runs with increasing step size. The initial step-size $\eta_0$ and the increase rate $c>1$, whereby $\eta_{t+1}= c\eta_t$, have been chosen by grid-searching over a small set of parameters. In fact, we are not able to compute the exact (best) value of $\nu_\mu$ as per Assumption (A3), as we do not have access to the transition dynamics. In the setting of a 2-layer neural network parametrization, the error is tightly bounded by a line with slope c = -0.000105, which corresponds to the value of $\nu_\mu\simeq 9500$ in the bound in Theorem 4.3.
> > > > >
> > > > >
> > > > > **[Intuition on the results and proof sketch]** We have provided in our rebuttal both an intuitive interpretation of our results and a proof sketch for them, which we plan on including in the main body of our revised paper. We hope that with this inclusion we have addressed the reviewer's concern regarding the lack of "evidence or intuitively clear reasons for why the theory is true" in the main paper.
> > > > >
> > > > >
> > > > > Under the light of these considerations and our previous replies, we would like to kindly ask the reviewer to reconsider the weaknesses they have highlighted [we believe we have addressed all of them] and reassess their evaluation of our paper. We remain open to addressing any further queries or concerns.
> > > > >
> > > > >
> > > > > [1] Alekh Agarwal, Sham M. Kakade, Jason D. Lee, and Gaurav Mahajan. Optimality and approximation with policy gradient methods in markov decision processes. Conference on Learning Theory, 2020.
> > > > >
> > > > > [2] Jalaj Bhandari and Daniel Russo. On the linear convergence of policy gradient methods for f inite MDPs. In International Conference on Artificial Intelligence and Statistics, 2021.
> > > > >
> > > > > [3] Zaiwei Chen and Siva Theja Maguluri. Sample complexity of policy-based methods under off-policy sampling and linear function approximation. In International Conference on Artificial Intelligence and Statistics, 2022.
> > > > >
> > > > > [4] Yuhao Ding, Junzi Zhang, and Javad Lavaei. On the global optimum convergence of momentum-based policy gradient. In International Conference on Artificial Intelligence and Statistics, 2022.
> > > > >
> > > > > [5] Simon Du, Sham Kakade, Jason Lee, Shachar Lovett, Gaurav Mahajan, Wen Sun, and Ruosong Wang. Bilinear classes: A structural framework for provable generalization in RL. In International Conference on Machine Learning, 2021.
> > > > >
> > > > > [6] Maryam Fazel, Rong Ge, Sham Kakade, and Mehran Mesbahi. Global convergence of policy gradient methods for the linear quadratic regulator. In International Conference on Machine Learning, 2018.
> > > > >
> > > > > [7] Yuzheng Hu, Ziwei Ji, and Matus Telgarsky. Actor-critic is implicitly biased towards high entropy optimal policies. In International Conference on Learning Representations, 2022.

---

> > > > > > ### Author Response · Authors · 2023-08-20
> > > > > > **Continuing the reply**
> > > > > >
> > > > > > [8] Boyi Liu, Qi Cai, Zhuoran Yang, and Zhaoran Wang. Neural trust region/proximal policy optimization attains globally optimal policy. Advances in Neural Information Processing Systems, 2019.
> > > > > >
> > > > > > [9] Jincheng Mei, Chenjun Xiao, Csaba Szepesvari, and Dale Schuurmans. On the global convergence rates of softmax policy gradient methods. In International Conference on Machine Learning, 2020.
> > > > > >
> > > > > > [10] Lingxiao Wang, Qi Cai, Zhuoran Yang, and Zhaoran Wang. Neural policy gradient methods: Global optimality and rates of convergence. In International Conference on Learning Representations, 2020.
> > > > > >
> > > > > > [11] Tengyu Xu, Zhe Wang, and Yingbin Liang. Improving sample complexity bounds for (natural) actor-critic algorithms. In Advances in Neural Information Processing Systems, 2020.
> > > > > >
> > > > > > [12] Rui Yuan, Robert M. Gower, and Alessandro Lazaric. A general sample complexity analysis of vanilla policy gradient. In International Conference on Artificial Intelligence and Statistics, 2022.
> > > > > >
> > > > > > [13] Rui Yuan, Simon Shaolei Du, Robert M. Gower, Alessandro Lazaric, and Lin Xiao. Linear convergence of natural policy gradient methods with log-linear policies. In International Conference on Learning Representations, 2023.
> > > > > >
> > > > > > [14] Antoine Moulin, and Gergely Neu. Optimistic Planning by Regularized Dynamic Programming. In International Conference on Machine Learning, 2023.

---

> > > > > > ### Comment · Reviewer_rns2 · 2023-08-21
> > > > > >
> > > > > > Regarding the plots:
> > > > > >
> > > > > > 1. Sub-linear rate: The original rebuttal did not include the loglog plot, and I am not convinced without seeing the figure. However, if your explanation matches the result, it would be convincing.
> > > > > >
> > > > > > 2. Linear rate: my impression by looking at the existing plots is that you fit a linear line to be above the data. However, this can always be done irrespective of what the data is. It is also always possible to bound the convergence rate in the form $C_1 exp(-tc) + C_0$ for appropriate constants (sufficiently large), so merely the existence of such a bound does not say much, I believe. It would be more convincing if you for example fit the line on the first half of the data, and it continues being valid for the remaining data. But this doesn't seem to be the case for example in Figure 2.(b). For the other plots, it was also not clear to me how the cutoff for training was chosen, and whether the bound would stay valid if you continued training. It is difficult to give an opinion on an additional experiment you did, without seeing the figure.
> > > > > >
> > > > > > Regarding existing papers, intuition, proof sketch:
> > > > > > As you point out, there are regularly papers published with primarily theoretical results. However, there are also theoretical papers published that include experiments or intuitively clear results, and I prefer the latter. My understanding is that conference papers should be mostly self-contained in the main part of the paper with additional information in the appendix. In the current paper, I can not be confident that any of the results presented in the paper are correct without studying the appendix in detail. Consequently, I don't feel I learned anything by reading the main paper. I think the paper would be stronger by including self-contained results in the main paper, e.g., experimental results that are clear, intuitive explanations (e.g., how different parameters or design choices affect the performance with a demonstrated correctness), self-contained proofs of important and interesting sub-results, etc. The proof sketch and further intuitions you provided help, but they were not enough to convince me.

---

> > > > > > > ### Author Response · Authors · 2023-08-21
> > > > > > > **Additional reply**
> > > > > > >
> > > > > > > We thank the reviewer for their quick reply.
> > > > > > >
> > > > > > > Unfortunately, we cannot incorporate further plots at this point. That said, we emphasize once more that the contribution of our work is theoretical and largely consists in the update rule we propose in Algorithm 1. We remain open to answering  specific questions about our results, for instance about Theorem 4.3, should the reviewer have any particular concern.

---

> > > > > > > > ### Comment · Reviewer_rns2 · 2023-08-22
> > > > > > > >
> > > > > > > > Looking at Theorem 4.3 again, I was a concerned whether the added constant term might be too large and make the bound vacuous, so perhaps you can clarify.
> > > > > > > >
> > > > > > > > Assuming positive rewards, the bound is given in the form:
> > > > > > > > $V^* - V \leq C_0 \exp(-tc) + C_1$, where $C_1 = \frac{2(1+\nu_\mu)\sqrt{C_v\varepsilon_{approx}}}{1-\gamma}$.
> > > > > > > >
> > > > > > > > Setting some values like $\gamma = 0.99$, $\nu_\mu=1$, $C_v=1$ and $\varepsilon = 10^{-4}(V^*)^2$, gives $C_1 = 4V^*$ making the bound vacuous. The $\gamma$ setting is typical, I believe $C_v$ and $\nu_\mu$ will typically be much larger than 1 (e.g., your recent experimental result indicates $\nu_\mu \approx 9500$), and I believe 1\% relative error in approximating $Q$ (which I assume to be roughly the same size as $V$) is also reasonable.
> > > > > > > >
> > > > > > > > I understand the discussion period is supposed to be over by now, but hopefully it's still possible to receive a response.
> > > > > > > > If you could clarify or contrast this with other bounds in the literature it would help.

---

> > > > > > > > > ### Author Response · Authors · 2023-08-22
> > > > > > > > > **Additional reply**
> > > > > > > > >
> > > > > > > > > We thank the reviewer for giving us the opportunity to address their concerns regarding our theoretical results.
> > > > > > > > >
> > > > > > > > > Terms like $C_1$ are common in the literature on the convergence analysis of RL algorithms. For instance, the results in Section 6 of the seminal paper [1] contain an error floor term that shares with $C_1$ the dependency on $1/(1-\gamma)$ and on the square root of an approximation error. We point the reviewer to the reply to reviewer WBF5, and to Appendices E and F of our paper, for a detailed discussion on the terms that appear in the bounds in Theorem 4.3 and for pointers to previous works that contain similar terms.
> > > > > > > > >
> > > > > > > > > 1. Alekh Agarwal, Sham M. Kakade, Jason D. Lee, and Gaurav Mahajan. On the theory of policy gradient methods: Optimality, approximation, and distribution shift. Journal of Machine Learning Research, 2021.

---

> > > > > > > > > > ### Comment · Reviewer_rns2 · 2023-08-22
> > > > > > > > > >
> > > > > > > > > > Thanks.
> > > > > > > > > >
> > > > > > > > > > Do you also have any comment about the bound being vacuous/having a very conservative error floor?
> > > > > > > > > > Do these bounds (including past works) tell us something about how the algorithms behave in practice?

---

### Official Review · Reviewer_qVps · 2023-08-01

**Soundness:** 3 good
**Presentation:** 4 excellent
**Contribution:** 4 excellent
**Rating:** 7
**Confidence:** 2

**Summary:**

This paper presents a novel framework for analyzing and understanding policy optimization based on mirror descent that can leverage any general parameterizations of the policy class. The new framework induces a new update rule based on mirror descent, and the authors proved the new framework enjoys sublinear and linear convergence for general policy parameterization.



**Strengths:**

- The authors did a great job presenting the problem's background and motivation, and the differences between the new proposed novel framework and previous related literature, which is quite intuitive and easy to understand.
- The authors provided solutions (Sec. 3.1) to solve the new proposed update rule (Line 1 in Algorithm 1) by introducing a new mirror map, which is novel and easy to follow.
- The theoretical analysis for convergence and sample complexity are provided, which makes the whole framework novel and complete.

**Weaknesses:**

- Not necessarily a weak point, but it would be good to provide some empirical examples to explain the motivation or the algorithm.

- Not sure if the current framework can be extended to other settings such as online or offline RL.

**Questions:**

- Can authors illustrate if it is possible to extend the results to the offline RL or other general RL setting?
- I wonder if the policy is deterministic, will it be necessary to introduce the Line 1 update to analyze the general function approximation results?

---

> ### Author Rebuttal · Authors · 2023-08-08
>
> Thank you for your positive and encouraging remarks. Please find the answers to your questions in Experimental results for AMPO and below.
>
> **Application of AMPO to online/offline RL and other settings.**
> AMPO can be applied to online and offline RL settings, and has a great potential to be applied to other RL settings. As of now, and provided that we can evaluate the Q-function, we have obtained that AMPO can be applied to a general discounted MDP using the Bregman projected policy class induced by any parametrized functions, including tabular softmax, log-linear, and neural policy classes (Line 136-138 and Appendix C.1).
> For the online setting, the policy can be evaluated through roll-out (Algorithm 1 in [1]) or temporal difference learning [2], while the distribution $v^t$ in Algorithm 1 and in Assumptions (A1) and (A2) can be chosen as the discounted state visitation distribution of the current policy $d^t_\mu$. The experiments that we provide in the general comment correspond to this setting.
>
> Regarding the offline setting, the policy evaluation step can be done offline, e.g.\ using importance sampling (Section 3.1 in [3]). Moreover, the update in Line 1 of Algorithm 1 and the results in Theorem 4.3 can be applied to the offline setting, which is mentioned in Lines 165 and 264 of our paper, as the distribution $v^t$ is independent of the current policy $\pi^t$. In practice, one can choose $v^t$ to be the state-action distribution induced by an arbitrary behavior policy that generates the data.
> However, adapting AMPO to the offline setting requires further investigation. One of the major challenges of offline RL is dealing with the distribution shifts that stem from the mismatch between the trained policy $\pi^t$ and the behaviour policy. Several methods have been introduced to deal with this issue, such as constraining the current policy to be close to the behavior policy [2]. We leave introducing offline RL techniques in AMPO as future work and will include the above discussion in Appendix A.5.
>
> Additional settings that have been addressed using the policy mirror descent (PMD) framework are mean-field games [4] and constrained MDP [5]. We hope to build on the existing literature for these settings and see whether our results can bring any improvements. We thank the reviewer for pointing out this direction of research, which we will include in the future work section in Appendix A.5.
>
> **Deterministic policies.** The PMD framework that encompasses AMPO is based on the policy gradient theorem that we report in Equation (3). This expression for the gradient only holds when the policy is stochastic, while in the case of deterministic policies the gradient has a different expression, as studied by [6]. For this reason, AMPO in its current formulation is not suited for the deterministic policy setting.
>
> However, one may ensure that the policies outputted by AMPO are deterministic by tuning the mirror map and the step-size. This is possible with softmax policies, induced by the negative entropy mirror map, by sending the step-size to infinity, effectively turning the softmax operator into a max operator.
> Another simple way of introducing deterministic policies in our framework is to choose the mirror map to be the Euclidean norm and to choose the step-size large enough. Doing so will cause the Bregman projection to put all the probability on the action that corresponds to the maximum value of $f^\theta_s$. The policy can be deduced by the expression of the closed form update between Lines 193 and 194 with $\phi (x) = x$. In both cases, it is still necessary to perform Line 1 of Algorithm 1 in order to obtain the updated parametrization function.
>
> 1. Alekh Agarwal, Sham M. Kakade, Jason D. Lee, and Gaurav Mahajan. On the theory of policy gradient methods: Optimality, approximation, and distribution shift. Journal of Machine Learning Research, 2021.
> 2. Gerald Tesauro. Temporal difference learning and TD-Gammon. Communications of the ACM, 1995.
> 3. Sergey Levine, Aviral Kumar, George Tucker, and Justin Fu. Offline Reinforcement Learning: Tutorial, Review, and Perspectives on Open Problems, 2020.
> 4. Batuhan Yardim, Semih Cayci, Matthieu Geist, and Niao He. Policy Mirror Ascent for Efficient and Independent Learning in Mean Field Games. In International Conference on Machine Learning, 2023.
> 5. Dongsheng Ding, Kaiqing Zhang, Jiali Duan, Tamer Başar, and Mihailo R. Jovanović, Natural Policy Gradient Primal-Dual Method for Constrained Markov Decision Processes. Advances in Neural Information Processing Systems, 2020.
> 6. David Silver, Guy Lever, Nicolas Heess, Thomas Degris, Daan Wierstra, and Martin Riedmiller. Deterministic Policy Gradient Algorithms. In International Conference on Machine Learning, 2014.
> 7. Lin Xiao. On the convergence rates of policy gradient methods. Journal of Machine Learning Research, 2022.
> 8. Rui Yuan, Simon Shaolei Du, Robert M. Gower, Alessandro Lazaric, and Lin Xiao. Linear convergence of natural
> policy gradient methods with log-linear policies. In International Conference on Learning Representations, 2023.
> 9. Yanli Liu, Kaiqing Zhang, Tamer Basar, and Wotao Yin. An improved analysis of (variance- reduced) policy gradient and natural policy gradient methods. Advances in Neural Information Processing Systems, 2020.
> 10. Semih Cayci, Niao He, and Rayadurgam Srikant. Linear convergence of entropy-regularized natural policy gradient with linear function approximation, 2021.
> 11. Zaiwei Chen, Sajad Khodadadian, and Siva Theja Maguluri. Finite-sample analysis of off-policy natural actor–critic with linear function approximation. IEEE Control Systems Letters, 2022.

---

> > ### Comment · Reviewer_qVps · 2023-08-20
> > **Response**
> >
> > Thank the authors for the detailed response. I have no other questions and would keep my score.

---

### Author Rebuttal · Authors · 2023-08-08


**Thank you**  We would like to thank the reviewers for taking the time to review our paper and for their careful remarks.
In particular, we thank Reviewer **qVps**, **rns2**, **oNAx** and **zS6s** for recognizing the importance of our contributions, considering that our work is substantial and promising. We also thank Reviewer **rns2** and **zS6s** for acknowledging the technical soundness of our analysis. In this rebuttal, we address a common point raised by the reviewers regarding the empirical validation of our work first and reviewer-specific remarks below. We will integrate all reviewers' suggestions to improve the clarity of the paper. Please note that, following the reviewers' comments, we have implemented AMPO and added a PDF file that contains new empirical results for AMPO. We will refer to these empirical results in our response (Figures 1 and 2).

### Answer for Reviewers **qVps**, **rns2** and **zS6s**: Experimental results for AMPO

As Reviewer **zS6s**  pointed out, the main contribution of our paper is theoretical and comparing the performance of AMPO with fine-tuned existing algorithms is not the focus of our work. However, we also agree with the suggestions of Reviewers **qVps**, **rns2** and **zS6s** to provide a practical implementation of AMPO and to validate our theoretical findings with experiments. As a result, we have implemented AMPO with neural network parametrization and evaluated its performances in the well-known CartPole environment for several hyperparameter choices. We compare these performances with the convergence rates obtained in the AMPO paper and, where possible, with existing convergence rates from the literature such as  [1].



**Implementation details** As Reviewer **oNAx** and **zS6s** mentioned, AMPO is straightforward to implement. Our implementation is based upon the PPO code from PureJaxRL [2], which obtains the estimates of the $Q$-function through generalized advantage estimation (GAE) and performs the policy update using ADAM optimization and mini-batches. To implement AMPO, we (i) replaced the PPO loss with the expression to minimize in Equation (12) from our paper, (ii) replaced the softmax projection with the Bregman projection, (iii) saved the constants $\lambda$ along the sampled trajectories in order to compute Equation (12), and (iv) implemented the case of increasing step-sizes. The implementation details will be included in an additional section dedicated to the experimental results for AMPO in the appendix.

**Empirical evaluation** In the attached pdf, we provide empirical evaluation for 3 different settings of AMPO:
1. constant step-size, 2 hidden layers NN as parametrization function, negative entropy mirror map;
2. increasing step-size, 1 hidden layer NN as parametrization function, negative entropy mirror map;
3. increasing step-size, 2 hidden layers NN as parametrization function, negative entropy and hyperbolic entropy mirror map.

Here, the negative entropy mirror map (Example 3.6) corresponds to choosing $\phi(x) = e^x$ in Equation (12), and the hyperbolic entropy mirror map (Appendix C.2) corresponds to choosing $\phi(x) = \sinh(x)$ in Equation (12). Their derivations are both provided in Appendix C.2. Note that the case of 1 hidden layer NN as parametrization function in Setting (2) corresponds exactly to the theoretical case of shallow NN studied in our work in Section 4.2, and the case of 2 hidden layers NN as parametrization function in Settings (1) and (3) corresponds to the typical case used in practice for the task considered. For each setting, we provide both the performance of a sampled run of AMPO (Figure 1) and the average performance over 20 runs of AMPO (Figure 2). Additionally, we provide the performance of PPO compared against Setting (3), which was found to be the most successful. We note that the value of the optimal policy for the CartPole environment is 500.

From Setting (1), we observe that the error of AMPO with respect to the optimal policy can be bounded by our predicted convergence rate $O(1/T)$ (which is plotted in log-scale as $- \log(t) + c$ for a hand-picked constant $c$ in Figures 1(a) and 2(a)). We also observe that our $O(1/T)$ convergence rate is tighter than the $O(1/\sqrt{T})$ convergence rate previously obtained in the literature [1] (and plotted as $-0.5 \log(t) + c'$ for a hand-picked constant $c'$).

From Settings (2) and (3), we observe that the error of AMPO with respect to the optimal policy can be bounded by the linear convergence rate we predicted, for varying types of parametrization and mirror maps, i.e., expressions of $\phi$. To the best of our knowledge, this is the first time the hyperbolic entropy mirror map has been applied in RL. Lastly, we observe that the performance of AMPO, although not fine-tuned, is competitive when compared to PPO for the Cartpole environment.

We thank the reviewers for suggesting that we look into the empirical aspects of AMPO, as the empirical evaluation we have provided corroborates our theoretical findings on the performance of AMPO, and in our view, justifies further investigations with regards to its fine-tuning. We will include this empirical evaluation in the revised version of the paper. In particular, due to the limited space, we will include the first row of plots (a) and (c) in Figure 2 in the main body of the revised version of our paper, to show the efficiency of AMPO both with constant and increasing step-sizes.

1. Boyi Liu, Qi Cai, Zhuoran Yang, and Zhaoran Wang. Neural trust region/proximal policy optimization attains globally optimal policy. Advances in Neural Information Processing Systems, 2019.
2. Chris Lu, Jakub Grudzien Kuba, Alistair Letcher, Luke Metz, Christian Schroeder de Witt, and Jakob Foerster. Discovered Policy Optimisation. Advances in Neural Information Processing Systems, 2022.

---

### Decision · Program_Chairs · 2023-09-21

**Decision:**

Accept (poster)

**Comment:**

The paper provides the first linear convergence result for a policy gradient-based method under general policy parametrization going beyond the standard tabular or log-linear setting. The contributions are mainly on the theoretical side.  The paper was thoroughly considered by five reviewers, who are knowledgeable and familiar with the topic. Most reviewers agree with the soundness and clarity of the results after the extensive discussion, albeit some doubts on the assumptions made.  There were also some concerns with the lack of empirical validation, which has been adequately addressed by the authors with preliminary experiments provided during the rebuttal.

Therefore, I am recommending for acceptance and ask authors to incorporate reviewers' suggestions and enrich the numerical results in the final version.